# DEEP GENERATIVE SYMBOLIC REGRESSION

**Samuel Holt**
University of Cambridge
sih31@cam.ac.uk

**Zhaozhi Qian**
University of Cambridge
zq224@maths.cam.ac.uk

**Mihaela van der Schaar**
University of Cambridge
The Alan Turing Institute
mv472@cam.ac.uk

## ABSTRACT

Symbolic regression (SR) aims to discover concise closed-form mathematical equations from data, a task fundamental to scientific discovery. However, the problem is highly challenging because closed-form equations lie in a complex combinatorial search space. Existing methods, ranging from heuristic search to reinforcement learning, fail to scale with the number of input variables. We make the observation that closed-form equations often have structural characteristics and invariances (e.g., the commutative law) that could be further exploited to build more effective symbolic regression solutions. Motivated by this observation, our key contribution is to leverage pre-trained deep generative models to capture the intrinsic regularities of equations, thereby providing a solid foundation for subsequent optimization steps. We show that our novel formalism unifies several prominent approaches of symbolic regression and offers a new perspective to justify and improve on the previous ad hoc designs, such as the usage of cross-entropy loss during pre-training. Specifically, we propose an instantiation of our framework, Deep Generative Symbolic Regression (DGSR). In our experiments, we show that DGSR achieves a higher recovery rate of true equations in the setting of a larger number of input variables, and it is more computationally efficient at inference time than state-of-the-art RL symbolic regression solutions.

## 1 INTRODUCTION

Symbolic regression (SR) aims to find a concise equation $f$ that best fits a given dataset $\mathcal{D}$ by searching the space of mathematical equations. The identified equations have concise closed-form expressions. Thus, they are interpretable to human experts and amenable to further mathematical analysis (Augusto & Barbosa, 2000).

Fundamentally, two limitations prevent the wider ML community from adopting SR as a standard tool for supervised learning. That is, SR is only applicable to problems with few variables (e.g., three) and it is very computationally intensive. This is because the *space of equations* grows exponentially with the equation length and has both discrete ($\times, +, \sin$) and continuous (2.5) components. Although researchers have attempted to solve SR by heuristic search (Augusto & Barbosa, 2000; Schmidt & Lipson, 2009; Stinstra et al., 2008; Udrescu & Tegmark, 2020), reinforcement learning (Petersen et al., 2020; Tang et al., 2020), and deep learning with pre-training (Biggio et al., 2021; Kamienny et al., 2022), achieving both high scalability to the number of input variables and computational efficiency is still an open problem.

We believe that learning a good *representation* of the equation is the key to solve these challenges. Equations are complex objects with many unique invariance structures that could guide the search. Simple equivalence rules (such as commutativity) can rapidly build up with multiple variables or terms, giving rise to complex structures that have many equation invariances.

Importantly, these equation equivalence properties have not been adequately reflected in the representations used by existing SR methods. First, existing **heuristic search** methods represent equations as *expression trees* (Jin et al., 2019), which can *only* capture commutativity ($x_1x_2 = x_2x_1$) via swapping the leaves of a binary operator ($\times, +$). However, trees cannot capture many other properties such as distributivity ($x_1x_2 + x_1x_3 = x_1(x_2 + x_3)$). Second, existing **pre-trained encoder-decoder methods** represent equations as *sequences of tokens*, i.e., $x_1 + x_2 \doteq$ ("$x_1$", " $+$ ", "$x_2$"), just as

sentences of words in natural language (Valipour et al., 2021). The sequence representation cannot encode any invariance structure, e.g., $x_1 + x_2$ and $x_2 + x_1$ will be deemed as two different sequences. Finally, existing **RL methods** for symbolic regression do not learn representations of equations. For each dataset, these methods learn a specific *policy network* to generate equations that fit the data well, hence they need to re-train the policy from scratch each time a new dataset $\mathcal{D}$ is observed, which is computationally intensive.

On the quest to apply symbolic regression to a larger number of input variables, we investigate a deep conditional generative framework that attempts to fulfill the following desired properties:
**(P1) Learn equation invariances:** the equation representations learnt should encode both the equation equivalence invariances, as well as the invariances of their associated datasets.
**(P2) Efficient inference:** performing gradient refinement of the generative model should be computationally efficient at inference time.
**(P3) Generalize to unseen variables:** can generalize to unseen input variables of a higher dimension from those seen during pre-training.

To fulfill P1-P3, we propose the Deep Generative Symbolic Regression (**DGSR**) framework. Rather than represent equations as trees or sequences, DGSR *learns* the representations of equations with a deep generative model, which have excelled at modelling complex structures such as images and molecular graphs. Specifically, DGSR leverages pre-trained conditional generative models that correctly encode the equation invariances. The equation representations are learned using a deep generative model that is composed of invariant neural networks and trained using an end-to-end loss function inspired by Bayesian inference. Crucially, this end-to-end loss enables both pre-training and gradient *refinement* of the pre-trained model at inference time, allowing the model to be more computationally efficient (P2) and generalize to unseen input variables (P3).

**Contributions.** Our contributions are two-fold: ① In Section 3, we outline the DGSR framework, that can perform symbolic regression on a larger number of input variables, whilst achieving less inference time computational cost compared to RL techniques (P2). This is achieved by learning better representations of equations that are aware of the various equation invariance structures (P1). ② In section 5.1, we benchmark DGSR against the existing symbolic regression approaches on standard benchmark problem sets, and on more challenging problem sets that have a larger number of input variables. Specifically, we demonstrate that DGSR has a higher recovery rate of the true underlying equation in the setting of a larger number of input variables, whilst using less inference compute compared to RL techniques, and DGSR achieves significant and state-of-the-art true equation recovery rate on the SRBench ground-truth datasets compared to the SRBench baselines. We also gain insight and understanding of how DGSR works in Section 5.2, of how it can discover the underlying true equation—even when pre-trained on datasets where the number of input variables is less than the number of input variables seen at inference time (P3). As well as be able to capture these equation equivalences (P1) and correctly encode the dataset $\mathcal{D}$ to start from a good equation distribution leading to efficient inference (P2).

## 2 PROBLEM FORMALISM

The standard task of a symbolic regressor method is to return a closed-form equation $f$ that best fits a given dataset $\mathcal{D} = \{(\mathbf{X}_i, y_i)\}_{i=1}^n$, i.e., $y_i \approx f(\mathbf{X}_i), \forall i \in [1:n]$, for all samples $i$. Where $y_i \in \mathbb{R}$, $\mathbf{X}_i \in \mathbb{R}^d$ and $d$ is the number of input variables, i.e., $\mathbf{X} = [\mathbf{x}_1, \ldots, \mathbf{x}_d]$.

**Closed-form equations.** The equations that we seek to discover are closed-form, i.e., it can be expressed as a finite sequence of operators $(\times, +, -, \ldots)$, input variables $(x_1, x_2, \ldots)$ and numeric constants $(3.141, 2.71, \ldots)$ (Borwein et al., 2013). We define $f$ to mean the functional form of an equation, where it can have numeric constant placeholders $\beta$'s to replace numeric constants, e.g., $f(x) = \beta_0 x + \sin(x + \beta_1)$. To discover the full equation, we need to infer the functional form and then estimate the unknown constants $\beta$'s, if any are present (Petersen et al., 2020). Equations can also be represented as a sequence of discrete tokens in prefix notation $\bar{f} = [\bar{f}_1, \ldots, \bar{f}_{|\bar{f}|}]$ (Petersen et al., 2020) where each token is chosen from a library of possible tokens, e.g., $[+, -, \div, \times, x_1, \exp, \log, \sin, \cos]$. The tokens $\bar{f}$ can then be instantiated into an equation $f$ and evaluated on an input $\mathbf{X}$. In existing works, the numeric constant placeholder tokens are learnt through a further secondary non-linear optimizer step using the Broyden–Fletcher–Goldfarb–Shanno (BFGS) algorithm (Fletcher, 2013; Biggio et al., 2021). In lieu of extensive notation, we define when evaluating $f$ to also infer any placeholder tokens using BFGS.

**A generative view of SR.** We prvoide a probabilistic interpretation of the data generating process in Figure 1, where we treat the true equation $f$ as a (latent) random variable following a prior distribution $p(f)$. Therefore a dataset can be interpreted as an evaluation of $f$ on sampled points $\mathbf{X} \sim \mathcal{X}$, i.e., $\mathcal{D} = \{(\mathbf{X}_i, f(\mathbf{X}_i))\}_{i=1}^n, f \sim p(f)$. Crucially, SR can be seen as performing probabilistic inference on the posterior distribution $p(f|\mathcal{D})$. Therefore at inference time, it is natural to formulate SR into a maximum a posteriori (MAP) [1] estimation problem, i.e., $f^* = \arg\max_f p(f|\mathcal{D})$. Thus, SR can be solved by: (1) estimating the posterior $p(f|\mathcal{D})$ conditioned on the observations $\mathcal{D}$—with *pre-trained* deep conditional generative models, $p_\theta(f|\mathcal{D})$ with model parameters $\theta$, (2) further *refining* this posterior at inference time and (3) finding the *maximum* a posteriori (MAP) estimate via a discrete search method.

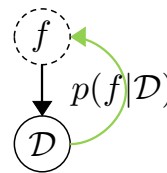

Figure 1: The data generating process.

# 3  DEEP GENERATIVE SR FRAMEWORK

We now outline the Deep Generative SR (DGSR) framework. The key idea is to use both equation and dataset-invariant aware neural networks combined with an end-to-end loss inspired by Bayesian inference. As we shall see in the following, this allows us to learn the equation and dataset invariances (P1), pre-train on a pre-training set and gradient refine on the observed dataset $\mathcal{D}$ at inference time—leading to both a more efficient inference procedure (P2) and generalize to unseen input variables (P3) at inference time. Principally, the framework consists of two steps: (1) a **Pre-training step**, Section 3.1, where an equation and dataset-invariant aware encoder-decoder model learns the posterior distribution $p_\theta(f|\mathcal{D})$ with parameters $\theta$ by *pre-training*, and (2) an **Inference step**, Section 3.2, that uses an optimization method to gradient *refine* this posterior and a discrete search method to find an approximate of the *maximum* of this posterior. For each step, in the following we justify each component in turn, providing the desired properties it must satisfy and provide a suitable instantiation for each component in the overall framework.

## 3.1  PRE-TRAINING STEP

**Learning invariances in the dataset.** We seek to learn the invariances of datasets (P1). Intuitively, a dataset that is defined by a latent (unobserved) equation $f$ should have a representation that is invariant to the number of samples $n$ in the dataset $\mathcal{D}$. Principally, we specify that to achieve this the architecture of the encoder-decoder of the conditional generative model, $p_\theta(f|\mathcal{D})$, should satisfy the following two properties: (1) have an encoding function $h$ that is *permutation invariant* over the encoded input-output pairs $\{(\mathbf{X}_i, y_i)\}_{i=1}^n$ from $g$, and can handle a different number of samples $n$, i.e., $\mathbf{V} = h(\{g(\mathbf{X}_i, y_i)\}_{i=1}^n)$ (Lee et al., 2019). Where $g : \mathcal{X}^d \to \mathcal{Z}^d$ is an encoding function from the individual input variables in $[x_{i1}, \ldots, x_{id}] = \mathbf{X}_i$ of the points in $\mathcal{X}$. (2) Have a decoder that is *autoregressive*, that decodes the latent vector $\mathbf{V}$ to give an output probability of an equation $f$, which allows sampling of equations. Suitable encoder-decoder models (e.g., Transformers (Biggio et al., 2021), RNNs (Sutskever et al., 2014), etc.) can be used that satisfy these two properties. The conditional generative model has parameters $\theta = \{\zeta, \phi\}$, where the encoder has parameters $\zeta$ and the decoder parameters $\phi$, detailed in Figure 2.

Specifically, we instantiate DGSR with a set transformer (Lee et al., 2019) encoder that satisfies (1) and a specific transformer decoder that satisfies (2). This specific transformer decoder leverages the hierarchical tree state representation during decoding (Petersen et al., 2020). Where the encoder that has encoded a dataset $\mathcal{D}$ into a latent vector $\mathbf{V} \in \mathbb{R}^w$ is fed into a transformer decoder (Vaswani et al., 2017). Here, the decoder generates each token of the equation $\bar{f}$ autoregressively, that is, it samples from $p(\bar{f}_i | \bar{f}_{1:(1-i);\theta;\mathcal{D}})$. During sampling of each token, the existing generated tokens $\bar{f}_{1:(1-i)}$ are processed into their hierarchical tree state representation (Petersen et al., 2020) and are encoded with an embedding into an additional latent vector that is concatenated to the encoder latent vector, forming a total latent vector of $\mathbf{U} \in \mathbb{R}^{w+d_s}$ to be used in decoding, where $d_s$ is the additional state dimension. We detail this in Appendix B, and show other architectures can be used in Appendix U.

We pre-train on a pre-training set consisting of $m$ datasets $\{\mathcal{D}^{(j)}\}_{j=1}^m$, where $\mathcal{D}^{(j)}$ is defined by sampling $f^{(j)} \sim p(f)$ from a given prior $p(f)$ (see Appendix J on how to specify $p(f)$). Then, to construct each dataset we evaluate $f^{(j)}$ on $n^{(j)}$ [2] random points in $\mathcal{X}$, i.e., $\mathcal{D}^{(j)} = \{(f^{(j)}(\mathbf{X}_i^{(j)}), \mathbf{X}_i^{(j)})\}_{i=1}^{n^{(j)}}$.

---

[1] We define all acronyms in a glossary in Appendix A.

[2] For generality we note that DGSR can handle datasets of different sample sizes.

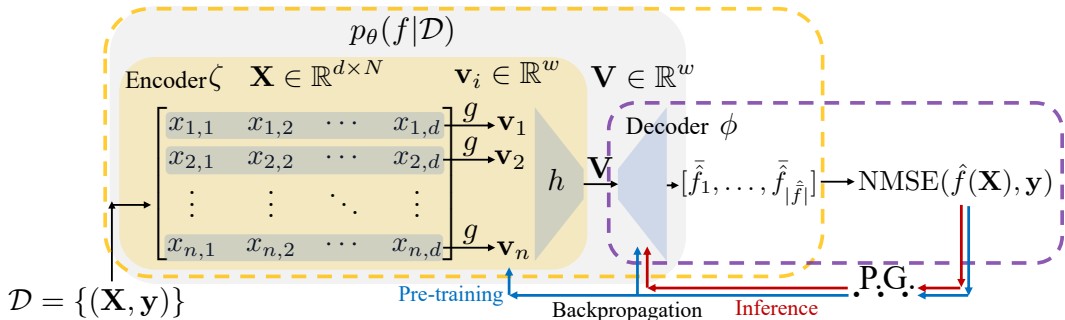

Figure 2: Block diagram of DGSR. DGSR is able to learn the invariances of equations and datasets $\mathcal{D}$ (P1) by having both: **(1)** an encoding architecture that is *permutation invariant* across the number of samples $n$ in the observed dataset $\mathcal{D} = \{(\mathbf{X}_i, y_i)\}_{i=1}^n$, and **(2)** a Bayesian inspired end-to-end loss NMSE function, Eq. 1 from the encoded dataset $\mathcal{D}$ to the outputs from the predicted equations, i.e., NMSE$(\hat{f}(\mathbf{X}), \mathbf{y})$. The highlighted boundaries show the subset of pre-trained encoder-decoder methods and RL methods.

**Loss function.** We seek to learn invariances of equations (P1). Intuitively this is achieved by using a loss where different forms of the same equation have the same loss. To learn *both* the invariances of equations and datasets, we require an end-to-end loss from the observed dataset $\mathcal{D}$ to the predicted outputs from the equations generated, to train the conditional generator, $p_\theta(f|\mathcal{D})$. To achieve this, we maximize the likelihood $p(\mathcal{D}|f)$ distribution under a Monte Carlo scheme to incorporate the prior $p(f)$ (Harrison et al., 2018). A natural and common assumption of the observed datasets is that they are sampled with Gaussian i.i.d. noise (Murphy, 2012). Therefore to maximize this augmented likelihood distribution we minimize the normalized mean squared error (NMSE) loss over a mini-batch of $t$ datasets, where for each dataset $\mathcal{D}^{(j)}$ we sample $k$ [3] equations from the conditional generator, $p_\theta(f|\mathcal{D})$.

$$\mathcal{L}(\theta) = \frac{1}{t}\sum_{j=1}^{t}\frac{1}{k}\sum_{c=1}^{k}\frac{1}{\sigma_{y^{(j)}}}\frac{1}{n^{(j)}}\sum_{i=1}^{n^{(j)}}(y_i^{(j)} - \hat{f}^{(c)}(X_i^{(j)}))^2, \qquad \hat{f}^{(j)} \sim p_\theta(f|\mathcal{D}^{(j)}) \qquad (1)$$

Where $\sigma_{y^{(j)}}$ is the standard deviation of the outputs $\mathbf{y}^{(j)}$. We wish to minimize this end-to-end NMSE loss, Equation 1; however, the process of turning an equation of tokens $\bar{f}$ into an equation $f$ is a non-differentiable step. Therefore, we require an end-to-end non-differentiable optimization algorithm. DGSR is agnostic to the exact choice of optimization algorithm used to optimize this loss, and any relevant end-to-end non-differentiable optimization algorithm can be used. Suitable methods are policy gradient approaches, which include policy gradients with genetic programming (Petersen et al., 2020; Mundhenk et al., 2021). To use these, we reformulate the loss of Equation 1 for each equation into a reward function of $R(\theta) = 1/(1 + \mathcal{L}(\theta))$, that is optimizable using policy gradients (Petersen et al., 2020). Here both the encoder and decoder are trained during pre-training, i.e., optimizing the parameters $\theta$ and is further illustrated in Figure 2 with a block diagram. We formulate this reward function and optimize it with a vanilla policy gradient method (Williams, 1992), with further pseudocode and details of DGSR pre-training in Appendix D.

## 3.2 INFERENCE STEP

We seek to have *efficient* inference (P2) and *generalize* to unseen input variables (P3). Intuitively we achieve this by refining our pre-trained conditional generative model to the observed dataset $\mathcal{D}$ at inference time. Thereby, allowing: (1) the model to start from a good initial distribution of the approximate posterior, $p_\theta(f|\mathcal{D})$ by encoding the observed dataset $\mathcal{D}$ which can be further gradient refined in *fewer* steps (P2) and (2) through gradient refinement *generalize* to generate equations that have unseen input variables compared to those seen at inference time (P3). Principally, at inference time DGSR is able to provide a good initial distribution of the approximate posterior, $p_\theta(f|\mathcal{D})$ by encoding the observed dataset $\mathcal{D}$. Then, it uses the same end-to-end NMSE loss in Eq. 1 and a policy gradient optimization algorithm, that of neural guided priority queue training (NGPQT) of Mundhenk et al. (2021), to be competitive to the existing state-of-the-art, detailed in Appendix C and pseudocode in D. Using this optimization algorithm, the initial approximation is converged to a distribution that has a high probability density where the true equation of interest $f^*$ lies. This allows

---

[3]Where $t$ and $k$ are hyperparameters.

Table 1: Comparison of related works. Columns: Learn Eq. Invariances (P1)—can it learn equation invariances? Eff. Inf. Refinement (P2)—can it perform gradient refinement computationally efficiently at inference time (i.e., update the decoder weights)? Generalize unseen vars.? (P3)—can it generalize to unseen input variables from that those seen during pre-training. References:[1](Petersen et al., 2020),[2](Mundhenk et al., 2021),[3](Costa et al., 2020),[4](Biggio et al., 2021),[5](Valipour et al., 2021),[6](d'Ascoli et al., 2022),[7](Kamienny et al., 2022).

| Approach | Methods | Loss | Model | Pre-train $p_\theta(f\|\mathcal{D})$? | Learn Eq. Invariances? (P1) | Eff. Inf. Refinement (P2) | Generalize unseen? (P3) |
|---|---|---|---|---|---|---|---|
| RL | [1,2,3] | NSME$(\hat{f}(X_i)), y_i)$ | $p_\theta(f)$ | ✗ | ✗ | ✗- Train from scratch | - |
| Encoder | [4,5,6,7] | CE$(\hat{f}, f^*)$ | $p_\theta(f\|\mathcal{D})$ | ✓ | ✓ | ✗- Cannot gradient refine | ✗ |
| DGSR | **This work** | Eq. 1, NSME$(\hat{f}(X_i)), y_i)$ | $p_\theta(f\|\mathcal{D})$ | ✓ | ✓ | ✓- Can gradient refine | ✓ |

sampled equations drawn from $p_\theta(f|\mathcal{D})$ to have a high probability of generating the true equation $f^*$. We achieve this by only refining the decoder weights $\phi$ (Figure 2) and keeping the encoder weights fixed. Furthermore, we show empirically that other optimization algorithms can be used with an ablation of these in Section 5.2 and Appendix E.

Finally, our goal is to find the single best fitting equation for the observed dataset $\mathcal{D}$. We achieve this by using a *discrete search* method to find the maximum a posteriori estimate of the refined posterior, $p_\theta(f|\mathcal{D})$. This is achieved by a simple Monte Carlo scheme that samples $k$ equations and scores each one based on its (NMSE) fit, then returns the equation with the best fit. Principally, there exist other discrete search methods that can be used as well, e.g., Beam search (Steinbiss et al., 1994).

## 4 RELATED WORK

In the following we review the existing deep SR approaches, and summarize their main differences in Table 1. We provide an extended discussion of additional related works, including heuristic-based methods and methods that use a prior in Appendix F. We illustrate in Figure 2 that RL and pre-trained encoder-decoder methods can be seen as ad hoc subsets of the DGSR framework.

**RL methods.** These works use a *policy network*, typically implemented with RNNs, to output a sequence of tokens (*actions*) to form an equation. The output equation obtains a *reward* based on some goodness-of-fit metric (e.g., RMSE). Since the tokens are discrete, the method uses policy gradients to train the policy network. Most existing works focus on improving the pioneering policy gradient approach for SR (Petersen et al., 2020; Costa et al., 2020; Landajuela et al., 2021), however the policy network is randomly initialized and tends to output ill-informed equations at the beginning, which slows down the procedure. Furthermore, the policy network needs to be re-trained each time a new dataset $\mathcal{D}$ is available.

**Hybrid RL and GP methods.** These methods combine RL with genetic programming (GPs). Mundhenk et al. (2021) use a policy network to seed the starting population of a GP algorithm, instead of starting with a random population as in a standard GP. Other works use RL to adjust the probabilities of genetic operations (Such et al., 2017; Chang et al., 2018; Chen et al., 2018; Mundhenk et al., 2021; Chen et al., 2020). Similarly, these methods cannot improve with more learning from other datasets and have to re-train their models from scratch, making inference slow at test time.

**Pre-trained encoder-decoder methods.** Unlike RL, these methods pre-train an encoder-decoder neural network to model $p(f|\mathcal{D})$ using a curated dataset (Biggio et al., 2021). Specifically, Valipour et al. (2021) propose to use standard language models, e.g., GPT. At inference time, these methods *sample* from $p_\theta(f|\mathcal{D})$ using the pre-trained network, thereby achieving low complexity at inference— that is efficient inference. These methods have two key limitations: (1) they use cross-entropy (CE) loss for pre-training and (2) they cannot *gradient refine* their model, leading to sub-optimal solutions. First (1), cross entropy, whilst useful for comparing categorical distributions, does not account for equations that are equivalent mathematically. Although prior works, specifically Lample & Charton (2019), observed the "surprising" and "very intriguing" result that sampling multiple equations from their pre-trained encoder-decoder model yielded some equations that are equivalent mathematically, when pre-trained using a CE loss. Furthermore, the pioneering work of d'Ascoli et al. (2022) has shown this behavior as well. Whereas using our proposed end-to-end NMSE loss, Eq. 1—i.e., will have the same loss value for different equivalent equation forms that are mathematically equivalent—therefore this loss is a natural and principled way to incorporate the equation equivalence property, inherent to symbolic regression. Second (2), DGSR is to the best of our knowledge the first SR method to be able to perform *gradient refinement* of a pre-trained encoder-decoder model using our end-to-end NMSE loss, Eq. 1—to update the weights of the decoder at inference time. We note

that there exists other non-gradient refinement approaches, that cannot update their decoder's weights. These consist of: (1) optimizing the constants in the generated equation form with a secondary optimization step (commonly using the BFGS algorithm) (Petersen et al., 2020; Biggio et al., 2021), and (2) using the MSE of the predicted equation(s) to guide a beam search sampler (d'Ascoli et al., 2022; Kamienny et al., 2022). As a result, to generalize to equations with a greater number of input variables pre-trained encoder-decoder methods require large pre-training datasets (e.g., millions of datasets (Biggio et al., 2021)), and even larger generative models (e.g., $\sim 100$ million parameters (Kamienny et al., 2022)).

## 5 EXPERIMENTS AND EVALUATION

We evaluate DGSR on a set of common equations in natural sciences from the standard SR benchmark problem sets and on a problem set with a large number of input variables ($d = 12$).

**Benchmark algorithms.** We compare against Neural Guided Genetic Programming (**NGGP**) Mundhenk et al. (2021); as this is the current state-of-the-art for SR, superseding DSR (Petersen et al., 2020). We also compare with genetic programming (**GP**) (Fortin et al., 2012) which has long been an industry standard and compare with Neural Symbolic Regression that Scales (**NESYMRES**), an pre-trained encoder-decoder method. We note that NESYMRES was only pre-trained on a large three input variable dataset, and thus can only be used and is included on problem sets that have $d \leq 3$. Further details of model selection, hyperparameters and implementation details are in Appendix G [4].

**Dataset generation.** Each symbolic regression "problem set" is defined by the following: a set of $\omega$ unique ground truth equations—where each equation $f^*$ has $d$ input variables, a domain $\mathcal{X}$ over which to sample $10d$ input variable points (unless otherwise specified) and a set of allowable tokens. For each equation $f^*$ an inference time training and test set are sampled independently from the defined problem set domain, each of $10d$ input-output samples, to form a dataset $\mathcal{D} = \{\mathbf{X}_i, f^*(\mathbf{X}_i)\}_{i=1}^{10d}$. The training dataset is used to optimize the loss at inference time and the test set is only used for evaluation of the best equations found at the end of inference. Inference runs for 2 million equation evaluations, unless the true equation is found early—stopping the procedure. To construct the pre-training set $\{\mathcal{D}^{(j)}\}_{j=1}^{m}$, we use the concise equation generation method of Lample & Charton (2019). This uses the library of tokens for a particular problem set and is detailed further in Appendix J, with details of training and how to specify $p(f)$.

**Benchmark problem sets.** We note that we achieve similar performance to the standard SR benchmark problem sets in Appendix H and therefore seek to evaluate DGSR on more challenging SR benchmarks with more variables ($d \geq 2$), whilst benchmarking on realistic equations that experts wish to discover. We use equations from the Feynman SR database (Udrescu & Tegmark, 2020), to provide more challenging equations of a larger number of input variables. These are derived from the *Feynman Lectures on Physics* (Feynman et al., 1965). We randomly selected a subset of $\omega = 7$ equations with two input variables (Feynman $d = 2$), and a further, more challenging, subset of $\omega = 8$ equations with five input variables (Feynman $d = 5$). Additionally, we sample an additional Feynman dataset of $\omega = 32$ equations with $d = \{3, 4, 6, 7, 8, 9\}$ input variables (Additional Feynman). We also benchmark on SRBench (La Cava et al., 2021), which includes a further $\omega = 133$ equations, of $\omega = 119$ of the Feynman equations and $\omega = 14$ ODE-Strogatz (Strogatz, 2018) equations. Finally, we consider a more challenging problem set consisting of $d = 12$ variables of $\omega = 7$ equations synthetically generated (Synthetic $d = 12$). We detail all problem sets in Appendix I.

**Evaluation.** We evaluate against the standard symbolic regression metric of recovery rate ($A_{\text{Rec}}\%$)— the percentage of runs where the true equation $f^*$ was found, over a set number of $\kappa$ random seed runs (Petersen et al., 2020). This uses the strictest definition of symbolic equivalence, by a computer algebraic system (Meurer et al., 2017). We also evaluate the average number of equation evaluations $\gamma$ until the true equation $f^*$ is found. We use this metric as a proxy for computational complexity across the benchmark algorithms, as testing many generated equations is a bottleneck in SR (Biggio et al., 2021; Petersen et al., 2020), discussed further in Appendix K. Unless noted further we follow the experimental setup of Petersen et al. (2020) and use their complexity definition, also detailed in Appendix K. We run all problems $\kappa = 10$ times using a different random seed for each run (unless otherwise specified), and pre-train with 100K generated equations for each benchmark problem set.

---

[4]Additionally, the code is available at https://github.com/samholt/DeepGenerativeSymbolicRegression and have a broader research group codebase at https://github.com/vanderschaarlab/DeepGenerativeSymbolicRegression

Table 2: Average recovery rate ($A_{Rec}\%$) and the average number of equation evaluations $\gamma$ across the benchmark problem sets, with 95 % confidence intervals. Individual rates and equation evaluations are detailed in Appendices L O P Q. Where: $\omega$ is the number of unique equations $f^*$ in a benchmark problem set, $\kappa$ is the number of random seed runs, and $d$ is the number of input variables in the problem set.

| | Problem set | $\omega$ | $d$ | $\kappa$ | DGSR | NGGP | NESYMRES | GP |
|---|---|---|---|---|---|---|---|---|
| Average Rec. | Feynman (d=2) | 7 | 2 | 40 | **85.36 ± 0.69** | **85.71 ± 0.00** | 57.14 ± 0.00 | 50.00 ± 7.20 |
| Rate (%) $A_{Rec}\%$ | Feynman (d=5) | 8 | 5 | 40 | **69.69 ± 3.38** | 63.44 ± 6.64 | NA | 15.00 ± 12.39 |
| | Additional Feynman | 32 | {3,4,6,7,8,9} | 10 | **67.81 ± 4.60** | **67.81 ± 3.00** | NA | - |
| | Synthetic (d=12) | 7 | 12 | 20 | **37.86 ± 5.62** | 28.57 ± 0.00 | NA | 0 ± 0 |
| Average | Feynman (d=2) | 7 | 2 | 40 | 66,404 | 112,798 | 256 | 4,033 |
| Eq. Evals $\gamma$ | Feynman (d=5) | 8 | 5 | 40 | 731,442 | 912,594 | NA | 198,455 |
| | Additional Feynman | 32 | {3,4,6,7,8,9} | 10 | 318,042 | 328,499 | NA | - |
| | Synthetic (d=12) | 7 | 12 | 20 | 271,302 | 828,905 | NA | - |

## 5.1 MAIN RESULTS

The average recovery rate ($A_{Rec}\%$) and the average number of equation evaluations for the benchmark problem sets are tabulated in Table 2. DGSR achieves a higher recovery rate with more input variables, specifically in the problem sets of Feynman $d = 5$, Additional Feynman and Synthetic $d = 12$. We note that NESYMRES achieves the lowest number of equation evaluations, however, suffers from a significantly lower recovery rate.

**Standard benchmark problem sets.** DGSR is state-of-the-art on SRBench (La Cava et al., 2021) for true equation recovery on the ground truth unique equations, with a significant increase of true equation recovery of 63.25% compared to the previous benchmark method of 52.65% in SRBench, Appendix S. DGSR also achieves a new significant state-of-the-art high recovery rate on the R rationals (Krawiec & Pawlak, 2013) problem set with a 10% increase in recovery rate, Appendix H. It also achieves the same performance as state-of-the-art (NGGP) in the standard benchmark problem sets that have a small number of input variables, of the Nguyen (Uy et al., 2011) and Livermore problem sets (Mundhenk et al., 2021) detailed in Appendix H.

## 5.2 INSIGHT AND UNDERSTANDING OF HOW DGSR WORKS

In this section we seek to gain further insight of *how* DGSR achieves a higher recovery rate with a larger number of input variables, whilst having fewer equation evaluations compared to RL techniques. In the following we seek to understand if DGSR is able to: capture these equation equivalences (P1), at refinement perform inference computationally efficiently (P2) and generalize to unseen input variables of a higher dimension from those seen during pre-training (P3).

**Can DGSR capture the equation equivalences? (P1).** To explore if DGSR is learning these equation equivalences, we turn off early stopping and count the number of unique ground truth $f^*$ equivalent equations that are discovered, as shown in Figure 3 (a). Empirically we observe that DGSR is able to correctly capture equation equivalences and exploits these to generate many unique equivalent—yet true equations, with 10 of these tabulated in Table 3. We note that the RL method,

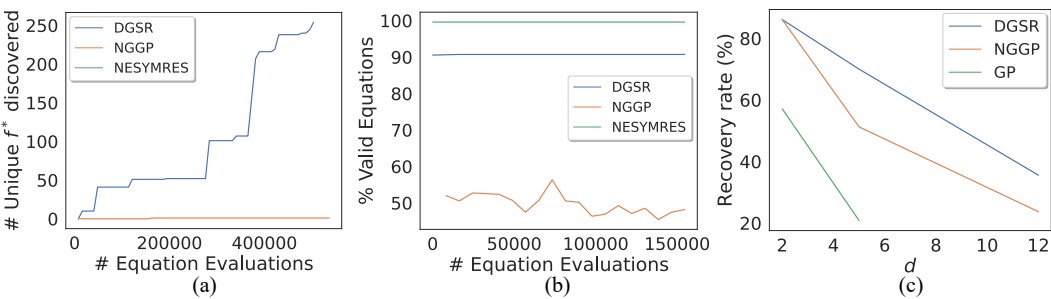

Figure 3: (a) Number of unique ground truth $f^*$ equivalent equations discovered for problem Feynman-7 (A. L), (b) Percentage of valid equations generated from a sample of $k$ for problem Feynman-7 (A. L), (c) Average recovery rate of Feynman $d = 2$, Feynman $d = 5$ and Synthetic $d = 12$ benchmark problem sets plotted against the number of input variables $d$.

Table 3: DGSR equivalent $f^*$ generated equations at inference time, for problem Feynman-7.

| True equation ($f^*$) $\parallel$ | Equivalent generated equations | |
|---|---|---|
| $\frac{3}{2}x_1x_2$ | $x_1(x_2 + \frac{x_2x_2}{x_2+x_2})$ | $x_1(x_2 + \frac{x_2}{\frac{1}{x_2}(x_2+x_2)})$ |
| $\frac{3}{2}x_1x_2$ | $x_2(x_1 + x_2\frac{x_1}{x_2+x_2})$ | $x_2(x_1 + \frac{x_1x_2}{x_2+x_2})$ |
| $\frac{3}{2}x_1x_2$ | $x_1(x_2\frac{x_2}{x_2+x_2} + x_2)$ | $x_2(x_1\frac{x_1}{x_1+x_1} + x_1)$ |
| $\frac{3}{2}x_1x_2$ | $x_1(x_2\frac{x_2}{x_2+x_2} + x_2)$ | $x_2(x_1 + x_2\frac{x_1}{x_2+x_2})$ |
| $\frac{3}{2}x_1x_2$ | $x_1(x_2\frac{x_1}{x_1+x_1} + x_2)$ | $x_2(x_1\frac{x_2}{x_2+x_2} + x_1)$ |
| $\frac{3}{2}x_1x_2$ | $x_1(x_2\frac{x_1}{x_1+x_1} + x_2)$ | $x_1(x_1\frac{x_2}{x_1+x_1} + x_2)$ |
| $\frac{3}{2}x_1x_2$ | $x_2(x_1 + \frac{x_1}{\frac{1}{x_1}(x_1+x_1)})$ | $x_2(x_1 + x_2\frac{x_1}{x_2+x_2})$ |
| $\frac{3}{2}x_1x_2$ | $x_2(x_1\frac{x_1}{x_1+x_1} + x_1)$ | $x_2(x_1 + \frac{x_1}{\frac{1}{x_2}(x_2+x_2)})$ |
| $\frac{3}{2}x_1x_2$ | $x_2(x_1\frac{x_2}{x_2+x_2} + x_1)$ | $x_1(x_2\frac{x_2}{x_2+x_2} + x_2)$ |
| $\frac{3}{2}x_1x_2$ | $x_2(x_1\frac{x_2}{x_2+x_2} + x_1)$ | $x_1(x_2 + (x_2 + x_1\frac{-x_2}{x_1+x_1}))$ |

Figure 4: (a-b) Pareto front of test NMSE against equation complexity. Labelled: (a) Feynman-8, (b) Feynman-13. Ground truth equation complexity is the red line. Equations discovered are listed in A. N. (c) Negative log-likelihood of the ground truth true equation $f^*$ for problem Feynman-7 (A. L).

NGGP is also able to find the true equation. Furthermore, we highlight that all these equations are equivalent achieving zero test NMSE and can be simplified into $\tilde{f}^*$, detailed further in Appendix M.

Moreover, DGSR is able to learn how to generate valid equations more readily. This is important as an SR method needs an equation to be valid for it to be evaluated, that is, one where the generated tokens $\bar{f}$ can be instantiated into an equation $f$ and evaluated (e.g., $\log(-2x_1^2)$ is not valid). Figure 3 (b) shows that DGSR has *learnt* how to generate valid equations, that also have a high probability of containing the true equation $f^*$. Whereas the RL method, NGGP generates mostly invalid equations. We note that the pre-trained encoder-decoder method, NESYMRES generates almost perfectly valid equations—however struggles to produce the true equation $f^*$ in most problems, as seen in Table 2.

We analyze some of the most challenging equations to recover, that all methods failed to find. We observe that DGSR can still find good fitting equations that are concise, i.e., having a low test NMSE with a low equation complexity. A few of these are shown with Pareto fronts in Figure 4 and in Appendix N. We highlight that for a good SR method we wish to determine concise and best fitting equations. Otherwise, it is undesirable to over-fit with an equation that has many terms, having a high complexity—that fails to generalize well.

Additionally, we analyze the challenging real-world setting of low data samples with noise, in Appendix L. Here we observe that DGSR is state-of-the-art with a significant average recovery rate increase of at least $10\%$ better than that of NGGP in this setting, and reason that DGSR is able to exploit the encoded prior $p(f)$.

Furthermore, we perform an ablation study to investigate how useful pre-training and an encoder is for recovering the true equation, in Table 4. This demonstrates empirically for DGSR pre-training increases the recovery rate of the true equation, and highlights that the decoder also benefits from pre-training implicitly modelling $p(f)$ without the encoder. We also ablate pre-training DGSR with a cross-entropy loss on the output of the decoder instead and observe that an end-to-end NMSE loss benefits the recovery rate. This supports our belief that with our invariant aware model and end-to-end loss, DGSR is able to learn the equation and dataset invariances (P1) to have a higher recovery rate.

Table 4: DGSR ablation study using average recovery rate ($A_{\text{Rec}}\%$) on the Feynman $d = 5$ benchmark problem set. Where $d$ is the number of input variables.

| Study | Config | | Average recovery rate (%) $A_{\text{Rec}}\%$ |
|---|---|---|---|
| Pre-training | Pre-trained ✓ | Encoder ✓ | **67.50** |
| & Encoder | Pre-trained ✓ | Encoder ✗ | 66.66 |
| | Pre-trained ✗ | Encoder ✗ | 60.41 |
| Pre-training Loss | NMSE | | **67.50** |
| | Cross Entropy | | 62.50 |
| Pre-trained dataset $(d = 5) = (d_{\text{inference}} = 5)$ | | | **67.50** |
| Pre-trained dataset $(d = 2) < (d_{\text{inference}} = 5)$ | | | 61.29 |

**Can DGSR perform computationally efficient inference? (P2).** We wish to understand if our pre-trained conditional generative model, $p_\theta(f|\mathcal{D})$ can encode the observed dataset $\mathcal{D}$ to start with a good *initial* distribution that is further refined. We do this by evaluating the negative log-likelihood of the true equation $f^*$ during inference, as plotted in Figure 4 (c). We observe that DGSR finds the true equation $f^*$ in few equation evaluations, by correctly conditioning on the observed dataset $\mathcal{D}$ to start with a distribution that has a high probability of sampling $f^*$, which is then further refined. This also indicates that DGSR has learnt a better representation of the true equation $f^*$ (P1) where equivalent equation forms are inherently represented compared to the pre-trained encoder-decoder method, NESYMRES which can only represent one equation form. In contrast, NGGP starts with a random initial equation distribution and eventually converges to a suitable distribution, however this requires a greater number of equation evaluations. Here the pre-trained encoder-decoder method, NESYMRES is unable to refine its equation distribution model. This leads it to have a constant probability of sampling the true equation $f^*$, which in this problem is too low to be sampled and was not discovered after the maximum of 2 million equations sampled. We note that in theory, one could obtain the true equation $f^*$ via an uninformed random search, however this would take a prohibitively large amount of equation samples and hence equation evaluations to be feasible.

Furthermore, DGSR is capable of being used with other optimizers, and show this in Figure 5, where it uses the optimizer of Petersen et al. (2020). This is an ablated version of the optimizer from NGGP; that is a policy gradient method without the GP component. Empirically we demonstrate that using this different optimizer, DGSR still achieves a greater and significant computational inference efficiency compared to RL methods using the same optimizer. Where DGSR uses a total of $\gamma = 29,356$ average equation evaluations compared to the state-of-the-art RL method with $\gamma = 151,231$ average equation evaluations on the Feynman $d = 2$ problem set (Appendix R).

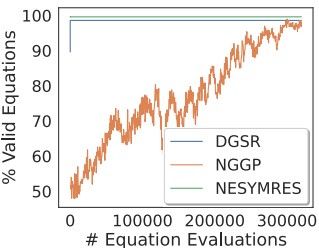

Figure 5: Percentage of valid equations generated from a sample of $k$ equations on the Feynman-7 problem (A. L), with a different optimizer, that of Petersen et al. (2020).

**Can DGSR generalize to unseen input variables of a higher dimension? (P3).** We observe in Table 4 that even when DGSR is pre-trained with a smaller number of input variables than those seen at inference time, it is still able to learn a useful equation representation (P1) that aids generalizing to the unseen input variables of a higher dimension. Here, we verify this by pre-training on a dataset with $d = 2$ and evaluating on the Feynman $d = 5$ problem set.

## 6 Discussion and Future work

We hope this work provides a practical framework to advance deep symbolic regression methods, which are immensely useful in the natural sciences. We note that DGSR has the following limitations of: (1) may fail to discover highly complex equations, (2) optimizing the numeric constants can get stuck in local optima and (3) it assumes all variables in $f$ are observed. Each of these pose exciting open challenges for future work, and are discussed in detail in Appendix V. Of these, we envisage Deep Generative SR enabling future works of tackling even larger numbers of input variables and assisting in the automation of the problem of scientific discovery. Doing so has the opportunity to accelerate the scientific discovery of equations that determine the true underlying processes of nature and the world.

ACKNOWLEDGEMENTS.

SH would like to acknowledge and thank AstraZeneca for funding. This work was additionally supported by the Office of Naval Research (ONR) and the NSF (Grant number: 1722516). Moreover, we would like to warmly thank all the anonymous reviewers, alongside research group members of the van der Scaar lab, for their valuable input, comments and suggestions as the paper was developed—where all these inputs ultimately improved the paper. Furthermore, SH would like to thank G-research for a small grant.

**Ethics Statement.** We envisage DGSR as a tool to *help* human experts discover underlying equations from processes, however emphasize that the equations discovered would need to be further verified by a human expert or in an experimental setting. Furthermore, the data used in this work is synthetically generated from given equation problem sets, and no human-derived data was used.

**Reproducibility Statement.** To ensure reproducibility, we outline in Section 5: (1) the benchmark algorithms used and include their implementation details, including their hyperparameters and how they were selected fully, in Appendix G. (2) How we generated the inference datasets for a single equation $f$ in a problem set of $\omega$ equations and provide full details of the pre-training dataset generation and inference dataset generation in Appendix J. (3) Which benchmark problem sets we used and provide full problem set details, including all equations in a problem set, token set used and the domain to sample $\mathbf{X}$ points from in Appendix I. (4) The evaluation metrics used, how these are computed over random seed runs and detail all of these further in Appendix K. Finally, the code is available at https://github.com/samholt/DeepGenerativeSymbolicRegression.

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

CONTENTS OF SUPPLEMENTARY MATERIALS

IMPLEMENTATION DETAILS:

RELATED WORK AND METHODOLOGY:

RESULTS:

ABLATIONS:

LIMITATIONS AND OPEN CHALLENGES:

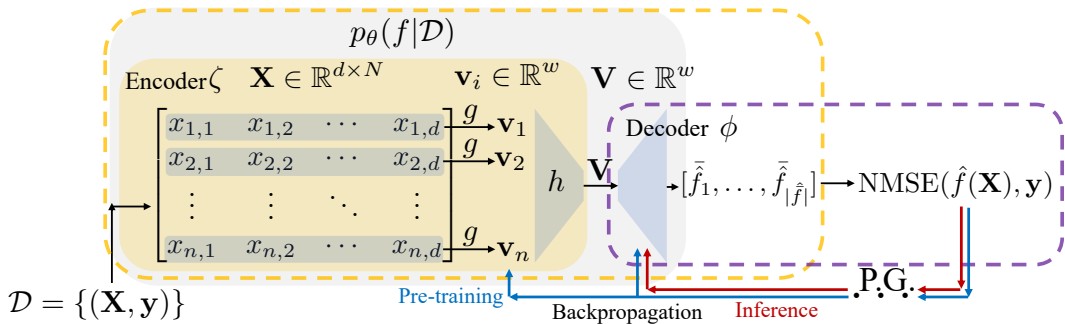

$$\mathcal{D} = \{(\mathbf{X}, \mathbf{y})\}$$

Figure 6: Block diagram of DGSR. DGSR is able to learn the invariances of equations and datasets $\mathcal{D}$ (P1) by having both: **(1)** An encoding architecture that is *permutation invariant* across the number of samples $n$ in the observed dataset $\mathcal{D} = \{(\mathbf{X}_i, y_i)\}_{i=1}^n$, **(2)** An Bayesian inspired end-to-end loss NMSE function, Eq. 1 from the encoded dataset $\mathcal{D}$ to the outputs from the predicted equations, i.e., $\text{NMSE}(\hat{f}(\mathbf{X}), \mathbf{y})$. The highlighted boundaries show the subset of pre-trained encoder-decoder methods and RL methods.

## A GLOSSARY OF TERMS

We provide a short glossary of key terms, in Table 5.

Table 5: Glossary of key terms.

| Term | Definition |
|------|------------|
| SR | Symbolic regression |
| GP | Genetic programming |
| DGSR | Deep generative symbolic regression |
| NGGP | Neural guided genetic programming |
| NESYMRES | Neural Symbolic Regression that Scales |
| DSR | Deep symbolic regression |
| NMSE | Normalized mean squared error |
| Conditional generative model | $p_\theta(f|\mathcal{D})$ |

## B DGSR INSTANTIATION

We outline our instantiation of DGSR with the following architecture, for the conditional generator we split it into two parts that of an encoder and a decoder with total parameters $\theta = \{\zeta, \phi\}$, where the encoder has parameters $\zeta$ and the decoder parameters $\phi$, detailed in Figure 6. See Appendix G for implementation hyperparameters and further details.

**Encoder.** We use a set transformer (Lee et al., 2019) to encode a dataset $\mathcal{D} = \{(\mathbf{X}_i, y_i)\}_{i=1}^n$ into a latent vector $\mathbf{V} \in \mathbb{R}^w$, where $w \in \mathbb{N}$. Defined with $n$ as the number of samples, $y_i \in \mathbb{R}$ and $\mathbf{X}_i \in \mathbb{R}^d$ is of $d$ variable dimension. It is also possible to represent the input float values $\mathcal{D}$ into a multi-hot bit representation according to the half-precision IEEE-754 standard, as is common in some encoder symbolic regression methods (Biggio et al., 2021; Kamienny et al., 2022). We note that in our experimental instantiation we did not represent the inputs in this way and instead, feed the float values directly into the set transformer. Empirically we observed similar performance therefore chose not to include this extra encoding, on our benchmark problem sets evaluated. However, we highlight that the user should follow best practices, as if their problem has observations $\mathcal{D}$ that have drastically different values, the multi-hot bit representation encoding could be useful (Biggio et al., 2021).

**Decoder.** The latent vector, $\mathbf{V} \in \mathbb{R}^w$ is fed into the decoder. We instantiated this with a standard transformer decoder (Vaswani et al., 2017). The decoder generates each token of the equation $\bar{f}$ autoregressively, that is sampling from $p(\bar{f}_i|\bar{f}_{1:(1-i);\theta;\mathcal{D}})$. We process the existing generated tokens $\bar{f}_{1:(1-i)}$ into their hierarchical tree state representation (Petersen et al., 2020), which are provided as

inputs to the transformer decoder, a representation of the parent and sibling nodes of the token being sampled (Petersen et al., 2020). The hierarchical tree state representation is generated according to Petersen et al. (2020) and is encoded with a fixed size embedding and fed into a standard transformer encoder (Vaswani et al., 2017). This generates an additional latent vector which is concatenated to the encoder latent vector, forming a total latent vector of $\mathbf{U} \in \mathbb{R}^{w+d_s}$, where $d_s$ is the additional state dimension. The decoder generates each token sequentially, outputting a categorical distribution to sample from. From this, it is straightforward to apply token selection constraints based on the previously generated tokens. We incorporate the token selection constraints of Mundhenk et al. (2021); Petersen et al. (2020), which includes: limiting equations to a minimum and maximum length, children of an operator should not all be constants, the child of a unary operator should not be the inverse of that operator, descendants of trigonometric operators should not be other trigonometric operators etc. During pre-training and inference, when we encode a dataset $\mathcal{D}$ into the latent vector $\mathbf{V}$ and sample $k$ equations from the decoder. This is achieved by tiling (repeating) the encoded dataset latent vector $k$ times and running the decoder over the respective latent vectors to form a batch size of $k$ equations.

**Optimization Algorithm.** We wish to minimize this end-to-end NMSE loss, Equation 1; however, the process of turning an equation of tokens $\bar{f}$ into an equation $f$ is a non-differentiable step. Therefore, we require an end-to-end non-differentiable optimization algorithm. DGSR is agnostic to the exact choice of optimization algorithm used to optimize this loss, and any relevant end-to-end non-differentiable optimization algorithm can be used. Suitable methods are policy gradients approaches, which include policy gradients with genetic programming (Petersen et al., 2020; Mundhenk et al., 2021). To use these, we reformulate the loss of Equation 1 for each equation into a reward function of $R(\theta) = 1/(1 + \mathcal{L}(\theta))$, that is optimizable using policy gradients (Petersen et al., 2020). Here both the encoder and decoder are trained during pre-training, i.e., optimizing the parameters $\theta$ and is further illustrated in Figure 2 with a block diagram. To be competitive to the existing state-of-the-art we formulate this reward function and optimize it with neural guided priority queue training (NGPQT) of Mundhenk et al. (2021), detailed in Appendix C. Furthermore, we provide pseudocode for DGSR in Appendix D and show empirically other optimization algorithms can be used with an ablation of these in Section 5.2 and Appendix E.

## C  NGPQT OPTIMIZATION METHOD

The work of Mundhenk et al. (2021) introduces the hybrid neural-guided genetic programming method to SR. This uses an RNN (LSTM) for the generator (i.e., decoder) (Petersen et al., 2020) with a genetic programming component and achieves state-of-the-art results on a range of symbolic regression benchmarks (Mundhenk et al., 2021). This optimization method is applicable to any neural network autoregressive model that can output an equation. Equations sampled from the generator are used to seed a starting population of a random restart genetic programming component that gradually learns better starting populations of equations, by optimizing the parameters $\theta$ of the generator. Specifically, this RL optimization method, formulates the generator as a reinforcement learning *policy*. This is optimized over a mini-batch of $t$ datasets, where for each dataset we sample $k$ equations from the conditional generator, $p_\theta(f|\mathcal{D})$. Therefore, we sample a total of $kt$ equations $\mathcal{F}$ per mini-batch. We note that at inference time, we only have one dataset therefore $t = 1$. We evaluate each equation $f \in \mathcal{F}$ under the reward function $R(f) = 1/(1 + \mathcal{L}(f))$ and perform gradient descent on a RL policy gradient loss function (Mundhenk et al., 2021). Here we define $\mathcal{L}(f)$ as the normalized mean squared error (NMSE) for a single equation, i.e.

$$\mathcal{L}(f) = \frac{1}{\sigma_y} \frac{1}{n} \sum_{i=1}^{n} (y_i - f(X_i))^2 \qquad (2)$$

Where $\sigma_y$ is the standard deviation of the observed outputs $\mathbf{y}$. We note that instead of optimizing Equation 1 directly, we achieve the same goal of optimizing Equation 1 by optimizing the reward function $R(f)$ for each equation $f \in \mathcal{F}$ over a batch of $kt$ equations, $\mathcal{F}$.

**Choice of RL policy gradient loss function.** There exist multiple RL policy gradient loss functions that are applicable (Mundhenk et al., 2021). The optimization method of Mundhenk et al. (2021) propose to use *priority queue training* (PQT) (Abolafia et al., 2018). Detailed further, some of these RL policy gradient loss functions are (Mundhenk et al., 2021):

- *Priority Queue Training* (PQT). The generator is trained with a continually updated buffer of the top-$q$ best fitting equations, i.e., a maximum-reward priority queue (MRPQ) of maximum size $q$ (Abolafia et al., 2018). Training is performed over equations in the MRPQ using a supervised learning objective: $\mathcal{L}(\theta) = \frac{1}{q} \sum_{f \in \mathcal{F}} \nabla_\theta \log p_\theta(f|\mathcal{D})$.

- *Vanilla policy gradient* (VPG). Uses the REINFORCE algorithm (Williams, 1992). Training is performed over the equations in the batch $\mathcal{F}$ with the loss function: $\mathcal{L}(\theta) = \frac{1}{k} \sum_{f \in \mathcal{F}} (R(f) - b) \nabla_\theta \log p_\theta(f|\mathcal{D})$, where $b$ is a baseline, defined with an exponentially-weighted moving average (EWMA) of rewards.

- *Risk-seeking policy gradient* (RSPG). Uses a modified VPG to optimize for the best case reward (Petersen et al., 2020) rather than the average reward: $\mathcal{L}(\theta) = \frac{1}{\epsilon k} \sum_{f \in \mathcal{F}} (R(f) - R_\epsilon) \nabla_\theta \log p_\theta(f|\mathcal{D}) \mathbf{1}_{R(f) > R_\epsilon}$, where $\epsilon$ is a hyperparameter that controls the degree of risk-seeking and $R_\epsilon$ is the empirical $(1 - \epsilon)$ quantile of the rewards of $\mathcal{F}$.

Furthermore, we follow Mundhenk et al. (2021) and also include a common additional term in the loss function proportional to the entropy of the distribution at each position along the equation generated (Mundhenk et al., 2021; Petersen et al., 2020). Specifically, these are the same equation complexity regularization methods of Mundhenk et al. (2021), using the hierarchical entropy regularizer and the soft length prior from Landajuela et al. (2021).

The hierarchical entropy regularizer encourages the decoder (decoding tokens sequentially) to perpetually explore early tokens (without getting stuck committing to early tokens during training, dubbed the "early commitment problem") (Landajuela et al., 2021). Whereas the soft length prior discourages the equation from being either too short or too long, which is superior to a hard length prior that forces each equation to be generated between a pre-specified minimum and maximum length. Using a soft length prior, the generator can learn the optimal equation length, which has been shown to improve learning (Landajuela et al., 2021).

The groundbreaking work of Balla et al. (2022), further shows that unit regularization can be added to improve the SR method, and provides a useful decomposition of the complexity regularization into two components of the number of tokens ("activation functions") and the number of numeric constants—whereby it can be beneficial to tune these regularization terms separately. The full analysis of all regularization terms is out of scope for this work, however we leave this as an exciting direction for future work to explore.

For the genetic programming component, we follow the same setup as in Mundhenk et al. (2021). This uses a standard genetic programming formulation DEAP (Fortin et al., 2012), introducing a few improvements, these being: equal probability amongst the mutation types (e.g., uniform, node replacement, insertion and shrink mutation), incorporating the equation constraints from Petersen et al. (2020) (also discussed in Appendix B) and the initial equations population is seeded by that of the generator equation samples.

Unless otherwise specified, we use the same neural guided PQT (NGPQT) optimization method at inference time for DGSR—specifically this the PQT method, and we use the same optimization implementation of Mundhenk et al. (2021) of filtering the equations first by the empirical $(1 - \epsilon)$ quantile of the rewards of $\mathcal{F}$, and then second filtering these for the top-$q$ best fitting equations for use in PQT. Furthermore, we pre-train with the optimization method of VPG.

## D  DGSR PSEUDOCODE AND SYSTEM DESCRIPTION

We outline the DGSR system with Figure 6. The conditional generative model, $p_\theta(f|\mathcal{D})$ is comprised of an encoder and a decoder, with total parameters $\theta$. Specifically the parameters of the encoder $\zeta$ and decoder $\phi$ are a subset of the total model parameters, i.e., $\theta = \{\zeta, \phi\}$. During pre-training we update the parameters for both the encoder and the decoder, that is $\theta$, whilst at inference time we only update the parameters of the decoder $\phi$. We denote the best equation found during inference as $f^a$, not be confused with the true underlying equation $f^*$ for that problem. If DGSR identifies the true equation then $f^a$ is equivalent to $f^*$, i.e., $f^a = f^*$ (Mundhenk et al., 2021). The pre-training pseudocode for DGSR is detailed in Algorithm 1 and the inference pseudocode in Algorithm 2. For comprehensiveness we repeat the pre-training and inference training details here, however, note these

details can also be found in Appendix J, with further details of the loss optimization methods in Appendix C.

**Pre-training.** Using the specifications of $p(f)$, we can generate an almost unbounded number of equations. We pre-compile 100K equations and train the conditional generator on these using a mini-batch of $t$ datasets, following Biggio et al. (2021). The overall pre-training algorithm is detailed in Algorithm 1. During pre-training we use the vanilla policy gradient (VPG) loss function to train all the conditional generator parameters $\theta$ (i.e., the encoder and decoder parameters). This is optimized over a mini-batch of $t$ datasets, where for each dataset we sample $k$ equations from the conditional generator, $p_\theta(f|\mathcal{D})$. Therefore, we sample a total of $kt$ equations $\mathcal{F}$ per mini-batch. For each equation $f \in \mathcal{F}$ we compute the normalized mean squared error (NMSE), i.e.

$$\mathcal{L}(f) = \frac{1}{\sigma_y}\frac{1}{n}\sum_{i=1}^{n}(y_i - f(X_i))^2, \qquad\qquad f \sim p_\theta(f|\mathcal{D}) \qquad (3)$$

Where $\sigma_y$ is the standard deviation of the observed outputs $\mathbf{y}$. We formulate each equations NMSE loss as a reward function by the following $R(f) = 1/(1 + \mathcal{L}(f))$. Training is performed over the equations in the batch $\mathcal{F}$ with the vanilla policy gradient loss function of,

$$\mathcal{L}(\theta) = \frac{1}{k}\sum_{f\in\mathcal{F}}(R(f) - b)\nabla_\theta \log p_\theta(f|\mathcal{D}) \qquad (4)$$

Where $b$ is a baseline, defined with an exponentially-weighted moving average (EWMA) of rewards, i.e., $b_t = \alpha\,\mathbb{E}[R(f)] + (1-\alpha)b_{t-1}$. Furthermore, we follow Mundhenk et al. (2021) and also include a common additional term in the loss function proportional to the entropy of the distribution at each position along the equation generated (Mundhenk et al., 2021; Petersen et al., 2020). Specifically, these are the same equation complexity regularization methods of Mundhenk et al. (2021), using the hierarchical entropy regularizer and the soft length prior from Landajuela et al. (2021).

---

**Algorithm 1** Deep Generative Symbolic Regression Pre-training

---

**Input** mini-batch size of $t$ datasets to use in training, batch size of $k$ equations to sample per dataset, number of epochs, prior of equations $p(f)$, domain $\mathcal{X}$, loss function $\mathcal{L}(\theta)$ for training the generator, including corresponding hyperparameters (e.g., EWMA coefficient for VPG, risk factor for RSPG, priority queue size for PQT)
**Output** Pre-trained conditional generator $p_\theta(f|\mathcal{D})$

    Initialize conditional generator with parameters $\theta$, defining the posterior $p_\theta(f|\mathcal{D})$
    **for** $e$ in $\{1,\ldots,\text{epochs}\}$ **do**
        $L \leftarrow 0$
        **for** j in $\{1,\ldots,t\}$ **do**
            $f^* \sim p(f)$
            $\mathbf{X} \sim \mathcal{X}$
            $\mathcal{D} \leftarrow \{(f^*(\mathbf{X}), \mathbf{X})\}$
            $\mathcal{F}_{\text{generator}} \leftarrow \{f^{(i)} \sim p_\theta(f|\mathcal{D})\}_{i=1}^{k}$   ▷ Sample $k$ equations from the conditional generator
            $\mathcal{R} \leftarrow \{R(f,\mathcal{D})\forall f \in \mathcal{F}_{\text{generator}}\}$   ▷ Compute rewards
            $L \leftarrow L + \mathcal{L}(\theta)$   ▷ Compute the generator loss (e.g., using VPG)
        **end for**
        $\theta \leftarrow \theta + \nabla_\theta L$   ▷ Train the generator over the batch of $t$ datasets
    **end for**

---

**Inference training.** We detail the inference training routine in Algorithm 2. A training set and a test are sampled independently from the observed dataset $\mathcal{D}$. The training dataset is used to optimize the loss at inference time and the test set is only used for evaluation of the best equations found at the end of inference, which unless the true equation $f^*$ is found runs for 2 million equation evaluations on the training dataset. During inference we use the neural guided PQT (NGPQT) optimization method, outlined in Appendix C, and only train the decoder in the conditional generative model, i.e., only the decoders parameters $\phi$ are updated. To construct the loss function we sample a batch of $k$ equations from the conditional generative model $p_\theta(f|\mathcal{D})$ and use these to seed a genetic programming component (DEAP (Fortin et al., 2012)). The genetic programming component evaluates the equations fitness by its individual reward by computing the equations NMSE and then

the reward of that loss (i.e., using $R(f) = 1/(1 + \mathcal{L}(f))$ again). After a pre-defined number of genetic programming rounds (defined by a hyperparameter), we join the initial generated equations from the conditional generator to those of the best equations identified by the former genetic programming component. We then use this set of equations to train the priority queue training (PQT) loss function, and only update the parameters of the decoder. To construct the PQT loss, we continually update a buffer of the top-$q$ best fitting equations, i.e., a maximum-reward priority queue (MRPQ) of maximum size $q$ (Abolafia et al., 2018). Training is performed over equations in the MRPQ using a supervised learning objective:

$$\mathcal{L}(\phi) = \frac{1}{q} \sum_{f \in \mathcal{F}} \nabla_\phi \log p_\phi(f|\mathcal{D}) \tag{5}$$

Additionally, we also include a common additional term in the loss function proportional to the entropy of the distribution at each position along the equation generated (Mundhenk et al., 2021; Petersen et al., 2020). Specifically, these are the same equation complexity regularization methods of Mundhenk et al. (2021), using the hierarchical entropy regularizer and the soft length prior from Landajuela et al. (2021). Furthermore, we keep track of the best equation seen, by the highest reward, denoted as $f^a$.

---

**Algorithm 2** Deep Generative Symbolic Regression Inference

**Input** batch size of $k$ equations to sample, observed dataset $\mathcal{D}$, loss function $\mathcal{L}(\theta)$ for training the generator, including corresponding hyperparameters (e.g., EWMA coefficient for VPG, risk factor for RSPG, priority queue size for PQT), pre-trained conditional generator $p_\theta(f|\mathcal{D})$
**Output** Best fitting equation $f^a$ for observed dataset $\mathcal{D}$

    Load pre-trained conditional generator with parameters $\theta$, defining the posterior $p_\theta(f|\mathcal{D})$
    **while** total equation evaluations below budget **do**
        $\mathcal{F}_{\text{generator}} \leftarrow \{f^{(i)} \sim p_\theta(f|\mathcal{D})\}_{i=1}^k$         $\triangleright$ Sample $k$ equations from the conditional generator
        $\mathcal{F}_{\text{GP}} \leftarrow \text{GP}(\mathcal{F}_{\text{generator}})$         $\triangleright$ Seed GP component, as defined in Appendix C
        $\mathcal{F}_{\text{train}} \leftarrow \mathcal{F}_{\text{generator}} \cup \mathcal{F}_{\text{GP}}$         $\triangleright$ Join generated equations and best GP equations
        $\mathcal{R} \leftarrow \{R(f) \forall f \in \mathcal{F}_{\text{train}}\}$         $\triangleright$ Compute rewards
        $\theta \leftarrow \phi + \nabla_\phi \mathcal{L}(\phi)$         $\triangleright$ Train the generator (e.g., using PQT)
        **if** $\max \mathcal{R} > R(f^a)$ **then** $f^a \leftarrow f^{\arg \max \mathcal{R}}$         $\triangleright$ Update the best equation seen
    **end while**

---

**Advantage over cross entropy loss**. Specifically using an end-to-end-loss during pre-training and inference is key, as the conditional generator is able to *learn* and *exploit* the unique equation invariances and equivalent forms. For example two equivalent equations that have different forms (e.g.,$x_1(x_2 + \sin(x_3)) = x_1 x_2 + x_1 \sin(x_3)$) will still have an identical NMSE loss (Equation 1). Whereas existing pre-training methods that pre-train using a cross entropy loss $\mathcal{L}_{\text{CE}}(f^*, \hat{f})$, between the known ground truth true equation $f^*$ and the predicted equation $\hat{f}$, have a non-identical loss, failing to capture equational equivalence relations. Furthermore the existing pre-training methods trained in this way require the ground truth equation $f^*$ to train their conditional generative model. However this is unknown at inference time (as it is this that we wish to find), therefore they cannot update their posterior at inference and are limited to only sample from it. We empirically illustrate this in Figure 4 (c). Given this, the existing pre-trained encoder-decoder methods require an exponentially larger model and pre-training dataset size when pre-training on a dataset of increasing covariate dimension (Kamienny et al., 2022).

## E  OTHER OPTIMIZATION ALGORITHMS

DGSR supports other optimization algorithms that can optimize a non-differentiable loss function. It is common to reformulate the NMSE loss per equation into a reward via $R(f) = 1/(1 + \mathcal{L}(f))$, which can then be optimized using policy gradients (Petersen et al., 2020). Suitable policy gradient algorithms are outlined in Appendix C, which include vanilla policy gradients, risk-seeking policy gradients and priority queue training. We note that it is possible to use DGSR with other RL optimization methods such as distributional RL optimization Bellemare et al. (2017). See Appendix R for other optimization results, without the genetic programming component.

## F    EXTENDED RELATED WORK

Table 6: Comparison of related works. Columns: Learn Eq. Invariances (P1)—can it learn equation invariances? Eff. Inf. Refinement (P2)—can it perform gradient refinement computationally efficiently at inference time (i.e., update the decoder weights)? Generalize unseen vars. ? (P3)—can it generalize to unseen input variables from that those seen during pre-training. References:[1](Petersen et al., 2020),[2](Mundhenk et al., 2021),[3](Costa et al., 2020),[4](Jin et al., 2019),[5](Biggio et al., 2021),[6](Valipour et al., 2021),[7](d'Ascoli et al., 2022),[8](Kamienny et al., 2022).

| Approach | Methods | Loss | Model | Pre-train $p_\theta(f\|\mathcal{D})$ ? | Learn Eq. Invariances ? (P1) | Eff. Inf. Refinement (P2) | Generalize unseen ? (P3) |
|---|---|---|---|---|---|---|---|
| RL | [1,2,3] | NSME($\hat{f}(X_i)), y_i$) | $p_\theta(f)$ | ✗ | ✗ | ✗- Train from scratch | - |
| Prior | [4] | RSME($\hat{f}(X_i)), y_i$) | $p_\theta(f)$ | ✗ | ✗ | ✗- Train from scratch | - |
| Encoder | [5,6,7,8] | CE($\hat{f}, f^*$) | $p_\theta(f\|\mathcal{D})$ | ✓ | ✗ | ✗- Cannot gradient refine | ✗ |
| DGSR | **This work** | Eq. 1, NSME($\hat{f}(X_i)), y_i$) | $p_\theta(f\|\mathcal{D})$ | ✓ | ✓ | ✓- Can gradient refine | ✓ |

In the following we review the existing deep SR approaches, and summarize their main differences in Table 1. We provide an extended discussion of additional related works, including heuristic-based methods in Appendix F. We illustrate in Figure 2 that RL and pre-trained encoder-decoder methods can be seen as ad hoc subsets of the DGSR framework.

In the following, we provide an extended discussion of related works, including the additional related work of heuristic-based methods.

**RL methods.** These works use a *policy network*, typically implemented with RNNs, to output a sequence of tokens (*actions*) in order to form an equation. The output equation obtains a *reward* based on some goodness-of-fit metric (e.g., RMSE). Since the tokens are discrete, the method uses policy gradients to train the policy network. Most existing works focus on improving the pioneering policy gradient approach for SR, that of Petersen et al. (2020) (Costa et al., 2020; Landajuela et al., 2021). However, the policy network is randomly initialized (without pre-training) and tends to output ill-informed equations at the beginning, which slows down the procedure. Furthermore, the policy network needs to be re-trained each time a new dataset $\mathcal{D}$ is available.

**Hybrid RL and GP methods.** These methods combine RL with genetic programming (GPs). Mundhenk et al. (2021) use a policy network to seed the starting population of a GP algorithm, instead of starting with a random population as in a standard GP. Other works use RL to adjust the probabilities of genetic operations (Such et al., 2017; Chang et al., 2018; Chen et al., 2018; Mundhenk et al., 2021; Chen et al., 2020). Similarly, these methods cannot improve with more learning from other datasets and have to re-train the model from scratch, making inference slow at test time.

**Pre-trained encoder-decoder methods.** Unlike RL, these methods pre-train an encoder-decoder neural network to model $p(f|\mathcal{D})$ using a curated dataset (Biggio et al., 2021). Specifically, Valipour et al. (2021) propose to use standard language models, e.g., GPT. At inference time, these methods *sample* from $p_\theta(f|\mathcal{D})$ using the pre-trained network, thereby achieving low complexity at inference— that is efficient inference. These methods have two key limitations: (1) they use cross-entropy (CE) loss for pre-training and (2) they cannot *gradient refine* their model, leading to sub-optimal solutions. First (1), cross entropy, whilst useful for comparing categorical distributions, does not account for equations that are equivalent mathematically. Although prior works, specifically Lample & Charton (2019), observed the "surprising" and "very intriguing" result that sampling multiple equations from their pre-trained encoder-decoder model yielded some equations that are equivalent mathematically, when pre-trained using a CE loss. Furthermore, the pioneering work of d'Ascoli et al. (2022) has shown this behavior as well. Whereas using our proposed end-to-end NMSE loss, Eq. 1—will have the same loss value for different equivalent equation forms that are mathematically equivalent—therefore this loss is a natural and principled way to incorporate the equation equivalence property, inherent to symbolic regression. Second (2), DGSR is to the best of our knowledge the first SR method to be able to perform *gradient refinement* of a pre-trained encoder-decoder model using our end-to-end NMSE loss, Eq. 1—to update the weights of the decoder at inference time. We note that there exists other non-gradient refinement approaches, that cannot update their decoder's weights. These consist of: (1) optimizing the constants in the generated equation form with a secondary optimization step (commonly using the BFGS algorithm) (Petersen et al., 2020; Biggio et al., 2021), and (2) using the MSE of the predicted equation(s) to guide a beam search sampler (d'Ascoli et al., 2022; Kamienny et al., 2022). As a result, to generalize to equations with a greater number of input variables pre-trained encoder-decoder methods require large pre-training datasets (e.g., millions of

datasets (Biggio et al., 2021)), and even larger generative models (e.g., $\sim 100$ million parameters (Kamienny et al., 2022)).

**Using priors.** The work of Jin et al. (2019) explicitly uses a simple pre-determined prior over the equation token set and updates this using an MCMC algorithm. They propose encoding this simple prior by hand, which has the drawbacks of, not being able to condition on the observations $\mathcal{D}$ (such as the equation classes and domains of $\mathbf{X} \in \mathcal{X}$ and $\mathbf{y} \in \mathcal{Y}$), not learnt automatically from datasets and is too restrictive to capture the conditional dependence of tokens as they are generated.

**Heuristic Based Methods.** Many symbolic regression algorithms use *search heuristics* designed for equations. Examples include genetic programming (GP) (Augusto & Barbosa, 2000; Schmidt & Lipson, 2009), simulated annealing (Stinstra et al., 2008) and AI Feynman (Udrescu & Tegmark, 2020). Genetic programming symbolic regression (Augusto & Barbosa, 2000; Schmidt & Lipson, 2009; Bäck et al., 2018) starts with a population of random equations, and evolves them through selecting the fittest for crossover and mutation to improve their fitness function. Although often useful, genetic programs suffer from scaling poorly to larger dimensions and are highly sensitive to hyperparameters. Alternatively, it is possible to use simulated annealing for symbolic regression, however again this has difficulty to scale to larger dimensions. AI Feynman (Udrescu & Tegmark, 2020) tackles the problem by applying a set of sequential heuristics (e.g., solve via dimensional analysis, translational symmetry, multiplicative separability, polynomial fit etc), and divides and transforms the dataset into simpler pieces that are then processed separately in a recursive manner. Their tool is a problem simplification tool for symbolic regression, and uses neural networks to identify these simplifying properties, such as translational symmetry and multiplicative separability. The respective sub-problems can be tackled by any symbolic regression algorithm, and Udrescu & Tegmark (2020) use a simple inner search algorithm of either a polynomial fit or a brute-force search. Furthermore, using brute-force search fails to scale to higher dimensions and is computationally inefficient, as it unable to leverage any structure making the search more tractable. Additionally, there exist other works to create an interpretable model from a black box model (Crabbe et al., 2020; Alaa & van der Schaar, 2019).

# G  BENCHMARK ALGORITHMS

In this section we detail benchmark algorithms, consisting of the following: (1) types of benchmark algorithms selected, (2) hyperparameters and implementation details and (3) a discussion on inclusion criteria for benchmark model selection.

**Benchmark Algorithm Selection**. The benchmark symbolic regression algorithms we selected to compare against are: Neural Guided Genetic Programming (NGGP) Mundhenk et al. (2021), as this is the current state-of-the-art for symbolic regression, superseding DSR (Petersen et al., 2020). Genetic programming (GP) (Fortin et al., 2012) which has long been an industry standard, and compare with Neural Symbolic Regression that Scales (NESYMRES), an pre-trained encoder-decoder method.

**Neural Guided Genetic Programming (NGGP).** (Mundhenk et al., 2021). We use their code and implementation provided, following their proposal to set the generator to be a single-layer LSTM RNN with 32 hidden nodes. We further follow their hyperparameter settings (Mundhenk et al., 2021), unless otherwise noted. This uses the same optimization method of NGPQT, as described further in Appendix C.

**Genetic programming (GP).** (Fortin et al., 2012). We use the software package "DEAP" (Fortin et al., 2012), following the same symbolic regression GP setup as Petersen et al. (2020), using their stated hyperparameters. This uses an initial population of equations generated using the "full" method (Koza, 1992) with a depth randomly selected between $d_{\min}$ and $d_{\max}$. Following Petersen et al. (2020) we do not include GP post-hoc constraints other than constraining the maximum length of the equations generated to a specified maximum length of 30 unless otherwise defined and setting the maximum number of possible constants to 3.

**Neural Symbolic Regression that Scales (NESYMRES).** (Biggio et al., 2021). We use their code and implementation provided, following their proposal, and using their pre-trained model used in their paper. To ensure fair comparison, we used the largest beam size that the authors proposed in the range of possible beam sizes, that of a beam size of 256. As NESYMRES was only pre-trained on a dataset with variables $d \leq 3$, we only evaluated it on problem sets where $d \leq 3$.

**Deep Generative Symbolic Regression (DGSR).** This work. We define the architecture in Appendix B. The *encoder* uses the set transformer from Lee et al. (2019). Using the notation from Lee et al. (2019) the encoder is formed by 3 induced set attention blocks (ISABs) and one final component of pooling by multi-head attention (PMA). This uses a hidden dimension of 32 units, one head, one output feature and 64 inducing points. The *decoder* uses a standard transformer decoder (where standard transformer models are from the core PyTorch library (Paszke et al., 2019)). The decoder consists of 2 layers, with a hidden dimension of 32, one attention head and zero dropout. The dimension of the input and output tokens is the size of the token library used in training plus two (i.e., for a padding token and a start and stop token). The inputs are encoded using an embedding of size 32 and have an additional positional encoding added to them, following standard practice (Vaswani et al., 2017). We also mask the target output tokens to prevent information leakage during the forward step, again following standard practice (Vaswani et al., 2017). The decoder generates each token of the equation $\bar{f}$ autoregressively, that is sampling from $p(\bar{f}_i|\bar{f}_{1:(1-i);\theta;\mathcal{D}})$ following the same sampling procedure from Petersen et al. (2020). We process the existing generated tokens $\bar{f}_{1:(1-i)}$ into their hierarchical tree state representation, detailed in Petersen et al. (2020), providing as inputs to the transformer decoder a representation of the parent and sibling nodes of the token being sampled (Petersen et al., 2020). The hierarchical tree state representation is encoded with a fixed size embedding of 16 units, with an additional positional encoding added to it (Vaswani et al., 2017), and fed into a standard transformer encoder, of 3 layers, with a hidden dimension of 32, one attention head and zero dropout. This generates an additional latent vector which is concatenated to the encoder (of the observations $\mathcal{D}$) latent vector, forming a total latent vector of $\mathbf{U} \in \mathbb{R}^{w+d_s}$, where $d_s = 32$, and $w = 32$. We also use the Broyden–Fletcher–Goldfarb–Shanno (BFGS) algorithm (Fletcher, 2013) for inferring numeric constants if any are present in the equations generated, following the setup of Petersen et al. (2020). In total our conditional generative model has a total of 122,446 parameters, which we note is approximately two orders of magnitude less than other pre-trained encoder-decoder methods (NESYMRES with 26 million parameters, and E2E with 86 million parameters).

**Notable exclusions.** The benchmark symbolic regression algorithms selected are all suitable for discovering equations of up to three variables, can accommodate numerical constants within the equations and have made their code accessible to benchmark against. There exist other symbolic regression algorithms that do not exhibit these features, therefore we do not include them to compare against:

- AI Feynman (AIF): Udrescu & Tegmark (2020) developed a problem simplification tool for symbolic regression, this divides and transforms the dataset into simpler pieces that are then processed separately in a recursive manner (also discussed in Appendix F). The respective sub-problems can be tackled by any symbolic regression algorithm, where Udrescu & Tegmark (2020) use a simple inner search algorithm of either a polynomial fit or a brute-force search. Although as others have noted (Petersen et al., 2020), more challenging sub-problems which have numerical constants or are non-separable, still require a more comprehensive underlying symbolic regression method to solve the sub-problem. Therefore, AIF could be used as a pre-processing problem-simplification step and combined with any symbolic regression method. However, we analyse in this work the underlying symbolic regression search problem, e.g., after any problem simplification steps have been applied. Furthermore, the simple inner search that AIF uses (polynomial fit or brute force search) is computationally expensive, and scales poorly to increasing variable size. It can rely on further information about the equation to be provided such as the units for each input variable, which is unrealistic in most settings where the units are often *unknown* and the output does not have a physical interpretation (e.g., A dataset with many non-physical variables).

- End-to-end symbolic regression with transformers (E2E): Kamienny et al. (2022) introduced an pre-trained encoder-decoder method that also provides an initial estimate of the constant floats, which are further refined in a standard secondary optimization step (using BFGS). Their transformer model contains a total of 86 million parameters, trained on a dataset of millions of equations. It was not possible to compare against E2E, as the authors had not released their code at the time of completing this work. Furthermore, it would be infeasible to re-implement and train such a large transformer model without a pre-trained model, therefore we exclude this as a symbolic regression benchmark in this work.

# H    STANDARD BENCHMARK PROBLEM RESULTS

Additionally, we evaluated DGSR against the popular standard benchmark problem sets of Nguyen (Uy et al., 2011), Nguyen with constants (Petersen et al., 2020), R rationals (Krawiec & Pawlak, 2013) and Livermore (Mundhenk et al., 2021).

**Nguyen problem set.** The average recovery rates on the Nguyen problem set (Uy et al., 2011) can be seen in Table 7. The Nguyen symbolic regression benchmark suite Uy et al. (2011), consists of 12 commonly used benchmark problem equations and has been extensively benchmarked against in symbolic regression prior works (White et al., 2013; Petersen et al., 2020; Mundhenk et al., 2021). We observe that DGSR achieves similar performance to NGGP. Note that NGGP optimized its hyperparameters for the Nguyen problem set, specifically for the Nguyen-7 problem (empirically observing NGGP having a high sensitivity to hyperparameters). Whereas DGSR did not have its many hyperparameters optimized (Appendix J), instead we only optimized one hyperparameter that of the learning rate to the same Nguyen-7 problem, empirically finding a learning rate of 0.0001 to be best, this was used across this benchmark problem set and the variation including constants, seen next (we performed the same hyperparameter tuning of the learning rate for NGGP, however found the default already optimized). We benchmarked against NGGP directly as it is currently state-of-the-art on the Nguyen problem set and defer the reader to other benchmark algorithms on the Nguyen problem set in Mundhenk et al. (2021); Petersen et al. (2020).

**Nguyen with constants problem set.** The average recovery rates on the Nguyen with constants problem set (Petersen et al., 2020) can be seen in Table 8. This benchmark problem set was introduced by Petersen et al. (2020), being a variation of a subset of the Nguyen problem set with constants to also optimize for. We observe that DGSR achieves the same average recovery rate to NGGP. We benchmarked against NGGP directly as it is currently state-of-the-art on the Nguyen with constants problem set, and defer the reader to other benchmark algorithms on the Nguyen with constants problem set in Petersen et al. (2020).

**R rationals problem set.** The average recovery rates on the R rationals problem set (Krawiec & Pawlak, 2013) can be seen in Table 9. We use the original problem set and the extended domain problem set, as defined in Mundhenk et al. (2021), indicated with *. We observe DGSR has a higher average recovery rate than state-of-the-art NGGP and even finding the original equation for the problem R3, which was not previously possible with the existing state-of-the-art (that of NGGP). Again, we benchmarked against NGGP directly as it is currently state-of-the-art on the R rationals problem set and defer the reader to other benchmark algorithms on the R rationals problem set in Mundhenk et al. (2021). Note that NGGP optimized its hyperparameters for the R rationals problem set, specifically for R-3*. Whereas DGSR did not have its many hyperparameters optimized (Appendix J), instead we only optimized one hyperparameter that of the learning rate to the same

Table 7: Average recovery rate ($A_{\mathrm{Rec}}\%$) on the Nguyen problem set with 95 % confidence intervals. Averaged over $\kappa = 10$ random seeds.

| Benchmark | Equation | DGSR | NGGP |
|---|---|---|---|
| Nguyen-1 | $x_1^3 + x_1^2 + x_1$ | 100 | 100 |
| Nguyen-2 | $x_1^4 + x_1^3 + x_1^2 + x_1$ | 100 | 100 |
| Nguyen-3 | $x_1^5 + x_1^4 + x_1^3 + x_1^2 + x_1$ | 100 | 100 |
| Nguyen-4 | $x_1^6 + x_1^5 + x_1^4 + x_1^3 + x_1^2 + x_1$ | 100 | 100 |
| Nguyen-5 | $\sin(x_1^2)\cos(x_1) - 1$ | 100 | 100 |
| Nguyen-6 | $\sin(x_1) + \sin(x_1 + x_1^2)$ | 100 | 100 |
| Nguyen-7 | $\log(x_1 + 1) + \log(x_1^2 + 1)$ | 60 | 100 |
| Nguyen-8 | $\sqrt{x_1}$ | 90 | 100 |
| Nguyen-9 | $\sin(x_1) + \sin(x_2^2)$ | 100 | 100 |
| Nguyen-10 | $2\sin(x_1)\cos(x_2)$ | 100 | 100 |
| Nguyen-11 | $x_1^{x_2}$ | 100 | 100 |
| Nguyen-12 | $x_1^4 - x_1^3 + \frac{1}{2}x_2^2 - x_2$ | 0 | 0 |
| Average recovery rate (%) $A_{\mathrm{Rec}}\%$ | | $87.50 \pm 4.07$ | $\mathbf{91.67 \pm 0.00}$ |

Table 8: Average recovery rate ($A_{\text{Rec}}\%$) on the Nguyen with constants problem set with 95% confidence intervals. Averaged over $\kappa = 10$ random seeds.

| Benchmark | Equation | DGSR | NGGP |
|---|---|---|---|
| Nguyen-1$^c$ | $3.39x_1^3 + 2.12x_1^2 + 1.78x_1$ | 100 | 100 |
| Nguyen-5$^c$ | $\sin(x_1^2)\cos(x_1) - 0.75$ | 100 | 100 |
| Nguyen-7$^c$ | $\log(x_1 + 1.4) + \log(x_1^2 + 1.3)$ | 100 | 100 |
| Nguyen-8$^c$ | $\sqrt{1.23x_1}$ | 100 | 100 |
| Nguyen-10$^c$ | $\sin(1.5x_1)\cos(0.5x_2)$ | 100 | 100 |
| Average recovery rate (%) $A_{\text{Rec}}\%$ | | **100 $\pm$ 0** | **100 $\pm$ 0** |

problem of R-3$^*$, empirically finding a learning rate of 0.0001 to be best, this was used across this benchmark problem set. For fair comparison we performed the same learning rate hyperparameter optimization for NGGP and found a learning rate of 0.0001 to be best, achieving higher recovery rate results than originally reported (Mundhenk et al., 2021).

Table 9: Average recovery rate ($A_{\text{Rec}}\%$) on the R rationals problem set with 95% confidence intervals. Averaged over $\kappa = 10$ random seeds.

| Benchmark | Equation | DGSR | NGGP |
|---|---|---|---|
| R-1 | $\frac{(x_1+1)^3}{x_1^2 - x_1 + 1}$ | 0 | 0 |
| R-2 | $\frac{x_1^5 - 3x_1^3 + 1}{x_1^2 + 1}$ | 0 | 0 |
| R-3 | $\frac{x_1^6 + x_1^5}{x_1^4 + x_1^3 + x_1^2 + x_1 + 1}$ | 20 | 0 |
| R-1$^*$ | $\frac{(x_1+1)^3}{x_1^2 - x_1 + 1}$ | 100 | 100 |
| R-2$^*$ | $\frac{x_1^5 - 3x_1^3 + 1}{x_1^2 + 1}$ | 90 | 60 |
| R-3$^*$ | $\frac{x_1^6 + x_1^5}{x_1^4 + x_1^3 + x_1^2 + x_1 + 1}$ | 100 | 100 |
| Average recovery rate (%) $A_{\text{Rec}}\%$ | | **51.66 $\pm$ 7.23** | 43.33 $\pm$ 5.06 |

**Livermore problem set.** The average recovery rates on the Livermore problem set (Mundhenk et al., 2021) can be seen in Table 10. DGSR achieves a similar performance to NGGP. Again, we benchmarked against NGGP directly as it is currently state-of-the-art on the Livermore problem set and defer the reader to other benchmark algorithms on the Livermore problem set in Mundhenk et al. (2021).

# I BENCHMARK PROBLEM DETAILS

**Standard Symbolic Regression Benchmark problems**. Details of the standard symbolic regression benchmark problem sets that we compared against in Appendix H are tabulated in Table 32 and Table 33. Specifically, most standard symbolic regression benchmarks use the following token library $\mathcal{L}_{\text{Koza}} = \{+, -, \div, \times, x_1, \exp, \log, \sin, \cos\}$ and have a defined variable domain $\mathcal{X}$ and sampling specification for each problem (e.g., Table 32). We follow the same setup as Petersen et al. (2020).

**Feynman problem sets**. We use equations from the Feynman Symbolic Regression Database (Udrescu & Tegmark, 2020), to provide more challenging equations of multiple variables. These are derived from the *Feynman Lectures on Physics* (Feynman et al., 1965), and also specify the domain $\mathcal{X}$ of the variables. We filtered these equations to those that had tokens that exist within the standard library set of tokens, that of $\mathcal{L}_{\text{Koza}} = \{+, -, \div, \times, x_1, \exp, \log, \sin, \cos\}$, excluding the variable tokens (e.g., $\{x_1, x_2, \dots\}$). We randomly selected a subset of these equations with two variables (labelled Feynman $d = 2$), and a further, more challenging subset sampled with five variables (labelled Feynman $d = 5$). Details of the Feynman benchmark problem sets are tabulated in Tables 34, 35. We note that the token library does not include the "const" token, however equations which

Table 10: Average recovery rate ($A_{\text{Rec}}\%$) on the R rationals problem set with 95 % confidence intervals. Averaged over $\kappa = 10$ random seeds.

| Benchmark | Equation | DGSR | NGGP |
|---|---|---|---|
| Livermore-1 | $1/3 + x_1 + \sin(x_1^2)$ | 60 | 100 |
| Livermore-2 | $\sin(x_1^2)\cos(x_1) - 2$ | 100 | 100 |
| Livermore-3 | $\sin(x_1^3)\cos(x_1^2) - 1$ | 100 | 100 |
| Livermore-4 | $\log(x_1 + 1) + \log(x_1^2 + 1) + \log(x_1)$ | 100 | 100 |
| Livermore-5 | $x_1^4 - x_1^3 + x_1^2 - x_2$ | 50 | 20 |
| Livermore-6 | $4x_1^4 + 3x_1^3 + 2x_1^2 + x_1$ | 90 | 90 |
| Livermore-7 | $\sinh(x_1)$ | 0 | 0 |
| Livermore-8 | $\cosh(x_1)$ | 0 | 0 |
| Livermore-9 | $x_1^9 + x_1^8 + x_1^7 + x_1^6 + x_1^5 + x_1^4 + x_1^3 + x_1^2 + x_1$ | 30 | 10 |
| Livermore-10 | $6\sin(x_1)\cos(x_2)$ | 40 | 0 |
| Livermore-11 | $\frac{x_1^2 x_1^2}{x_1 + x_2}$ | 100 | 100 |
| Livermore-12 | $x_1^5 / x_2^3$ | 100 | 100 |
| Livermore-13 | $x_1^{1/3}$ | 100 | 100 |
| Livermore-14 | $x_1^3 + x_1^2 + x_1 + \sin(x_1) + \sin(x_1^2)$ | 100 | 100 |
| Livermore-15 | $x_1^{1/5}$ | 100 | 100 |
| Livermore-16 | $x_1^{2/5}$ | 60 | 90 |
| Livermore-17 | $4\sin(x_1)\cos(x_2)$ | 30 | 60 |
| Livermore-18 | $\sin(x_1^2)\cos(x_2) - 5$ | 100 | 60 |
| Livermore-19 | $x_1^5 + x_1^4 + x_1^2 + x_1$ | 100 | 100 |
| Livermore-20 | $\exp(-x_1^2)$ | 100 | 100 |
| Livermore-21 | $x_1^8 + x_1^7 + x_1^6 + x_1^5 + x_1^4 + x_1^3 + x_1^2 + x_1$ | 100 | 100 |
| Livermore-22 | $\exp(-0.5x_1^2)$ | 10 | 90 |
| Average recovery rate (%) $A_{\text{Rec}}\%$ | | $71.36 \pm 9.82$ | $\mathbf{73.63 \pm 7.26}$ |

do have numeric constants can still be recovered with the provided token library, e.g., Feynman-6 can be recovered by $\frac{x_1(x_2 \times x_2)}{x_1(x_2 \times x_2) + x_1(x_2 \times x_2)}$.

**Synthetic $d = 12$ problem set**. We use the same equation generation framework discussed in Appendix J to synthetically generate a problem set of equations with $d = 12$ variables. We use a token library set of $\mathcal{L}_{\text{Synth}} = \{+, -, \div, \times, x_1, \ldots, x_{12}\}$. We note that the size of the library set $\mathcal{L}_{\text{Synth}}$ is 16, which is greater than the size of the standard library set $\mathcal{L}_{\text{Koza}}$ of 9, this creates an exponentially larger symbolic regression search space for the symbolic regression methods, making this benchmark problem set more challenging. Details of the Synthetic $d = 12$ problem set are tabulated in Table 36. When generating the equations for this problem set, we set the number of leaves of the equations to be generated to $l_{\max} = 25, l_{\min} = 10$.

**SRBench**. See Appendix S for details of the SRBench (La Cava et al., 2021) dataset.

## J  DATASET GENERATION AND TRAINING

To construct the pre-training set $\{\mathcal{D}^{(j)}\}_{j=1}^{m}$, we use the pioneering equation generation method of Lample & Charton (2019), which is further extended by Biggio et al. (2021) to generate equations with constants. This equation generation framework allows us to generate equations that correspond to an input library of tokens from a particular benchmark problem, $f^{(j)} \sim p(f), \forall j \in [1 : m]$, where we sample $m = 100\text{K}$ equations. For each equation $f^{(j)}$, we further obtain a dataset $\mathcal{D}^{(j)} = \{(y_i^{(j)}, \mathbf{X}_i^{(j)})\}_{i=1}^{n^{(j)}}$ by evaluating $f^{(j)}$ on $n^{(j)}$ random points in $\mathcal{X}$, i.e., $y_i^{(j)} = f^{(j)}(\mathbf{X}_i^{(j)})$. We define the variable domain $\mathcal{X}$ from the dataset specification from the problem set and use the most common specification for $\mathcal{X}$ from that problem set (e.g., for Feynman $d = 2$ problem set we use $U(1, 5, 20)$, as defined in Table 34). This allows us to pre-train a deep conditional generative model, $p_\theta(f|\mathcal{D})$ to encode a particular prior $p(f)$ for a specific library of tokens and $\mathcal{X}$. For each

problem set encountered that has a different library of tokens or $\mathcal{X}$ we generated a pre-training set and pre-trained a conditional generative model for it.

**How to specify** $p(f)$**.** The user can specify $p(f)$ by first selecting the specific library of tokens they wish to generate equations with, which includes the maximum number of possible variables that could appear in the equations, e.g., $d = 5$ includes the tokens of $\{x_1, \ldots, x_5\}$. The framework of Lample & Charton (2019) generates equation trees, where each randomly generated equation tree has a pre-specified number of maximum leaves $l_{\max}$ and a minimum number of leaves $l_{\min}$, we set to $l_{\max} = 5, l_{\min} = 3$ unless otherwise specified. Secondly, each non-leaf node is sampled following a user specified unnormalized weighted distribution of each operator, and we use the one shown in Table 11. Following Lample & Charton (2019); Biggio et al. (2021), each leaf node has a 0.8 probability of being an input variable and 0.2 probability of being an integer. We constrain equation trees that contain a variable of a higher dimension, to also contain the lower dimensional variables, e.g., if $x_5$ is present, then we require $x_1, \ldots, x_4$ to also be present in the equation tree. The tree is traversed in pre-order to produce an equation traversal in prefix notation, that is then converted to infix notation and parsed with Sympy (Meurer et al., 2017) to generate a functional equation, $f^{(j)}$. If we desire to generate equations with arbitrary numeric constants we can further modify the equation to include numeric constant placeholders, which can be filled by sampling values from a defined distribution (e.g., uniform distribution, $U(-1, 1)$) (Biggio et al., 2021). Furthermore, we store equations as functions, to allow the input support variable points to be re-sampled during pre-training. This partially pre-generated set allows for faster generation of pre-training data for the mini-batches (Biggio et al., 2021). We further drop any generated dataset $\mathcal{D}$ that contain Nans, which can arise from invalid operations (e.g., taking the logarithm of negative values).

Table 11: We use the following unnormalized weighted distribution when sampling non-leaf nodes in the equation generation framework of Lample & Charton (2019).

| Operator | $+$ | $\times$ | $-$ | $\div$ | pow2 | pow3 | pow4 | pow5 | log | exp | sin | cos |
|---|---|---|---|---|---|---|---|---|---|---|---|---|
| Unnormalized Prob | 10 | 10 | 5 | 5 | 4 | 2 | 1 | 1 | 4 | 4 | 4 | 4 |

**Pre-training.** Using the specifications of $p(f)$, we can generate an almost unbounded number of equations. We pre-compile 100K equations and train the conditional generator on these using a mini-batch of $t$ datasets, following Biggio et al. (2021). The overall pre-training algorithm is detailed in Algorithm 1, in Appendix D. Additionally we construct a validation set of 100 equations using the same pre-training setup, with a different random seed and check and remove any of the validation equations from the pre-training set. Furthermore, we check and remove any test problem set equations from the pre-training and validation equation sets. During pre-training we use the vanilla policy gradient (VPG) loss function to train the conditional generator parameters $\theta$. This is detailed in Appendices D, C, and we use the hyperparameters: batch size of $k = 500$ equations to sample, mini-batch of $t = 5$ datasets, EWMA coefficient [5] $\alpha = 0.5$, entropy weight $\lambda_{\mathcal{H}} = 0.003$, minimum equation length = 4, maximum equation length = 30, Adam optimizer (Kingma & Ba, 2014) with a learning rate of 0.001 and an early stopping patience of a 100 iterations (of a mini-batch). Empirically we observe that early stopping occurs approximately after a set of 10K datasets has been trained on. Furthermore, we use vanilla policy gradients (PG) during pre-training as this optimizes the average distribution of $p_\theta(f|\mathcal{D})$ and was empirically found to perform the best, amongst other types of PG methods possible (e.g., NGPQT or RSPG).

**Inference.** We detail the inference training routine in Algorithm 2. A training set and a test are sampled independently from the defined problem equation domain, to form a dataset $\mathcal{D}$. The training dataset is used to optimize the loss at inference time and the test set is only used for evaluation of the best equations found at the end of inference, which unless the true equation $f^*$ is found runs for 2 million equation evaluations on the training dataset. During inference we use the neural guided PQT (NGPQT) optimization method, outlined in Appendix C. The hyperparameters for inference time are: batch size of $k = 500$ equations to sample, entropy weight $\lambda_{\mathcal{H}} = 0.003$, minimum equation length = 4, maximum equation length = 30, PQT queue size = 10, sample selection size = 1, GP generations per iteration = 25, GP cross over probability = 0.5, GP mutation probability = 0.5, GP tournament size = 5, GP mutate tree maximum = 3 and Adam optimizer (Kingma & Ba, 2014) with

---

[5]where EWMA baseline is defined as $b_t = \alpha \, \mathbb{E}[R(f)] + (1 - \alpha)b_{t-1}$

a learning rate of 0.001. We also used $\epsilon = 0.02$ for the risk seeking quantile parameter. We note that we use the same GP component hyperparameters as in NGGP (Mundhenk et al., 2021).

**Hyperparameter selection.** Unless stated otherwise we used the same hyperparameters from Mundhenk et al. (2021). We did not carry out an extensive hyperparameter optimization routine, as done by Petersen et al. (2020); Mundhenk et al. (2021) (due to limited compute available), rather we tuned the learning rate only over a grid search of $\{0.1, 0.0025, 0.001, 0.0001\}$. Empirically the learning rate of 0.001 performs best in pre-training and inference for DGSR tuned on the Feynman-2 benchmark problem, and therefore is used throughout unless otherwise stated. To ensure fair comparison we also tuned the learning rate of NGGP over the same grid search, empirically observing 0.0025 performs the best and is used throughout, unless otherwise stated.

A further description of the evaluation metrics and associated error bars are detailed in Appendix K.

**Compute details.** This work was performed using a Intel Core i9-12900K CPU @ 3.20GHz, 64GB RAM with a Nvidia RTX3090 GPU 24GB. Pre-training the conditional generator took on average 5 hours.

## K  EVALUATION METRICS

In the following we discuss each evaluation metric in further detail.

**Recovery rate.** ($A_{\mathrm{Rec}}\%$)—the percentage of runs where the true equation $f^*$ was found, over a set number of $\kappa$ random seed runs (Petersen et al., 2020). This uses the strictest definition of symbolic equivalence, by a computer algebraic system (Meurer et al., 2017). Specifically, this checks for equivalence of both the functional form and any numeric constants if they are present. Additionally, we quote the average 95% confidence intervals for a problem set, by computing the 95% interval for each problem across the random seed runs and then averaging across the set of problems included in a benchmark problem set. We note that recovery rate is a stricter symbolic regression metric as it checks for exact equation equivalence. Other symbolic regression methods have proposed to use in distribution accuracy, where the predicted outputs of the equation are within a percentage of the true y values observed, and similarly for out of distribution accuracy (Biggio et al., 2021; Petersen et al., 2020). Naturally, if we find the correct true equation $f^*$ for a given problem—then additional other evaluation metrics that measure fit are satisfied to a perfect score, such as a: test MSE of 0.0, test extrapolation MSE of 0.0, coefficient of determination $R^2$-score of 1.0 and an accuracy to tolerance $\tau$ of 100% for both in distribution and out of distribution.

**Equation evaluations**. We also evaluate the average number of equation evaluations $\gamma$ until the true equation $f^*$ was found. We use this metric as a proxy for computational complexity across the benchmark algorithms, as testing many generated equations is a bottleneck in SR (Biggio et al., 2021; Kamienny et al., 2022). For example, analysing the standard symbolic regression benchmark problem Nguyen-7$^c$, DGSR finds the true equation in $\gamma = 20{,}187$ equation evaluations, taking a total of 1 minute and 36.9 seconds; whereas NGGP finds the true equation in $\gamma = 30{,}112$ equation evaluations, taking a total of 2 minutes and 20.5 seconds, both results averaged over $\kappa = 10$ random seeds.

**Pareto front with complexity.** We use the Pareto front and corresponding complexity definition as detailed in Petersen et al. (2020). Included here for completeness, the Pareto front is computed using the simple complexity measure of $C(f) = \sum_i^{|f|} c(f_i)$. Where $c$ is the complexity for a token, as defined as: 1 for $+, -, \times$, input variables and numeric constants; 2 for $\div$; 3 for $\sin$ and $\cos$; and 4 for $\exp$ and $\log$.

## L  FEYNMAN D=2 RESULTS

**Feynman** $d = 2$ **recovery rates**. Average recovery rates on the Feynman $d = 2$ problem set can be seen in Table 12. The corresponding inference equation evaluations on the Feynman $d = 2$ problem set can be seen in Table 13.

**Noise ablation.** We empirically observe DGSRs average recovery rate also decreases with increasing noise in the observations $\mathcal{D}$, which is to be expected compared to other symbolic regression methods (Petersen et al., 2020). We show a comparison of DGSR against NGGP when increasing the noise

Table 12: Average recovery rate ($A_{\text{Rec}}\%$) on the Feynman $d = 2$ problem set with 95 % confidence intervals. Averaged over $\kappa = 40$ random seeds.

| Benchmark | Equation | DGSR | NGGP | NESYMRES | GP |
|---|---|---|---|---|---|
| Feynman-1 | $x_1 x_2$ | 100 | 100 | 100 | 95 |
| Feynman-2 | $\frac{x_1}{2(1+x_2)}$ | 97.5 | 100 | 000 | 55 |
| Feynman-3 | $x_1 x_2^2$ | 100 | 100 | 100 | 100 |
| Feynman-4 | $1 + \frac{x_1 x_2}{(1-(x_1 x_2/3))}$ | 0 | 0 | 0 | 0 |
| Feynman-5 | $\frac{x_1}{x_2}$ | 100 | 100 | 100 | 95 |
| Feynman-6 | $\frac{1}{2} x_1 x_2^2$ | 100 | 100 | 100 | 0 |
| Feynman-7 | $\frac{3}{2} x_1 x_2$ | 100 | 100 | 000 | 5 |
| Average recovery rate (%) $A_{\text{Rec}}\%$ | | $\mathbf{85.36 \pm 0.69}$ | $\mathbf{85.71 \pm 0.00}$ | $57.14 \pm 0.00$ | $50.00 \pm 7.20$ |

level on the Feynman $d = 2$ problem set in Figure 6. Following the noise setup of Petersen et al. (2020), we add independent Gaussian noise to the dependent output $\mathbf{y}$, i.e., $\tilde{y}_i = y_i + \epsilon_i, \forall i \in [1 : n]$, where $\epsilon_i \sim \mathcal{N}(0, \alpha y_{\text{RMS}})$, with $y_{\text{RMS}} = \sqrt{\sum_{i=1}^{n} y_i^2}$. That is the standard deviation is proportional to the root-mean-square of $\mathbf{y}$. In Figure 6 we vary the proportionality constant $\alpha$ from 0 (noiseless) to 0.1 and evaluated DGSR and NGGP across all the problems in the Feynman $d = 2$ benchmark problem set. Figure 6 is plotted using a 10-fold larger training dataset, following Petersen et al. (2020). On average DGSR can perform equally as well as NGGP with noisy data, if not better.

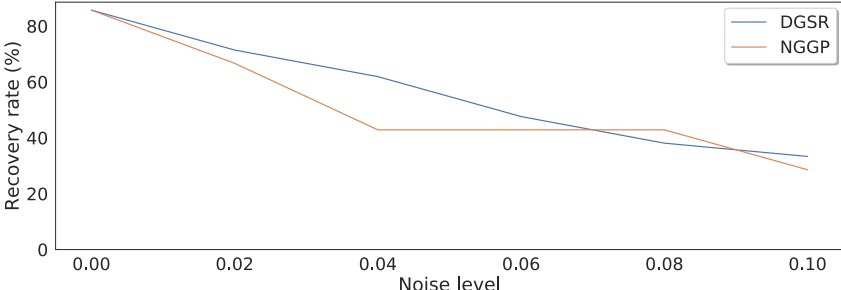

Figure 6: Noise ablation of average recovery rate on the Feynman $d = 2$ benchmark problem set against increasing noise level. Averaged over 3 random seeds.

**Data sub sample ablation with noise**. We empirically performed a data sub sample ablation with a small noise level of $\alpha = 0.001$, varying the inference training data samples from $n = 2$ to $n = 20$, with the results tabulated in Table 14 and plotted in Figure 7. Empirically this could suggest DGSR can leverage the encoded prior information in settings of noise and low data samples, which could be observed in real-world datasets.

Table 13: Inference equation evaluations on the Feynman $d = 2$ benchmark problem set with 95 % confidence intervals. DNF = Did not Find (i.e., recover the true equation given a budget of 2 million equation evaluations). Averaged over $\kappa = 40$ random seeds.

| Benchmark | Equation | DGSR | NGGP | NESYMRES | GP |
|---|---|---|---|---|---|
| Feynman-1 | $x_1 x_2$ | $10{,}002 \pm 18$ | $10{,}001 \pm 17$ | $256 \pm 0$ | $1{,}000 \pm 0$ |
| Feynman-2 | $\frac{x_1}{2(1+x_2)}$ | $197{,}062 \pm 108{,}505$ | $97{,}405 \pm 28{,}807$ | DNF | $10{,}467 \pm 2{,}162$ |
| Feynman-3 | $x_1 x_2^2$ | $10{,}010 \pm 16$ | $10{,}009 \pm 16$ | $256 \pm 0$ | $1{,}412 \pm 455$ |
| Feynman-4 | $1 + \frac{x_1 x_2}{(1-(x_1 x_2/3))}$ | DNF | DNF | DNF | DNF $\pm$ |
| Feynman-5 | $\frac{x_1}{x_2}$ | $10{,}008 \pm 20$ | $11{,}765 \pm 1{,}236$ | $256 \pm 0$ | $1{,}000 \pm 0$ |
| Feynman-6 | $\frac{1}{2} x_1 x_2^2$ | $118{,}335 \pm 37{,}221$ | $420{,}726 \pm 110{,}406$ | $256 \pm 0$ | DNF $\pm$ |
| Feynman-7 | $\frac{3}{2} x_1 x_2$ | $53{,}011 \pm 13{,}197$ | $126{,}884 \pm 31{,}620$ | DNF | $6{,}286 \pm 0$ |
| Average equation evaluations $\gamma$ | | $66{,}404$ | $112{,}798$ | $256$ | $4{,}033$ |

Table 14: Dataset sub sample ablation with noise of $\alpha = 0.001$. Average recovery rate ($A_{\text{Rec}}\%$) on the Feynman $d = 2$ problem set. Averaged over $\kappa = 10$ random seeds.

| Dataset samples ($n$) | DGSR | NGGP |
|---|---|---|
| 2 | 30.00 | 21.66 |
| 5 | 76.66 | 70.00 |
| 10 | 91.66 | 66.66 |
| 20 | 90.00 | 83.33 |

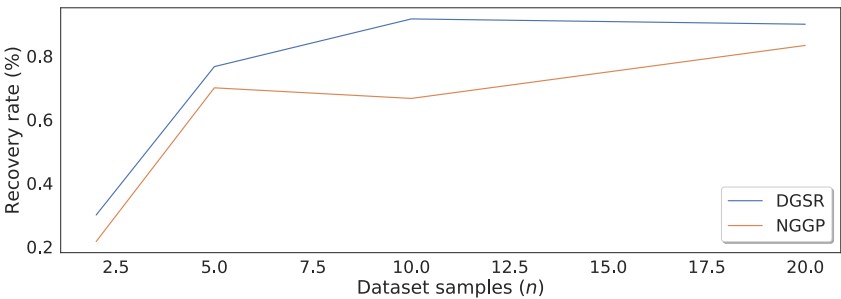

Figure 7: Dataset sub sample ablation with noise of $\alpha = 0.001$, with average recovery rate (%) on the Feynman $d = 2$ problem set against dataset of size $n$ samples. Averaged over 10 random seeds.

## M    FEYNMAN-7 EQUIVALENT EQUATIONS

**Exploiting equation equivalences**. Figure 3 (a) shows DGSR is able to correctly *capture* equation equivalences, and exploits these to generate many unique equivalent true equations. Although DGSR is able to generate many equivalent equations that are equivalent to the true equation $f^*$, we tabulate only the first 64 of these in Table 37. We note that all these equations are equivalent achieving zero test NMSE and can be simplified into $f^*$. We modified the standard experiment setting, to avoid early stopping once the true equation was found, and record only the true equations that have a unique form yet equivalent equation to $f^*$. Note that the true equation is $f^* = \frac{3}{2} x_1 x_2$ and using the defined benchmark token library set, the first shortest equivalent equation to $f^*$ is $x_1(x_2 + \frac{x_2 x_2}{x_2 + x_2})$.

## N    FEYNMAN D=5 PARETO FRONT EQUATIONS

**Finding accurate and simple equations**. Shown in Figure 8, for the most challenging equations to recover, DGSR can still find equations that are accurate and simple, i.e., having a low test NMSE and low complexity. The equations analyzed in the Pareto fronts in Figure 8, were chosen as none of the symbolic regression methods were able to find them. We note for a good symbolic regression method we wish to determine concise, simple (low complexity) and best fitting equations, otherwise it is undesirable to over-fit with an equation that has many terms having a high complexity, that fails to generalize well.

**Feynman-8 Pareto Front.** We tabulate five of the many equations along the Pareto front in Figure 8 (a) for the Feynman-8 problem, with DGSR equations in Table 15, NGGP equations in Table 16 and GP equations in Table 17. For completeness we duplicate some of Figure 4, here as Figure 8.

**Feynman-13 Pareto Front.** We tabulate five of the many equations along the Pareto front in Figure 8 (b) for the Feynman-13 problem, with DGSR equations in Table 18, NGGP equations in Table 19 and GP equations in Table 20.

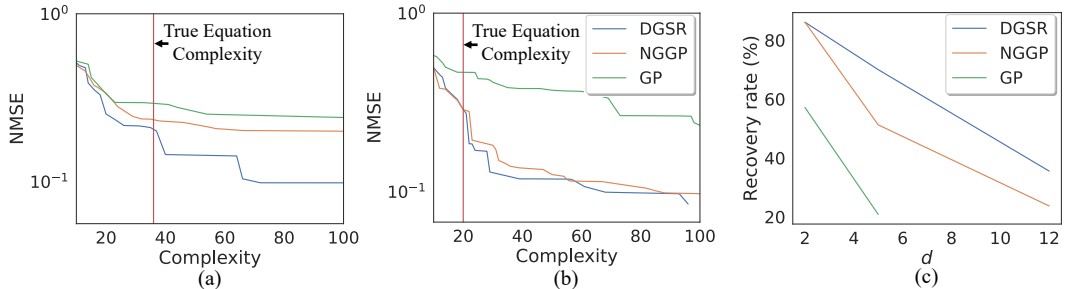

Figure 8: (a-b) Pareto front of test NMSE against equation complexity. Labelled: (a) Feynman-8, (b) Feynman-13. Ground truth equation complexity is the red line. Equations discovered are listed in Appendix N. (c) Average recovery rate of Feynman $d = 2$, Feynman $d = 5$ and synthetic $d = 12$ benchmark problems plotted against variable dimension $d$.

Table 15: DGSR equations from the Pareto plot in Figure 8 (a) for the Feynman-8 problem.

| Complexity | Test NMSE | Equation |
|---|---|---|
| 20 | 0.253 | $\frac{x_1 x_2}{x_2 + (2x_4 x_5 + x_5)/x_3}$ |
| 26 | 0.216 | $\frac{x_1 x_3}{x_3 + x_5 + (\sin(x_2) + \frac{x_4^2 x_5}{x_2})/x_2}$ |
| 33 | 0.212 | $\frac{x_1 x_3}{x_3 + x_4 + x_5(x_3/(x_4 + (x_2 + x_4)/x_2) + x_4 x_5/x_2)/x_2}$ |
| 37 | 0.200 | $\frac{x_1 x_2}{x_2 + x_5 + \log(x_2) + x_4^2 x_5(x_4 + x_5)/(x_3^2(x_2 + x_3))}$ |
| 64 | 0.142 | $\frac{x_1 x_3}{2x_3 + (-x_3 + x_5 \exp(x_5)/x_2)/(-x_3 + x_3(x_2 + x_5 + (3x_2 x_3/x_4 - x_5)/x_2)/x_4) + \exp(-x_2 + x_4)}$ |

Table 16: NGGP equations from the Pareto plot in Figure 8 (a) for the Feynman-8 problem.

| Complexity | Test NMSE | Equation |
|---|---|---|
| 20 | 0.337 | $\frac{x_1 x_3}{x_3 + x_4 + x_5(x_4 + x_5)/x_2^2}$ |
| 29 | 0.244 | $\frac{x_1 x_3}{x_3 + x_4 + (\sin(\log(x_3)) + x_4 x_5^2/x_2)/x_2}$ |
| 35 | 0.235 | $\frac{x_1 x_3}{x_3 + x_4 + (\log(x_1) + \sin(\log(x_5)) + x_4 x_5^2/x_2)/x_2}$ |
| 54 | 0.212 | $\frac{x_1 x_5}{x_5 + (x_3 + x_4 + (x_5 + (-x_2 + \sin(x_4))/x_2)/x_5) \exp(x_4(x_5 + (-x_2 + x_5/x_2)/x_2)/(x_3 + \sin(x_4)))/x_3}$ |
| 66 | 0.201 | $\frac{x_1}{x_4 + \cos(x_4/x_5) + x_5(x_4 + \cos(x_3) + x_5(\cos(x_4 - \exp(1/x_3)) + x_4^2 x_5/(x_2 x_3))/(x_2 x_3))/(x_2^2 x_3)}$ |

Table 17: GP equations from the Pareto plot in Figure 8 (a) for the Feynman-8 problem.

| Complexity | Test NMSE | Equation |
|---|---|---|
| 23 | 0.296 | $\frac{x_1}{(x_3/(x_4 x_5) + x_5)/x_3 + x_4 x_5/x_2}$ |
| 33 | 0.295 | $\frac{x_1}{(x_5 + (x_3 + x_5)/(x_5(x_5 + x_4 x_5/x_3)))/x_3 + x_4 x_5/x_2}$ |
| 41 | 0.288 | $\frac{x_1}{(x_5 + (x_3 + x_5)/(x_5 \log(\exp(x_4 + x_5^2/x_2))))/x_3 + x_4 x_5/x_2}$ |
| 44 | 0.277 | $\frac{x_1}{(x_5 + x_4 x_5/x_3)/\log(\exp(x_2)) + (x_3/(x_4(x_2 \log(x_4) + x_5)) + x_4)/x_3}$ |
| 54 | 0.252 | $\frac{x_1}{(x_3 + 1/x_4)/x_3 + x_5((x_3 x_4/(x_2 + \log(x_3)) + x_4)/x_3 + x_4^2/(x_1 x_3^2))/x_2}$ |

Table 18: DGSR equations from the Pareto plot in Figure 8 (b) for the Feynman-13 problem.

| Complexity | Test NMSE | Equation |
|---|---|---|
| 20 | 0.288 | $x_2 x_3(x_1 + x_2(x_3 + x_5)/x_5)/(x_4 x_5)$ |
| 22 | 0.185 | $x_2^2 x_3(x_1 + 2x_3 - x_4)/(x_4 x_5^2)$ |
| 24 | 0.170 | $x_2 x_3^2(x_1 x_2/x_4 + x_2)/(x_4 x_5^2)$ |
| 29 | 0.128 | $x_2^2 x_3^2(x_1 + x_2 - \log(x_5))/(x_4^2 x_5^2)$ |
| 45 | 0.118 | $x_2 x_3(x_2 x_3/x_5 - x_4 x_5 \sin(x_1)/(x_2 x_3))(x_1 + x_2 - \log(x_5))/(x_4^2 x_5)$ |

Table 19: NGGP equations from the Pareto plot in Figure 8 (b) for the Feynman-13 problem.

| Complexity | Test NMSE | Equation |
|---|---|---|
| 20 | 0.288 | $x_2x_3(x_1 + x_2x_3/x_5 + x_2)/(x_4x_5)$ |
| 22 | 0.281 | $(x_2x_3(x_1 + x_2x_3/x_5 + x_2) - x_5)/(x_4x_5)$ |
| 23 | 0.194 | $x_2^2x_3(x_1 + x_2x_3/x_4)/(x_4x_5^2)$ |
| 31 | 0.172 | $x_2^2x_3^2(x_1 + x_4)/(x_4^2x_5^2) + \cos(x_5)/x_5$ |
| 47 | 0.133 | $x_2x_3^2(x_1 + x_1x_4/x_3 + x_2^2)/(x_4^2x_5^2) + x_2x_3/x_5^3$ |

Table 20: GP equations from the Pareto plot in Figure 8 (b) for the Feynman-13 problem.

| Complexity | Test NMSE | Equation |
|---|---|---|
| 24 | 0.465 | $x_2(x_2 + x_3 + x_3(x_3 + \sin(x_4))\sin(x_5)/x_5 - x_4)$ |
| 25 | 0.429 | $(x_2 + \sin(x_4))(x_3 + x_3(x_3 + x_5)\sin(x_5)/x_5^2)$ |
| 28 | 0.427 | $(x_2 + \sin(x_4))(x_3 + x_3(x_3 + x_3/x_4)\sin(x_5)/x_5^2)$ |
| 35 | 0.382 | $(x_2 + \sin(x_4))(x_3 + x_3(x_3 + (x_2 + x_3)/(x_4 + \sin(x_2)))\sin(x_5)/x_5^2)$ |
| 50 | 0.370 | $(x_2 + \sin(x_4))(x_3 + x_3(x_2/(\exp(\cos(\exp(x_4))) + x_4/x_3) + x_3)\sin(x_5)/x_5^2)$ |

## O  FEYNMAN D=5 RESULTS

**Feynman $d = 5$ recovery rates**. Average recovery rates on the Feynman $d = 5$ problem set can be seen in Table 21. The corresponding inference equation evaluations on the Feynman $d = 5$ problem set can be seen in Table 22.

Table 21: Average recovery rate ($A_{\text{Rec}}\%$) on the Feynman $d = 5$ problem set with 95 % confidence intervals. Averaged over $\kappa = 40$ random seeds.

| Benchmark | Equation | DGSR | NGGP | GP |
|---|---|---|---|---|
| Feynman-8 | $\frac{x_1}{e^{\frac{x_4x_5}{x_2x_3}} + e^{\frac{-x_4x_5}{x_2x_3}}}$ | 0 | 0 | 0 |
| Feynman-9 | $x_1x_2x_3\log\frac{x_5}{x_4}$ | 55 | 97.5 | 0 |
| Feynman-10 | $\frac{x_1(x_3-x_2)x_4}{x_5}$ | 100 | 100 | 50 |
| Feynman-11 | $\frac{x_1x_2}{x_5(x_3^2-x_4^2)}$ | 100 | 37.5 | 10 |
| Feynman-12 | $\frac{x_1x_2^2x_3}{3x_4x_5}$ | 97.5 | 52.5 | 0 |
| Feynman-13 | $x_1(e^{\frac{x_2x_3}{x_4x_5}} - 1)$ | 5 | 22.5 | 0 |
| Feynman-14 | $x_5x_1x_2(\frac{1}{x_4} - \frac{1}{x_3})$ | 100 | 97.5 | 10 |
| Feynman-15 | $x_1(x_2 + x_3x_4\sin x_5)$ | 100 | 100 | 50 |
| Average recovery rate (%) $A_{\text{Rec}}\%$ | | **69.69 ± 3.38** | 63.44 ± 6.64 | 15.00 ± 12.39 |

## P  ADDITIONAL FEYNMAN RESULTS

The average recovery rates on an additional AI Feynman problem set (Udrescu & Tegmark, 2020) of 32 equations can be seen in Table 23. DGSR achieves a similar performance to NGGP. To get this problem set, we filtered all the Feynman equations that have equation tokens that are inside the Koza token library (detailed in Appendix I). DGSR can run on any token sets, however, requires changing the supported token set and pre-training a new conditional generative model when changing the token set to be used (if not done so already).

## Q  SYNTHETIC D=12 RESULTS

The recovery rates for the synthetic $d = 12$ problem set are tabulated in Table 24. The specification and generation details are in Appendix I. Empirically we investigated attempting to recover the

Table 22: Inference equation evaluations on the Feynman $d = 5$ benchmark problem set with 95% confidence intervals. DNF = Did not Find (i.e., recover the true equation given a budget of 2 million equation evaluations).

| Benchmark | Equation | DGSR | NGGP | GP |
|-----------|----------|------|------|-----|
| Feynman-8 | $\frac{x_1}{e^{\frac{x_4 x_5}{x_2 x_3}} + e^{\frac{-x_4 x_5}{x_2 x_3}}}$ | DNF | DNF | DNF |
| Feynman-9 | $x_1 x_2 x_3 \log \frac{x_5}{x_4}$ | $1,209,538 \pm 257,408$ | $503,158 \pm 150,969$ | DNF |
| Feynman-10 | $\frac{x_1(x_3 - x_2)x_4}{x_5}$ | $75,110 \pm 15,645$ | $98,331 \pm 27,968$ | DNF |
| Feynman-11 | $\frac{x_1 x_2}{x_5(x_3^2 - x_4^2)}$ | $979,457 \pm 131,399$ | $1,707,890 \pm 174,220$ | $27,780 \pm 0$ |
| Feynman-12 | $\frac{x_1 x_2^2 x_3}{3 x_4 x_5}$ | $558,393 \pm 135,956$ | $1,517,072 \pm 192,056$ | $45,609 \pm 17,663$ |
| Feynman-13 | $x_1(e^{\frac{x_2 x_3}{x_4 x_5}} - 1)$ | $1,949,722 \pm 94,348$ | $1,821,054 \pm 129,404$ | DNF |
| Feynman-14 | $x_5 x_1 x_2(\frac{1}{x_4} - \frac{1}{x_3})$ | $299,833 \pm 71,471$ | $662,238 \pm 139,361$ | $19,751 \pm 0$ |
| Feynman-15 | $x_1(x_2 + x_3 x_4 \sin x_5)$ | $48,047 \pm 11,838$ | $78,416 \pm 23,951$ | $30,386 \pm 12,078$ |
| Average equation evaluations $\gamma$ | | 731,442 | 912,594 | 198,455 |

equations with linear regression with polynomial features (testing both full, and sparse i.e., lasso-linear regression), however found this was not possible, and predictions from a linear model had a significantly higher test NMSE compared to DGSR and NGGP methods. We observe that DGSR can find the true equation when searching through a challenging very large equation space. Note that to generate equations of a suitable length we modified the maximum length of tokens that can be generated to 256 for all methods, i.e., DGSR, NGGP and GP.

## R  USING DIFFERENT OPTIMIZER RESULTS

**DGSR can be used with different PG optimizers.** DGSR can be used with other policy gradient optimization methods. We ablated the optimizer by switching off the GP component for both DGSR and NGGP, which becomes similar to the optimizer in Petersen et al. (2020) using PQT. Empirically we observe without the GP component the average recovery rate decreases, as others have shown for this optimizer (Mundhenk et al., 2021). However, we still observe that DGSR has a higher average recovery than that of NGGP, when both do not use the genetic programming component, whilst having a significantly lower number of equation evaluations, in Table 25 and Table 26.

## S  SRBENCH RESULTS

On SRBench (La Cava et al., 2021), we have evaluated DGSR against the ODE-Strogatz (Strogatz, 2018) problem set (14 unique equations), and 119 of the Feynman (Udrescu & Tegmark, 2020) unique equations, and 95 of the provided black-box regression datasets from PMLB (Romano et al., 2022). While we would love to perform a more comprehensive evaluation on all unique equations in SRBench with additional noise at this time, unfortunately we do not have the same resources of multiple 10's of GPUs. However, we believe that these combined results are strong enough to support all the major claims in this paper.

**SRBench ground truth unique equations.** We observe in the zero noise case on SRBench ground truth unique equations (ODE-Strogatz and Feynman), that DGSR achieves the highest symbolic recovery (solution) rate against the baselines provided of 63.25%, which is significant compared to the second best SRBench baseline of AI Feynman at 52.65%, shown in Figure 9, 10. Furthermore, analyzing the metric of equation accuracy, defined by $R^2_{\text{test}} > 0.99$ (i.e., the $R^2$ metric on the test samples is greater than 0.99), DGSR remains competitive placing 4th, with a mean equation accuracy rate of 90.94%. Whereas the other three best methods are genetic programming methods, MRGP, Operon and SBP-GP which have a mean accuracy rate of 96.13%, 93.92% and 93.65% respectively. Furthermore, DGSR on the same unique equations, as shown in the Figure 11, has the lowest simplified equation complexity for the highest comparative accuracy (complexity — DGSR: 15.42, MRGP: 157.62, Operon: 41.18, SBP-GP: 128.55) and lowest inference time in seconds for the highest comparative accuracy (inference time — DGSR: 706.94s, MRGP: 13,665.00s, Operon: 1,874.91s, SBP-GP: 27,713.85s). We highlight that it is possible to fit a more accurate equation which

Table 23: Average recovery rate ($A_{\mathrm{Rec}}\%$) on the additional Feynman problem set with 95 % confidence intervals. Averaged over $\kappa = 10$ random seeds. Here $d$ is the variable dimension.

| Benchmark | $d$ | Equation | DGSR | NGGP |
|---|---|---|---|---|
| Feynman-A-1 | 9 | $\frac{x_3 x_1 x_2}{(x_5-x_4)^2+(x_7-x_6)^2+(x_9-x_8)^2}$ | 0 | 0 |
| Feynman-A-2 | 8 | $\frac{x_1 x_2}{x_3 x_4} + \frac{x_1 x_5}{x_6 x_2^2 x_3 x_4}x_8$ | 100 | 30 |
| Feynman-A-3 | 6 | $x_1 e^{\frac{-x_2 x_5 x_3}{x_6 x_4}}$ | 70 | 90 |
| Feynman-A-4 | 6 | $x_1 x_4 + x_2 x_5 + x_3 x_6$ | 100 | 100 |
| Feynman-A-5 | 6 | $x_1(1 + \frac{x_5 x_6 \cos(x_4)}{x2*x3})$ | 100 | 100 |
| Feynman-A-6 | 6 | $x_1(1 + x_3)x_2$ | 100 | 100 |
| Feynman-A-7 | 4 | $\frac{x_1 x_4 x_2}{x_3}$ | 100 | 100 |
| Feynman-A-8 | 4 | $\frac{x_1 x_2 x_3}{x_4}$ | 100 | 100 |
| Feynman-A-9 | 4 | $\frac{1}{x_1-1}x_2\frac{x_4}{x_3}$ | 20 | 100 |
| Feynman-A-10 | 4 | $\frac{x_1 x_2 x_3}{2x_4}$ | 90 | 100 |
| Feynman-A-11 | 4 | $\frac{x_1 x_2 x_4}{x_3}$ | 100 | 100 |
| Feynman-A-12 | 4 | $x_1\left(\cos(x_2 x_3) + x_4 \cos(x_2 x_3)^2\right)$ | 0 | 0 |
| Feynman-A-13 | 4 | $-x_1 x_2\frac{x_3}{x4}$ | 100 | 100 |
| Feynman-A-14 | 4 | $\frac{x_1 x_3 + x_2 x_4}{x_1 + x_2}$ | 100 | 60 |
| Feynman-A-15 | 4 | $\frac{1}{2}x_1(x_2^2 + x_3^2 + x_4^2)$ | 0 | 0 |
| Feynman-A-16 | 3 | $-x_1 x_2 \cos(x_3)$ | 100 | 100 |
| Feynman-A-17 | 3 | $\frac{x3+x2}{1+\frac{x_3 x_2}{x1^2}}$ | 30 | 0 |
| Feynman-A-18 | 3 | $x_1 x_2 x_3$ | 100 | 100 |
| Feynman-A-19 | 3 | $x_1 x_2 x_3^2$ | 100 | 100 |
| Feynman-A-20 | 3 | $x_1 x_2\frac{x_3}{2}$ | 100 | 100 |
| Feynman-A-21 | 3 | $\frac{1}{x_1-1}x_2 x_3$ | 70 | 90 |
| Feynman-A-22 | 3 | $\frac{x_3}{1-\frac{x_2}{x_1}}$ | 100 | 100 |
| Feynman-A-23 | 3 | $x_1 x_3 x_2$ | 100 | 100 |
| Feynman-A-24 | 3 | $\frac{x_1 \sin\left(x_3\frac{x_2}{2}\right)^2}{\sin(x_2/2)^2}$ | 0 | 0 |
| Feynman-A-25 | 3 | $x_1(1 + x_2 \cos(x_3))$ | 100 | 100 |
| Feynman-A-26 | 3 | $\frac{1}{\frac{1}{x_1}+\frac{x_3}{x_2}}$ | 100 | 100 |
| Feynman-A-27 | 3 | $2x_1\left(1 - \cos(x_2 x_3)\right)$ | 100 | 90 |
| Feynman-A-28 | 3 | $\frac{x_1}{x_2(1+x_3)}$ | 100 | 100 |
| Feynman-A-29 | 7 | $\left(\frac{x_1 x_2 x_3 x_4 x_5}{4x_6 \sin(x_7/2)^2}\right)^2$ | 0 | 0 |
| Feynman-A-30 | 4 | $\frac{x_1}{1+x_1/(x_2 x_3^2)(1-\cos(x_4))}$ | 0 | 0 |
| Feynman-A-31 | 4 | $\frac{x_1(1-x_2^2)}{1+x_2 \cos(x_3-x_4)}$ | 0 | 0 |
| Feynman-A-32 | 4 | $x_1\frac{\sin(x_2/2)\sin(x_4 x_3/2)^2}{(x_2/2 \sin(x_3/2))}$ | 0 | 0 |
| Average recovery rate (%) $A_{\mathrm{Rec}}\%$ | | | **67.81 ± 4.60** | **67.81 ± 3.00** |
| Average equation evaluations $\gamma$ | | | 318,042 | 328,499 |

has a greater complexity (i.e., more terms), however for a good symbolic regression method we seek the simplest equation that explains the dataset accurately (therefore being a trade-off in accuracy and complexity).

**SRBench blackbox datasets.** On the SRBench black box datasets, in the zero noise case, we observe DGSR is competitive to the state-of-the-art, producing equations for blackbox-datasets that have a low median test RMSE, ranking fourth out of the SRBench implemented methods (median test RMSE — DGSR: 0.44, Operon: 0.36, SBP-GP: 0.42, FEAT: 0.43), shown in Figure 12. These equations also

Table 24: Average recovery rate ($A_{\text{Rec}}\%$) on the synthetic $d = 12$ problem set with 95 % confidence intervals. Averaged over $\kappa = 20$ random seeds. Here - indicates that the method was not able to find any true equations, therefore average number of equation evaluations until the true equation $f^*$ is discovered cannot be estimated.

| Benchmark | Equation | DGSR | NGGP | GP |
|---|---|---|---|---|
| Synthetic-1 | $x_{12} + x_9(x_{10} + x_{11}) + x_1 + x_2 + x_3 + x_4 + x_5 + x_6 + x_7 x_8$ | 20 | 0 | 0 |
| Synthetic-2 | $x_{10} + x_{11} + x_{12} + x_3(x_1 + x_2) + x_4 x_5 + x_6 + x_7 + x_8 + x_9$ | 100 | 100 | 0 |
| Synthetic-3 | $x_{10} + x_9(x_1 + x_2 + x_3 + x_4 + x_5 + x_6 + x_7 + x_8) + x_{11} + x_{12}$ | 0 | 0 | 0 |
| Synthetic-4 | $x_8(x_6 + x_7) - (x_{10} + x_{11}x_{12} + x_9)x_1 + x_2 + x_3 + x_4 + x_5$ | 0 | 0 | 0 |
| Synthetic-5 | $x_{10} + x_{11} + x_{12} + x_9(x_1 + x_2) - x_3 + x_4 + x_5 + x_6 + x_7 + x_8$ | 45 | 0 | 0 |
| Synthetic-6 | $x_1(x_{10} - x_{11}) - x_{12} + x_2 + x_3 + x_4 + x_5 + x_6 + x_7 + x_8 + x_9$ | 100 | 100 | 0 |
| Synthetic-7 | $x_1 x_2 - x_{11}(-x_{10} + x_6 + x_7) + x_8 - x_9 + x_{12} + x_3 + x_4 + x_5$ | 0 | 0 | 0 |
| Average recovery rate (%) $A_{\text{Rec}}\%$ | | **37.86 ± 5.62** | 28.57 ± 0.00 | 0 ± 0 |
| Average equation evaluations $\gamma$ | | 271,302 | 828,905 | - |

Table 25: Ablated optimizer where both DGSR and NGGP have their genetic programming component switched off, being similar to the one in Petersen et al. (2020). Top: Inference equation evaluations on the Feynman $d = 2$ benchmark problem set with standard deviations. DNF = Did not Find (i.e., recover the true equation given a budget of 2 million equation evaluations). Bottom: Average recovery rate ($A_{\text{Rec}}\%$) on the Feynman $d = 2$ benchmark problem set, with 95% confidence intervals, with rates in Table 26. Both averaged over $\kappa = 15$ random seeds.

| Benchmark | Equation | DGSR | NGGP |
|---|---|---|---|
| Feynman-1 | $x_1 x_2$ | $4,062 \pm 4,786$ | $22,688 \pm 25,943$ |
| Feynman-2 | $\frac{x_1}{2(1+x_2)}$ | $26,733 \pm 21,792$ | $356,867 \pm 125,319$ |
| Feynman-3 | $x_1 x_2^2$ | $5,281 \pm 4,426$ | $42,719 \pm 23,950$ |
| Feynman-4 | $1 + \frac{x_1 x_2}{(1-(x_1 x_2/3))}$ | DNF | DNF |
| Feynman-5 | $\frac{x_1}{x_2}$ | $1,938 \pm 1,802$ | $28,906 \pm 15,513$ |
| Feynman-6 | $\frac{1}{2}x_1 x_2^2$ | $30,773 \pm 27,702$ | $210,538 \pm 99,439$ |
| Feynman-7 | $\frac{3}{2}x_1 x_2$ | $107,350 \pm 104,483$ | $245,667 \pm 33,006$ |
| Average equation evaluations $\gamma$ | | **29,356** | 151,231 |
| Average recovery rate (%) $A_{\text{Rec}}\%$ | | **74.28 ± 8.75** | 70.47 ± 7.58 |

Table 26: Ablated optimizer where both DGSR and NGGP have their genetic programming component switched off, being similar to the one in Petersen et al. (2020). Average recovery rate ($A_{\text{Rec}}\%$) on the Feynman $d = 2$ problem set with 95% confidence intervals. Averaged over $\kappa = 15$ random seeds.

| Benchmark | Equation | DGSR | NGGP |
|---|---|---|---|
| Feynman-1 | $x_1 x_2$ | 100.00 | 100.00 |
| Feynman-2 | $\frac{x_1}{2(1+x_2)}$ | 93.33 | 93.33 |
| Feynman-3 | $x_1 x_2^2$ | 100.00 | 100.00 |
| Feynman-4 | $1 + \frac{x_1 x_2}{(1-(x_1 x_2/3))}$ | 0.00 | 0.00 |
| Feynman-5 | $\frac{x_1}{x_2}$ | 100.00 | 100.00 |
| Feynman-6 | $\frac{1}{2}x_1 x_2^2$ | 66.66 | 80.00 |
| Feynman-7 | $\frac{3}{2}x_1 x_2$ | 60.00 | 20.00 |
| Average recovery rate (%) $A_{\text{Rec}}\%$ | | **74.28 ± 8.75** | 70.47 ± 7.58 |

have a competitive test $R^2$ metric, with DGSR having a test median $R^2$ of 0.84 (ranking 6th out of the benchmark methods).

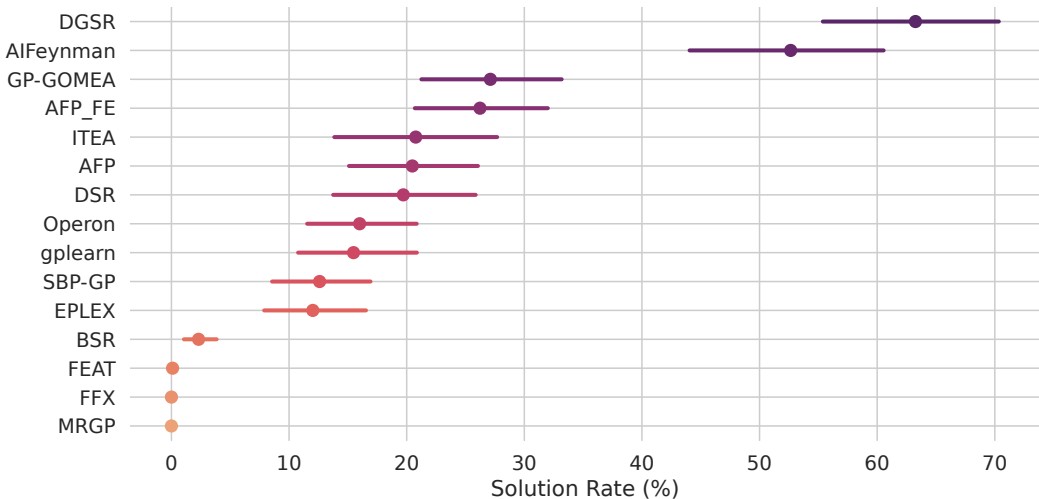

Figure 9: Symbolic recovery rate, labelled here as Symbolic solution rate (%) on the SRBench ground truth unique equations, and the SRBench provided methods. Points indicate the mean the test set performance on all ground truth problems, and bars show the 95% confidence interval.

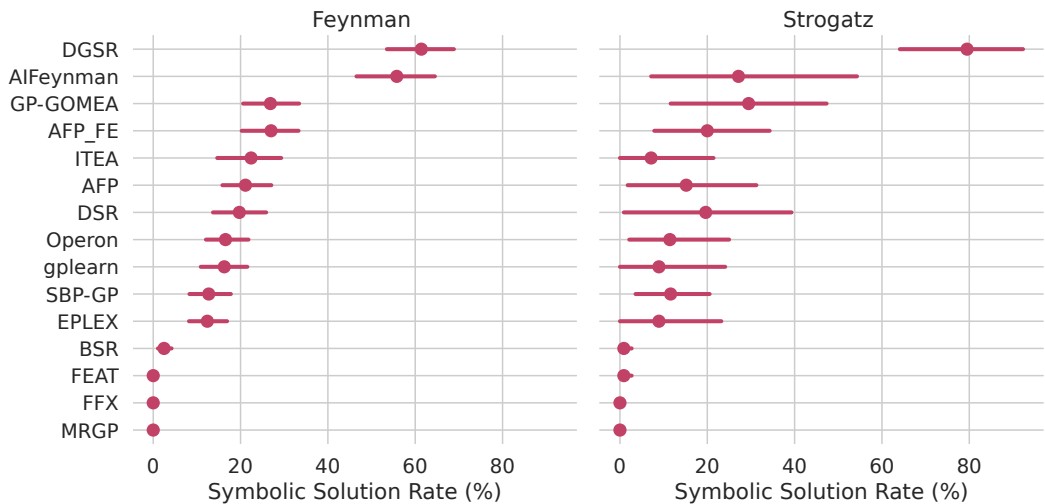

Figure 10: Symbolic recovery rate, split for each problem set type, labelled here as Symbolic solution rate (%) on the SRBench ground truth unique equations, and the SRBench provided methods. Points indicate the mean the test set performance on all ground truth problems, and bars show the 95% confidence interval.

## T  LOCAL OPTIMA

We discuss two sources of understood local optima in the symbolic regression literature, that of (1) the skeleton equation local optima (Mundhenk et al., 2021) and (2) the numerical constants local optima (Kamienny et al., 2022).

**(1).** DGSR specifically is assisted to avoid getting stuck in skeleton equation local optima, as it is optimized at inference with a combined policy gradient-based and genetic programming training optimization algorithm, that of neural guided priority queue training (NGPQT) of Mundhenk et al. (2021), detailed in Appendix C. Mundhenk et al. (2021) hypothesizes the improved performance over gradient-based training methods is due to the genetic programming component providing "fresh" new samples that help the optimization method escape local optima. We also observe the increase in

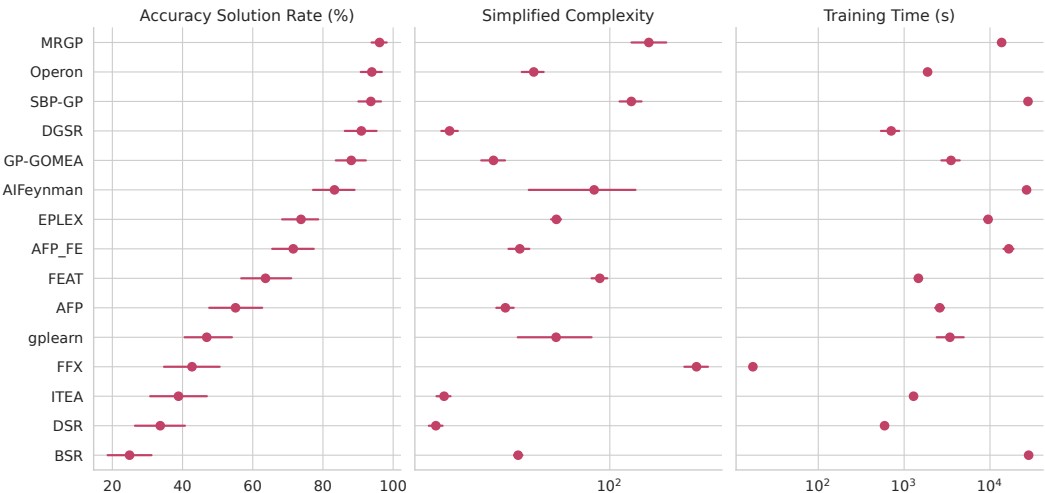

Figure 11: Equation accuracy, plotted here as the accuracy solution rate ($R^2_{\text{test}} > 0.99$), against equation complexity and inference training time on the SRBench ground truth unique equations, and the SRBench provided methods. Points indicate the mean the test set performance on all ground truth problems, and bars show the $95\%$ confidence interval.

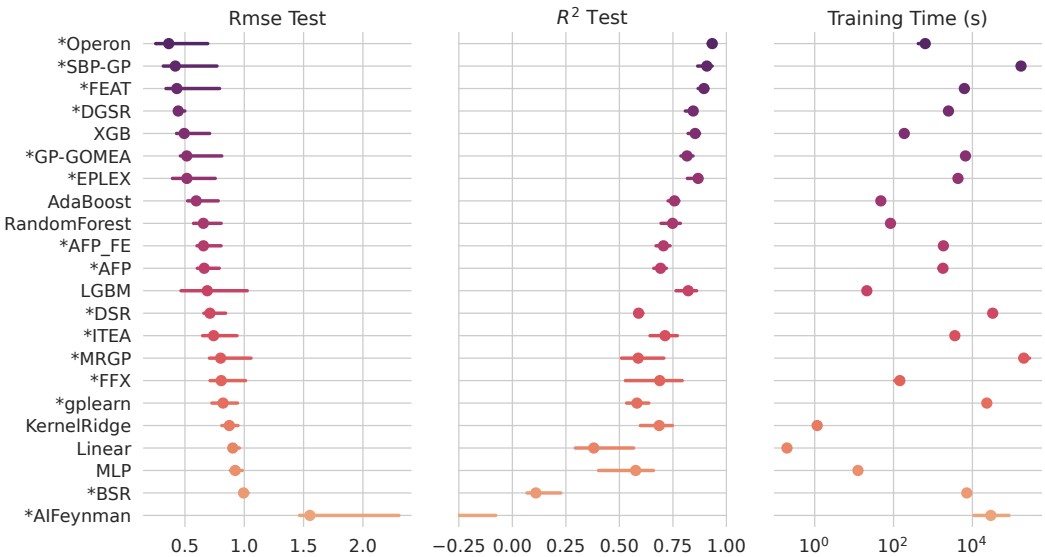

Figure 12: RMSE test error against $R^2_{\text{test}}$ and training time on the SRBench blackbox datasets, and the SRBench provided methods. Points indicate the mean of the median test set performance on all problems, and bars show the $95\%$ confidence interval. Methods marked with an asterisk are symbolic regression methods.

symbolic recovery rate using NGPQT optimization compared to pure policy gradient methods, as detailed in the optimization ablation study in Appendix E and results in Appendix R, i.e., (Feynman $d = 2$ problem set, average recovery rate — DGSR: $85.71\%$ compared to ablated DGSR with no genetic programming component : $74.28\%$).

**(2).** Like many existing works, the current DGSR suffers from the local optima of the numerical constants. DGSR uses the same setup of the numerical optimizer as Petersen et al. (2020), i.e., it first guesses an initial guess of $1.0$ for each constant, and then further refines them with the optimizer of Broyden–Fletcher–Goldfarb–Shanno algorithm (BFGS). However, the recent seminal work of Kamienny et al. (2022) propose a solution to mitigate this issue, and their approach could be

incorporated in future versions of DGSR. Specifically, we envisage future work using the generator to predict an initial guess of the numerical constants (rather than initializing them as constants), however note this is out of scope for this current work, therefore leave this as a future work we plan to implement and build on top of.

## U  ENCODER DECODER ARCHITECTURE ABLATION

DGSR can use other encoder-decoder architectures, specifically it is desirable to satisfy the invariant and equivariant properties outlined in section 3.1. We perform an ablation, training DGSR with a set transformer encoder and a LSTM RNN decoder, instead of the proposed transformer decoder. The results are tabulated in Table 27. We compared this ablated version of DGSR on the Feynman $d = 2$ problem set.

Table 27: DGSR decoder architecture ablation, we change the decoder architecture from a transformer in DGSR to that of a LSTM RNN. Average recovery rate ($A_{\text{Rec}}\%$) on the Feynman $d = 2$ problem set with 95% confidence intervals with average inference equation evaluations. Averaged over $\kappa = 20$ random seeds.

|  | Transformer decoder (DGSR) | LSTM RNN decoder |
|---|---|---|
| Average recovery rate (%) $A_{\text{Rec}}\%$ | **85.71 $\pm$ 0.00** | **85.71 $\pm$ 0.00** |
| Average equation evaluations $\gamma$ | **23,969** | 106,130 |

## V  LIMITATIONS AND OPEN CHALLENGES

In the following we discuss the limitations with open challenges.

**Complex equations**. DGSR may fail to discover highly complex equations. The difficulty in discovering the true equation $f^*$ could arise from three factors: (1) a large number of variables are required, (2) the equation $f^*$ involves many operators making the equation length long, or (3) the equation $f^*$ exhibits a highly nested structure. We note that these settings are inherently difficult, even for human experts, however, pose an exciting open challenge for future works.

**Unobserved variables**. DGSR assumes all variables in the true equation $f^*$ are observed. Therefore, it would fail to discover a true equation $f^*$ with unobserved variables present, however this setting is challenging and or even *impossible* without the use of additional assumptions (Reinbold et al., 2021; Lu et al., 2021). Furthermore, DGSR could still find a concise approximate equation using the observed variables.

**Local optima**. We detail both sources of local optima in the symbolic regression literature in detail in Appendix T, with respect to DGSR and future work associated to solve these. At a high level DGSR is able to overcome the local optima of the functional equation local optima by leveraging the genetic programming component, as other methods have shown (Mundhenk et al., 2021). Moreover, DGSR will suffer from the same constant token optimization optima as other methods, although there exists future work to address these, detailed in Appendix T.

## W  ENCODER ABLATION

We performed an additional encoder ablation, by replacing the set transformer in the encoder-decoder architecture with a plain transformer. We emphasize that we do not recommend using a plain transformer as the encoder for symbolic regression as it is not *permutation invariant* across the input samples $n$ in the dataset $\mathcal{D}$ (Section 3.1). The ablation results can be seen in the Table 28, with evaluation metrics evaluated across all the problems in the Feynman d=5 problem set.

## X  CROSS ENTROPY WITH NO REFINEMENT ABLATION

We also performed a further additional ablation of pre-training our encoder-decoder architecture with a cross entropy loss without refinement at inference time—only sampling from the decoder—thereby

Table 28: DGSR encoder architecture ablation, we change the encoder architecture from a set transformer in DGSR to that of a plain transformer for the encoder. Average recovery rate ($A_{\text{Rec}}\%$) on the Feynman $d = 5$ problem with average inference equation evaluations.

| | Set Transformer encoder (DGSR) | Plain Transformer encoder |
|---|---|---|
| Average recovery rate (%) $A_{\text{Rec}}\%$ | **67.50** | 67.18 |
| Average equation evaluations $\gamma$ | **23,969** | 348,666 |

not updating the decoders weights at inference time. This can be seen in Table 29, with evaluation metrics evaluated across all the problems in the Feynman d=5 problem set.

Table 29: DGSR training ablation, we pre-train using only a cross entropy loss and then do not perform refinement at inference time—thereby only sampling from the decoder in this ablation. Average recovery rate ($A_{\text{Rec}}\%$) on the Feynman $d = 5$ problem with average inference equation evaluations.

| | DGSR | Pre-train with CE and no refinement at inference |
|---|---|---|
| Average recovery rate (%) $A_{\text{Rec}}\%$ | **67.50** | 47.5 |
| Average equation evaluations $\gamma$ | **23,969** | 486,140 |

## Y  ADDITIONAL SYNTHETIC EXPERIMENTS

To provide further empirical results of DGSR, we synthetically generated $\omega$ equations using the concise and seminal equation generation framework of Lample & Charton (2019)—which is the same procedure used to generate the pre-training dataset equations. We tabulate these additional synthetic results for dimensions $d = \{2, 5\}$ in Table 30. To be consistent with prior work of Kamienny et al. (2022), that use this same experimental setup, we followed their experimental setup—which only evaluates each synthetic equation for one random seed. Here, we provide additional evaluation metrics that are computed *out of distribution*—that is we sample new points $\mathbf{X} \sim \mathcal{X}$ to form an out of distribution test set $\{f^*(\mathbf{X}), \mathbf{X}\}$. Specifically we further include the following metrics of: coefficient of determination $R^2$-score (La Cava et al., 2021), Test NMSE and accuracy to tolerance $\tau$ (Biggio et al., 2021; Kamienny et al., 2022). Where coefficient of determination $R^2$-score is defined as:

$$R^2 = 1 - \frac{\sum_{i=1}^{n}(y_i - \hat{y}_i)^2}{\sum_{i=1}^{n}(y_i - \bar{y})^2} \qquad\qquad \bar{y} = \frac{1}{n}\sum_{i=1}^{n} y_i \qquad (6)$$

Similarly the accuracy to tolerance $\tau$ is defined as (Kamienny et al., 2022):

$$\text{Acc}_\tau = \mathbb{1}\left(\max_{1 \leq i \leq n}\left|\frac{\hat{y}_i - y_i}{yi}\right| \leq \tau\right) \qquad (7)$$

where $\mathbb{1}$ is the indicator function.

We also provide a similar additional synthetic equation ablation of pre-training DGSR with a smaller number of input variables $d = 2$ than those seen at inference time $d = 5$—thereby further providing empirical evidence for the property of generalizing to unseen input variables (P3). This is detailed in Table 31.

Table 30: Additional synthetic experiments. Where: $\omega$ is the number of unique equations $f^*$ in a benchmark problem set, coefficient of determination $R^2$-score, $\text{Acc}_\tau$ is the accuracy to tolerance, where $\tau = 0.05$, MSE is the test mean squared error and $d$ is the number of input variables in the problem set. Here follow a different symbolic regression experimental setup, that of Kamienny et al. (2022) and synthetically a problem set of $\omega$ unique equations using the concise equation generator of Lample & Charton (2019) and only run each experiment for one random seed over a large number of unique equations $f^*$—as is recommended by Kamienny et al. (2022).

| | Problem set | $\omega$ | $d$ | DGSR (Ours) | NGGP (RL) |
|---|---|---|---|---|---|
| Average Rec. Rate (%) $A_{\text{Rec}}\%$ | Extended Synthetic (d=2) | 5,000 | 2 | **84.98** | 83.73 |
| | Extended Synthetic (d=5) | 5,000 | 5 | **84.18** | 83.19 |
| Average Eq. Evals $\gamma$ | Extended Synthetic (d=2) | 5,000 | 2 | **448,430** | 460,585 |
| | Extended Synthetic (d=5) | 5,000 | 5 | **527,879** | 582,542 |
| $R^2$-score | Extended Synthetic (d=2) | 5,000 | 2 | **0.9782** | 0.9720 |
| | Extended Synthetic (d=5) | 5,000 | 5 | **0.9864** | 0.9848 |
| Test NMSE | Extended Synthetic (d=2) | 5,000 | 2 | **0.00151** | 0.00371 |
| | Extended Synthetic (d=5) | 5,000 | 5 | **0.02096** | 0.02562 |
| $\text{Acc}_{0.05}$ | Extended Synthetic (d=2) | 5,000 | 2 | **84.14** | 81.34 |
| | Extended Synthetic (d=5) | 5,000 | 5 | **80.00** | 78.81 |

Table 31: DGSR ablation study using average recovery rate ($A_{\text{Rec}}\%$) on a generated synthetic dataset $d = 5$ benchmark problem set. Where $d$ is the number of input variables.

| Config | $\omega$ | Average recovery rate (%) $A_{\text{Rec}}\%$ |
|---|---|---|
| Pre-trained dataset $(d = 5) = (d_{\text{inference}} = 5)$ | 5,000 | **84.18** |
| Pre-trained dataset $(d = 2) < (d_{\text{inference}} = 5)$ | 3,000 | 78.93 |

Table 32: Standard symbolic regression benchmark problem specifications. Input variables can be $x_1, x_2$. $U(a, b, c)$ corresponds to $c$ random points uniformly sampled between $a$ to $b$ for each input variable separately, where the training and test datasets use different random seeds. $E(a, b, c)$ corresponds to $c$ points evenly spaced between $a$ and $b$ for each input variable; where training and test datasets use the same points. Where $\mathcal{L}_{\text{Koza}} = \{+, -, \div, \times, x_1, \exp, \log, \sin, \cos\}$.

| Name | Equation | Dataset | Library |
|---|---|---|---|
| Nguyen-1 | $x_1^3 + x_1^2 + x_1$ | $U(-1, 1, 20)$ | $\mathcal{L}_{\text{Koza}}$ |
| Nguyen-2 | $x_1^4 + x_1^3 + x_1^2 + x_1$ | $U(-1, 1, 20)$ | $\mathcal{L}_{\text{Koza}}$ |
| Nguyen-3 | $x_1^5 + x_1^4 + x_1^3 + x_1^2 + x_1$ | $U(-1, 1, 20)$ | $\mathcal{L}_{\text{Koza}}$ |
| Nguyen-4 | $x_1^6 + x_1^5 + x_1^4 + x_1^3 + x_1^2 + x_1$ | $U(-1, 1, 20)$ | $\mathcal{L}_{\text{Koza}}$ |
| Nguyen-5 | $\sin(x_1^2)\cos(x_1) - 1$ | $U(-1, 1, 20)$ | $\mathcal{L}_{\text{Koza}}$ |
| Nguyen-6 | $\sin(x_1) + \sin(x_1 + x_1^2)$ | $U(-1, 1, 20)$ | $\mathcal{L}_{\text{Koza}}$ |
| Nguyen-7 | $\log(x_1 + 1) + \log(x_1^2 + 1)$ | $U(0, 2, 20)$ | $\mathcal{L}_{\text{Koza}}$ |
| Nguyen-8 | $\sqrt{x_1}$ | $U(0, 4, 20)$ | $\mathcal{L}_{\text{Koza}}$ |
| Nguyen-9 | $\sin(x_1) + \sin(x_2^2)$ | $U(0, 1, 20)$ | $\mathcal{L}_{\text{Koza}} \cup \{x_2\}$ |
| Nguyen-10 | $2\sin(x_1)\cos(x_2)$ | $U(0, 1, 20)$ | $\mathcal{L}_{\text{Koza}} \cup \{x_2\}$ |
| Nguyen-11 | $x_1^{x_2}$ | $U(0, 1, 20)$ | $\mathcal{L}_{\text{Koza}} \cup \{x_2\}$ |
| Nguyen-12 | $x_1^4 - x_1^3 + \frac{1}{2}x_2^2 - x_2$ | $U(0, 1, 20)$ | $\mathcal{L}_{\text{Koza}} \cup \{x_2\}$ |
| Nguyen-1$^c$ | $3.39x_1^3 + 2.12x_1^2 + 1.78x_1$ | $U(-1, 1, 20)$ | $\mathcal{L}_{\text{Koza}} \cup \{\text{const}\}$ |
| Nguyen-5$^c$ | $\sin(x_1^2)\cos(x_1) - 0.75$ | $U(-1, 1, 20)$ | $\mathcal{L}_{\text{Koza}} \cup \{\text{const}\}$ |
| Nguyen-7$^c$ | $\log(x_1 + 1.4) + \log(x_1^2 + 1.3)$ | $U(0, 2, 20)$ | $\mathcal{L}_{\text{Koza}} \cup \{\text{const}\}$ |
| Nguyen-8$^c$ | $\sqrt{1.23x_1}$ | $U(0, 4, 20)$ | $\mathcal{L}_{\text{Koza}} \cup \{\text{const}\}$ |
| Nguyen-10$^c$ | $\sin(1.5x_1)\cos(0.5x_2)$ | $U(0, 1, 20)$ | $\mathcal{L}_{\text{Koza}} \cup \{x_2, \text{const}\}$ |
| R-1 | $\dfrac{(x_1+1)^3}{x_1^2 - x_1 + 1}$ | $E(-1, 1, 20)$ | $\mathcal{L}_{\text{Koza}}$ |
| R-2 | $\dfrac{x_1^5 - 3x_1^3 + 1}{x_1^2 + 1}$ | $E(-1, 1, 20)$ | $\mathcal{L}_{\text{Koza}}$ |
| R-3 | $\dfrac{x_1^6 + x_1^5}{x_1^4 + x_1^3 + x_1^2 + x_1 + 1}$ | $E(-1, 1, 20)$ | $\mathcal{L}_{\text{Koza}}$ |
| R-1$^*$ | $\dfrac{(x_1+1)^3}{x_1^2 - x_1 + 1}$ | $E(-10, 10, 20)$ | $\mathcal{L}_{\text{Koza}}$ |
| R-2$^*$ | $\dfrac{x_1^5 - 3x_1^3 + 1}{x_1^2 + 1}$ | $E(-10, 10, 20)$ | $\mathcal{L}_{\text{Koza}}$ |
| R-3$^*$ | $\dfrac{x_1^6 + x_1^5}{x_1^4 + x_1^3 + x_1^2 + x_1 + 1}$ | $E(-10, 10, 20)$ | $\mathcal{L}_{\text{Koza}}$ |

Table 33: Additional standard symbolic regression benchmark problem specifications. Input variables can be $x_1, x_2$. $U(a, b, c)$ corresponds to $c$ random points uniformly sampled between $a$ to $b$ for each input variable separately, where the training and test datasets use different random seeds. Where $\mathcal{L}_{\text{Koza}} = \{+, -, \div, \times, x_1, \exp, \log, \sin, \cos\}$.

| Name | Equation | Dataset | Library |
|---|---|---|---|
| Livermore-1 | $1/3 + x_1 + \sin(x_1^2)$ | $U(-10, 10, 1000)$ | $\mathcal{L}_{\text{Koza}}$ |
| Livermore-2 | $\sin(x_1^2)\cos(x_1) - 2$ | $U(-1, 1, 20)$ | $\mathcal{L}_{\text{Koza}}$ |
| Livermore-3 | $\sin(x_1^3)\cos(x_1^2) - 1$ | $U(-1, 1, 20)$ | $\mathcal{L}_{\text{Koza}}$ |
| Livermore-4 | $\log(x_1 + 1) + \log(x_1^2 + 1) + \log(x_1)$ | $U(0, 2, 20)$ | $\mathcal{L}_{\text{Koza}}$ |
| Livermore-5 | $x_1^4 - x_1^3 + x_1^2 - x_2$ | $U(0, 1, 20)$ | $\mathcal{L}_{\text{Koza}} \cup \{x_2\}$ |
| Livermore-6 | $4x_1^4 + 3x_1^3 + 2x_1^2 + x_1$ | $U(-1, 1, 20)$ | $\mathcal{L}_{\text{Koza}}$ |
| Livermore-7 | $\sinh(x_1)$ | $U(-1, 1, 20)$ | $\mathcal{L}_{\text{Koza}}$ |
| Livermore-8 | $\cosh(x_1)$ | $U(-1, 1, 20)$ | $\mathcal{L}_{\text{Koza}}$ |
| Livermore-9 | $x_1^9 + x_1^8 + x_1^7 + x_1^6 + x_1^5 + x_1^4 + x_1^3 + x_1^2 + x_1$ | $U(-1, 1, 20)$ | $\mathcal{L}_{\text{Koza}}$ |
| Livermore-10 | $6\sin(x_1)\cos(x_2)$ | $U(0, 1, 20)$ | $\mathcal{L}_{\text{Koza}} \cup \{x_2\}$ |
| Livermore-11 | $\frac{x_1^2 x_1^2}{x_1 + x_2}$ | $U(-1, 1, 50)$ | $\mathcal{L}_{\text{Koza}} \cup \{x_2\}$ |
| Livermore-12 | $x_1^5 / x_2^3$ | $U(-1, 1, 50)$ | $\mathcal{L}_{\text{Koza}} \cup \{x_2\}$ |
| Livermore-13 | $x_1^{1/3}$ | $U(0, 4, 20)$ | $\mathcal{L}_{\text{Koza}}$ |
| Livermore-14 | $x_1^3 + x_1^2 + x_1 + \sin(x_1) + \sin(x_1^2)$ | $U(-1, 1, 20)$ | $\mathcal{L}_{\text{Koza}}$ |
| Livermore-15 | $x_1^{1/5}$ | $U(0, 4, 20)$ | $\mathcal{L}_{\text{Koza}}$ |
| Livermore-16 | $x_1^{2/5}$ | $U(0, 4, 20)$ | $\mathcal{L}_{\text{Koza}}$ |
| Livermore-17 | $4\sin(x_1)\cos(x_2)$ | $U(0, 1, 20)$ | $\mathcal{L}_{\text{Koza}} \cup \{x_2\}$ |
| Livermore-18 | $\sin(x_1^2)\cos(x) - 5$ | $U(-1, 1, 20)$ | $\mathcal{L}_{\text{Koza}}$ |
| Livermore-19 | $x_1^5 + x_1^4 + x_1^2 + x_1$ | $U(-1, 1, 20)$ | $\mathcal{L}_{\text{Koza}}$ |
| Livermore-20 | $\exp(-x_1^2)$ | $U(-1, 1, 20)$ | $\mathcal{L}_{\text{Koza}}$ |
| Livermore-21 | $x_1^8 + x_1^7 + x_1^6 + x_1^5 + x_1^4 + x_1^3 + x_1^2 + x_1$ | $U(-1, 1, 20)$ | $\mathcal{L}_{\text{Koza}}$ |
| Livermore-22 | $\exp(-0.5x_1^2)$ | $U(-1, 1, 20)$ | $\mathcal{L}_{\text{Koza}}$ |

Table 34: Feynman benchmark problem specifications. Input variables can be $x_1, \ldots, x_2$. $U(a, b, c)$ corresponds to $c$ random points uniformly sampled between $a$ to $b$ for each input variable separately, where the training and test datasets use different random seeds. Where $\mathcal{L}_{\text{Koza}} = \{+, -, \div, \times, x_1, \exp, \log, \sin, \cos\}$.

| Name | Equation | Dataset | Library |
|---|---|---|---|
| Feynman-1 | $x_1 x_2$ | $U(1, 5, 20)$ | $\mathcal{L}_{\text{Koza}} \cup \{x_2\}$ |
| Feynman-2 | $\frac{x_1}{2(1+x_2)}$ | $U(1, 5, 20)$ | $\mathcal{L}_{\text{Koza}} \cup \{x_2\}$ |
| Feynman-3 | $x_1 x_2^2$ | $U(1, 5, 20)$ | $\mathcal{L}_{\text{Koza}} \cup \{x_2\}$ |
| Feynman-4 | $1 + \frac{x_1 x_2}{(1 - (x_1 x_2/3))}$ | $U(0, 1, 20)$ | $\mathcal{L}_{\text{Koza}} \cup \{x_2\}$ |
| Feynman-5 | $\frac{x_1}{x_2}$ | $U(1, 5, 20)$ | $\mathcal{L}_{\text{Koza}} \cup \{x_2\}$ |
| Feynman-6 | $\frac{1}{2} x_1 x_2^2$ | $U(1, 5, 20)$ | $\mathcal{L}_{\text{Koza}} \cup \{x_2\}$ |
| Feynman-7 | $\frac{3}{2} x_1 x_2$ | $U(1, 5, 20)$ | $\mathcal{L}_{\text{Koza}} \cup \{x_2\}$ |
| Feynman-8 | $\frac{x_1}{e^{\frac{x_4 x_5}{x_2 x_3}} + e^{\frac{-x_4 x_5}{x_2 x_3}}}$ | $U(1, 3, 50)$ | $\mathcal{L}_{\text{Koza}} \cup \{x_2, x_3, x_4, x_5\}$ |
| Feynman-9 | $x_1 x_2 x_3 \log \frac{x_5}{x_4}$ | $U(1, 5, 50)$ | $\mathcal{L}_{\text{Koza}} \cup \{x_2, x_3, x_4, x_5\}$ |
| Feynman-10 | $\frac{x_1(x_3 - x_2)x_4}{x_5}$ | $U(1, 5, 50)$ | $\mathcal{L}_{\text{Koza}} \cup \{x_2, x_3, x_4, x_5\}$ |
| Feynman-11 | $\frac{x_1 x_2}{x_5(x_3^2 - x_4^2)}$ | $U(1, 3, 50)$ | $\mathcal{L}_{\text{Koza}} \cup \{x_2, x_3, x_4, x_5\}$ |
| Feynman-12 | $\frac{x_1 x_2^2 x_3}{3 x_4 x_5}$ | $U(1, 5, 50)$ | $\mathcal{L}_{\text{Koza}} \cup \{x_2, x_3, x_4, x_5\}$ |
| Feynman-13 | $x_1(e^{\frac{x_2 x_3}{x_4 x_5}} - 1)$ | $U(1, 5, 50)$ | $\mathcal{L}_{\text{Koza}} \cup \{x_2, x_3, x_4, x_5\}$ |
| Feynman-14 | $x_5 x_1 x_2(\frac{1}{x_4} - \frac{1}{x_3})$ | $U(1, 5, 50)$ | $\mathcal{L}_{\text{Koza}} \cup \{x_2, x_3, x_4, x_5\}$ |
| Feynman-15 | $x_1(x_2 + x_3 x_4 \sin x_5)$ | $U(1, 5, 50)$ | $\mathcal{L}_{\text{Koza}} \cup \{x_2, x_3, x_4, x_5\}$ |

Table 35: Additional Feynman benchmark problem specifications. Input variables can be $x_1, \ldots, x_9$. $U(a, b, c)$ corresponds to $c$ random points uniformly sampled between $a$ to $b$ for each input variable separately, where the training and test datasets use different random seeds. Where $\mathcal{L}_{\text{Koza}} = \{+, -, \div, \times, x_1, \exp, \log, \sin, \cos\}$.

| Name | Equation | Dataset | Library |
|------|----------|---------|---------|
| Feynman-A-1 | $\frac{x_3 x_1 x_2}{(x_5 - x_4)^2 + (x_7 - x_6)^2 + (x_9 - x_8)^2}$ | $U(1, 2, 90)$ | $\mathcal{L}_{\text{Koza}} \cup \{x_2, \ldots, x_9\}$ |
| Feynman-A-2 | $\frac{x_1 x_2}{x_3 x_4} + \frac{x_1 x_5}{x_6 x_7^2 x_3 x_4} x_8$ | $U(1, 3, 80)$ | $\mathcal{L}_{\text{Koza}} \cup \{x_2, \ldots, x_8\}$ |
| Feynman-A-3 | $x_1 e^{\frac{-x_2 x_5 x_3}{x_6 x_4}}$ | $U(1, 5, 60)$ | $\mathcal{L}_{\text{Koza}} \cup \{x_2, \ldots, x_6\}$ |
| Feynman-A-4 | $x_1 x_4 + x_2 x_5 + x_3 x_6$ | $U(1, 5, 60)$ | $\mathcal{L}_{\text{Koza}} \cup \{x_2, \ldots, x_6\}$ |
| Feynman-A-5 | $x_1(1 + \frac{x_5 x_6 \cos(x_4)}{x2 * x3})$ | $U(1, 3, 60)$ | $\mathcal{L}_{\text{Koza}} \cup \{x_2, \ldots, x_6\}$ |
| Feynman-A-6 | $x_1(1 + x_3)x_2$ | $U(1, 5, 60)$ | $\mathcal{L}_{\text{Koza}} \cup \{x_2, \ldots, x_6\}$ |
| Feynman-A-7 | $\frac{x_1 x_4 x_2}{x_3}$ | $U(1, 5, 40)$ | $\mathcal{L}_{\text{Koza}} \cup \{x_2, \ldots, x_4\}$ |
| Feynman-A-8 | $\frac{x_1 x_2^2 x_3}{x_4}$ | $U(1, 5, 40)$ | $\mathcal{L}_{\text{Koza}} \cup \{x_2, \ldots, x_4\}$ |
| Feynman-A-9 | $\frac{1}{x_1 - 1} x_2 \frac{x_4}{x_3}$ | $U(2, 5, 40)$ | $\mathcal{L}_{\text{Koza}} \cup \{x_2, \ldots, x_4\}$ |
| Feynman-A-10 | $\frac{x_1 x_2 x_3}{2 x_4}$ | $U(1, 5, 40)$ | $\mathcal{L}_{\text{Koza}} \cup \{x_2, \ldots, x_4\}$ |
| Feynman-A-11 | $\frac{x_1 x_2 x_4}{x_3}$ | $U(1, 5, 40)$ | $\mathcal{L}_{\text{Koza}} \cup \{x_2, \ldots, x_4\}$ |
| Feynman-A-12 | $x_1(\cos(x_2 x_3) + x_4 \cos(x_2 x_3)^2)$ | $U(1, 3, 40)$ | $\mathcal{L}_{\text{Koza}} \cup \{x_2, \ldots, x_4\}$ |
| Feynman-A-13 | $-x_1 x_2 \frac{x_3}{x4}$ | $U(1, 5, 40)$ | $\mathcal{L}_{\text{Koza}} \cup \{x_2, \ldots, x_4\}$ |
| Feynman-A-14 | $\frac{x_1 x_3 + x_2 x_4}{x_1 + x_2}$ | $U(1, 5, 40)$ | $\mathcal{L}_{\text{Koza}} \cup \{x_2, \ldots, x_4\}$ |
| Feynman-A-15 | $\frac{1}{2} x_1(x_2^2 + x_3^2 + x_4^2)$ | $U(1, 5, 40)$ | $\mathcal{L}_{\text{Koza}} \cup \{x_2, \ldots, x_4\}$ |
| Feynman-A-16 | $-x_1 x_2 \cos(x_3)$ | $U(1, 5, 30)$ | $\mathcal{L}_{\text{Koza}} \cup \{x_2, \ldots, x_3\}$ |
| Feynman-A-17 | $\frac{x3 + x2}{1 + \frac{x_3 x_2}{x1^2}}$ | $U(1, 5, 30)$ | $\mathcal{L}_{\text{Koza}} \cup \{x_2, \ldots, x_3\}$ |
| Feynman-A-18 | $x_1 x_2 x_3$ | $U(1, 5, 30)$ | $\mathcal{L}_{\text{Koza}} \cup \{x_2, \ldots, x_3\}$ |
| Feynman-A-19 | $x_1 x_2 x_3^2$ | $U(1, 5, 30)$ | $\mathcal{L}_{\text{Koza}} \cup \{x_2, \ldots, x_3\}$ |
| Feynman-A-20 | $x_1 x_2 \frac{x_3}{2}$ | $U(1, 5, 30)$ | $\mathcal{L}_{\text{Koza}} \cup \{x_2, \ldots, x_3\}$ |
| Feynman-A-21 | $\frac{1}{x_1 - 1} x_2 x_3$ | $U(2, 5, 30)$ | $\mathcal{L}_{\text{Koza}} \cup \{x_2, \ldots, x_3\}$ |
| Feynman-A-22 | $\frac{x_3}{1 - \frac{x_2}{x_1}}$ | $U(3, 10, 30)$ | $\mathcal{L}_{\text{Koza}} \cup \{x_2, \ldots, x_3\}$ |
| Feynman-A-23 | $x_1 x_3 x_2$ | $U(1, 5, 30)$ | $\mathcal{L}_{\text{Koza}} \cup \{x_2, \ldots, x_3\}$ |
| Feynman-A-24 | $\frac{x_1 \sin(x_3 \frac{x_2}{2})^2}{\sin(x_2/2)^2}$ | $U(1, 5, 30)$ | $\mathcal{L}_{\text{Koza}} \cup \{x_2, \ldots, x_3\}$ |
| Feynman-A-25 | $x_1(1 + x_2 \cos(x_3))$ | $U(1, 5, 30)$ | $\mathcal{L}_{\text{Koza}} \cup \{x_2, \ldots, x_3\}$ |
| Feynman-A-26 | $\frac{1}{\frac{1}{x_1} + \frac{x_3}{x_2}}$ | $U(1, 5, 30)$ | $\mathcal{L}_{\text{Koza}} \cup \{x_2, \ldots, x_3\}$ |
| Feynman-A-27 | $2x_1(1 - \cos(x_2 x_3))$ | $U(1, 5, 30)$ | $\mathcal{L}_{\text{Koza}} \cup \{x_2, \ldots, x_3\}$ |
| Feynman-A-28 | $\frac{x_1}{x_2(1 + x_3)}$ | $U(1, 5, 30)$ | $\mathcal{L}_{\text{Koza}} \cup \{x_2, \ldots, x_3\}$ |
| Feynman-A-29 | $(\frac{x_1 x_2 x_3 x_4 x_5}{4 x_6 \sin(x_7/2)^2})^2$ | $U(1, 2, 70)$ | $\mathcal{L}_{\text{Koza}} \cup \{x_2, \ldots, x_7\}$ |
| Feynman-A-30 | $\frac{x_1}{1 + x_1/(x_2 x_3^2)(1 - \cos(x_4))}$ | $U(1, 3, 40)$ | $\mathcal{L}_{\text{Koza}} \cup \{x_2, \ldots, x_4\}$ |
| Feynman-A-31 | $\frac{x_1(1 - x_2^2)}{1 + x_2 \cos(x_3 - x_4)}$ | $U(1, 3, 40)$ | $\mathcal{L}_{\text{Koza}} \cup \{x_2, \ldots, x_4\}$ |
| Feynman-A-32 | $x_1 \frac{\sin(x_2/2) sin(x_4 x_3/2)}{(x_2/2 \sin(x_3/2))}^2$ | $U(4, 6, 40)$ | $\mathcal{L}_{\text{Koza}} \cup \{x_2, \ldots, x_4\}$ |

Table 36: Synthetic $d = 12$ benchmark problem specifications. $U(a, b, c)$ corresponds to $c$ random points uniformly sampled between $a$ to $b$ for each input variable separately, where the training and test datasets use different random seeds. Where $\mathcal{L}_{\text{Synth}} = \{+, -, \div, \times, x_1, \ldots, x_{12}\}$.

| Name | Equation | Dataset | Library |
|------|----------|---------|---------|
| Synthetic-1 | $x_{12} + x_9(x_{10} + x_{11}) + x_1 + x_2 + x_3 + x_4 + x_5 + x_6 + x_7 x_8$ | $U(-1, 1, 120)$ | $\mathcal{L}_{\text{Synth}}$ |
| Synthetic-2 | $x_{10} + x_{11} + x_{12} + x_3(x_1 + x_2) + x_4 x_5 + x_6 + x_7 + x_8 + x_9$ | $U(-1, 1, 120)$ | $\mathcal{L}_{\text{Synth}}$ |
| Synthetic-3 | $x_{10} + x_9(x_1 + x_2 + x_3 + x_4 + x_5 + x_6 + x_7 + x_8) + x_{11} + x_{12}$ | $U(-1, 1, 120)$ | $\mathcal{L}_{\text{Synth}}$ |
| Synthetic-4 | $x_8(x_6 + x_7) - (x_{10} + x_{11} x_{12} + x_9)x_1 + x_2 + x_3 + x_4 + x_5$ | $U(-1, 1, 120)$ | $\mathcal{L}_{\text{Synth}}$ |
| Synthetic-5 | $x_{10} + x_{11} + x_{12} + x_9(x_1 + x_2) - x_3 + x_4 + x_5 + x_6 + x_7 + x_8$ | $U(-1, 1, 120)$ | $\mathcal{L}_{\text{Synth}}$ |
| Synthetic-6 | $x_1(x_{10} - x_{11}) - x_{12} + x_2 + x_3 + x_4 + x_5 + x_6 + x_7 + x_8 + x_9$ | $U(-1, 1, 120)$ | $\mathcal{L}_{\text{Synth}}$ |
| Synthetic-7 | $x_1 x_2 - x_{11}(-x_{10} + x_6 + x_7) + x_8 - x_9 + x_{12} + x_3 + x_4 + x_5$ | $U(-1, 1, 120)$ | $\mathcal{L}_{\text{Synth}}$ |

Table 37: DGSR equivalent $f^*$ generated equations at inference time, for problem Feynman-7.

| True equation ($f^*$) | Equivalent generated equations | |
|---|---|---|
| $\frac{3}{2}x_1x_2$ | $x_1(x_2 + \frac{x_2x_2}{x_2+x_2})$ | $x_1(x_2 + \frac{x_2}{\frac{1}{x_2}x_1\frac{x_2+x_2}{x_1}})$ |
| $\frac{3}{2}x_1x_2$ | $x_2(x_1 + x_2\frac{x_1}{x_2+x_2})$ | $x_1(x_2x_2\frac{\frac{1}{x_2+x_2}}{x_2} + x_2)$ |
| $\frac{3}{2}x_1x_2$ | $x_1(x_2\frac{x_2}{x_2+x_2} + x_2)$ | $x_2(x_1\frac{x_2}{x_2} + x_1\frac{x_2}{x_2+x_2})$ |
| $\frac{3}{2}x_1x_2$ | $x_1(x_2\frac{x_2}{x_2+x_2} + x_2)$ | $x_2(x_1\frac{\frac{x_2}{x_2+x_2}}{x_1} + x_1)$ |
| $\frac{3}{2}x_1x_2$ | $x_1(x_2\frac{x_1}{x_1+x_1} + x_2)$ | $x_2(x_1\frac{\frac{x_1}{x_2}(x_1+x_1)}{x_2} + x_1)$ |
| $\frac{3}{2}x_1x_2$ | $x_1(x_2\frac{x_1}{x_1+x_1} + x_2)$ | $x_2(x_1\frac{\frac{x_2}{x_1}x_2}{x_1+x_1} + x_1)$ |
| $\frac{3}{2}x_1x_2$ | $x_2(x_1 + \frac{x_1}{\frac{1}{x_1}(x_1+x_1)})$ | $x_1(x_2\frac{x_2}{x_2+(x_2+(x_2(-1)+x_2))} + x_2)$ |
| $\frac{3}{2}x_1x_2$ | $x_2(x_1\frac{x_1}{x_1+x_1} + x_1)$ | $x_1(x_2\frac{x_2}{x_2\frac{x_2}{x_2}+x_2} + x_2)$ |
| $\frac{3}{2}x_1x_2$ | $x_2(x_1\frac{x_2}{x_2+x_2} + x_1)$ | $x_2(x_1\frac{x_2}{x_2}\frac{x_2}{x_2+x_2} + x_1)$ |
| $\frac{3}{2}x_1x_2$ | $x_2(x_1\frac{x_2}{x_2+x_2} + x_1)$ | $x_2(x_1\frac{x_2}{x_2+\frac{x_1x_2}{x_1}} + x_1)$ |
| $\frac{3}{2}x_1x_2$ | $x_1(x_2 + \frac{x_2}{\frac{1}{x_2}(x_2+x_2)})$ | $(x_1 + \frac{x_1}{\frac{1}{x_2}(x_2+x_2)})(x_2(-1) + (x_2+x_2))$ |
| $\frac{3}{2}x_1x_2$ | $x_2(x_1 + \frac{x_1x_2}{x_2+x_2})$ | $x_2(x_1 + \frac{x_1}{\frac{1}{x_2}(x_2\frac{x_1}{x_1}+x_2)})$ |
| $\frac{3}{2}x_1x_2$ | $x_2(x_1\frac{x_1}{x_1+x_1} + x_1)$ | $x_1(x_2\frac{x_2}{x_2+(x_2+(x_1(-1)+x_1))} + x_2)$ |
| $\frac{3}{2}x_1x_2$ | $x_2(x_1 + x_2\frac{x_1}{x_2+x_2})$ | $x_2(x_1 + \frac{x_1}{\frac{1}{x_2}\frac{x_1}{x_2}(x_1+x_1)})$ |
| $\frac{3}{2}x_1x_2$ | $x_2(x_1\frac{x_2}{x_2+x_2} + x_1)$ | $x_2(x_1 + \frac{x_1}{\frac{1}{x_2}(x_2\frac{x_1}{x_1}+x_2)})$ |
| $\frac{3}{2}x_1x_2$ | $x_1(x_1\frac{x_2}{x_1+x_1} + x_2)$ | $x_2(x_1 + \frac{x_2\frac{x_1}{\frac{1}{x_2}(x_2+x_2)}}{\frac{x_2}{x_1}})$ |
| $\frac{3}{2}x_1x_2$ | $x_2(x_1 + x_2\frac{x_1}{x_2+x_2})$ | $x_2(x_1 + \frac{x_2\frac{x_1}{\frac{1}{x_1}(x_1+x_1)}}{x_2})$ |
| $\frac{3}{2}x_1x_2$ | $x_2(x_1 + \frac{x_1}{\frac{1}{x_2}(x_2+x_2)})$ | $x_1(x_2 + \frac{x_2}{\frac{1}{x_2}x_2\frac{x_2+x_2}{x_2}})$ |
| $\frac{3}{2}x_1x_2$ | $x_1(x_2\frac{x_2}{x_2+x_2} + x_2)$ | $x_1(x_2\frac{x_2}{x_2+x_2} + (x_2 + (x_2(-1)+x_2)))$ |
| $\frac{3}{2}x_1x_2$ | $x_1(x_2 + (x_2 + x_1\frac{x_2}{x_1+x_1}(-1)))$ | $x_1(x_2 + \frac{x_1x_2}{x_2}\frac{x_2}{x_1+x_1})$ |
| $\frac{3}{2}x_1x_2$ | $x_2(x_1 + (x_1 + x_1\frac{x_2}{x_2+x_2}(-1)))$ | $x_2(x_1\frac{x_2}{x_2+\frac{x_2x_2}{x_2}} + x_1)$ |
| $\frac{3}{2}x_1x_2$ | $x_1(x_2 + (x_2 + x_2\frac{x_2}{x_2+x_2}(-1)))$ | $x_1(x_2\frac{x_2}{x_2\frac{x_2}{x_2}+x_2} + x_2)$ |
| $\frac{3}{2}x_1x_2$ | $x_2(x_1 + (x_1 + \frac{x_1}{\frac{1}{x_2}(x_2+x_2)}(-1)))$ | $x_2(x_1\frac{x_1}{x_1\frac{x_1+x_1}{x_1}} + x_1)$ |
| $\frac{3}{2}x_1x_2$ | $x_1(x_1\frac{x_1\frac{x_2}{x_1}}{x_1+x_1} + x_2)$ | $x_1(x_1\frac{x_2}{x_1+\frac{x_1x_2}{x_2}} + x_2)$ |
| $\frac{3}{2}x_1x_2$ | $x_1(x_2 + \frac{x_2}{\frac{1}{x_2}x_1\frac{x_2+x_2}{x_1}})$ | $x_2(x_1\frac{\frac{1}{x_2}x_2x_2}{x_2+x_2} + x_1)$ |
| $\frac{3}{2}x_1x_2$ | $x_1(x_2 + \frac{x_2}{\frac{1}{x_1}x_1\frac{x_2+x_2}{x_2}})$ | $x_2(x_1\frac{\frac{1}{x_2}x_2}{\frac{1}{x_1}(x_1+x_1)} + x_1)$ |
| $\frac{3}{2}x_1x_2$ | $x_1\frac{x_2}{x_2}(x_1\frac{x_2}{x_1+x_1} + x_2)$ | $x_1(x_2\frac{\frac{1}{x_2}(x_2+x_2)}{x_2} + x_2)$ |
| $\frac{3}{2}x_1x_2$ | $x_1(x_2\frac{x_2}{x_2+(x_2+(x_1(-1)+x_1))} + x_2)$ | $x_2(x_1\frac{x_2}{x_1\frac{x_2}{x_1}+x_2} + x_1)$ |
| $\frac{3}{2}x_1x_2$ | $x_2(x_1\frac{x_2}{x_1\frac{x_2}{x_1}+x_2} + x_1)$ | $x_1(x_2\frac{x_2}{x_2}\frac{x_2}{x_2+x_2} + x_2)$ |
| $\frac{3}{2}x_1x_2$ | $x_2(x_1 + \frac{x_1}{x_1+x_1}\frac{x_1x_2}{x_2})$ | $x_1(x_2\frac{x_2}{x_2+(x_2(x_1(-1)+x_1)+x_2)} + x_2)$ |
| $\frac{3}{2}x_1x_2$ | $x_2((x_1 + \frac{x_1}{\frac{1}{x_2}(x_2+x_2)}) + (x_1(-1)+x_1))$ | $x_2(x_1 + \frac{x_1}{\frac{1}{x_2}(x_2+(x_2+\frac{x_2(-1)+x_2}{x_2}))})$ |
| $\frac{3}{2}x_1x_2$ | $x_2(x_1\frac{x_2}{x_2+(x_2+(x_2(-1)+x_2))} + x_1)$ | $x_1(x_2\frac{x_2+x_2}{x_2+(x_2+(x_2+x_2))} + x_2)$ |
| $\frac{3}{2}x_1x_2$ | $x_1\frac{x_2(x_2\frac{x_2}{x_2+x_2}+x_2)}{x_2}$ | $x_1(x_2\frac{x_2}{x_2+(x_1(x_1(-1)+x_1)+x_2)} + x_2)$ |