# OpenReview forum: "Deep Generative Symbolic Regression"
_ICLR.cc/2023/Conference — ICLR 2023 poster_

### Official Review · Reviewer_JMzJ · 2022-10-18

**Confidence:** 4
**Correctness:** 2
**Technical Novelty And Significance:** 2
**Empirical Novelty And Significance:** 3
**Recommendation:** 5

**Clarity, Quality, Novelty And Reproducibility:**

**Clarity** - The main paper is clearly written, but a lot of important elements have to be looked up in the supplementary. Unfortunately, large parts of the supplementary material are a long version of the paper, with many duplicates (or close duplicates), which does not help clarity. The authors would be advised to structure the appendix for easier reference, and eliminate duplicates.

**Novelty** - The basic architecture (set transformer for encoder, and autoregressive decoder) is very close to Biggio (2021). The data generation comes from Lample (2019), and pre-training procedures and losses. To my knowledge, the gradient-based inference retraining proposed by the authors is a new idea. Note that previous authors use MSE-based fine-tuning of the raw predictions of their pre-trained model. For instance, D'Ascoli (2022) and Kamienny (2022) uses MSE error to guide beam search. As such, the "cannot refine" mention in table 1 seems misleading. The authors should mention fine-tuning in other transformer-based approaches, and discuss the specificities of their approach (there is an obvious trade-off between performance and speed, here).

**Quality** - see weaknesses, in the previous section. In my opinion, the claims P1, P2, and P3 are insufficiently substantiated.

**Reproducibility** -
The information provided by the authors should allow one to reproduce their results.

**Strength And Weaknesses:**

**Strength**

Symbolic Regression is an important field, and the authors propose an interesting architecture, with promising results (albeit on very small test sets). The paper is interesting, and clearly written.

**Weaknesses**

The tests sets are limited to very small benchmark suites. For instance, the results from table 2 (the main claims about the method success)  are based on 54 equations, and the test set substantiating P3 has only 8 equations (Feynman d=5). The authors should back up their claims with results from large generated datasets (e.g. using the Lample & Charton generator they use for pre-training data set). Without additional tests, claim P3 should be removed, because it is not substantiated.

The recovery rate, the only evaluation criterion presented in the main, makes very little sense in practical applications. First, it limits tests to artificial situations where the function to be found is already known to the experimenter. Second, it precludes the use of real constants in the formulas (which are always approximate). This is a standard problem in SR, and the main rationale for the approaches of Biggio and Kamienny (suggesting the authors are tackling a much simpler problem). Finally, it assumes an in-domain solution, that can be expressed within the pre-training model vocabulary - a very strong limitation. Alternate metrics exist, like R2, MSE on test data, or (better) extrapolation MSE on test data (i.e. for each test function split the datapoints into two subsets $(x,f(x))_{infer}$, used for inference fine-tuning, and $(x,f(x))_{test}$ used for testing). The authors should prefer them in the main.

On claim P2 (inference speed), the authors have a point that their method is faster than RL and GP approaches. But I fail to see how it can be faster than the end-to-end approaches of Biggio, d'Ascoli and Kamienny, which use *no gradient based-retraining*. While this is not necessarily a problem (most practical uses of SR can accommodate slow inference), this should be mentioned more clearly in the paper. Also, could inference speeds (in seconds) be provided?

On claim P1, the fact that symbolic transformers, trained on cross-entropy, can learn equivalence between different equations, has been observed in many previous papers. Lample (2019) mentions it (section 4.6), and D'Ascoli exploits this property to find alternate solutions (closed forms vs recurrences corresponding to the same data). D'Ascoli also observes that training from simplified formulas does not improve accuracy, which further suggests that equivalence has been learned. DGSR shows that training on MSE can also achieves this, a new result to my knowledge, but this is by no means a "first" (as table 1 suggests). This should be discussed, and claim P1 should be toned down.

In section 3.1, the authors insist on the permutation equivariance of features (the d dimensions of the problem). While this might help with benchmarks such as SRBench, where many equations display this feature, I am not sure it is a good practical idea. In most physical models, features will not be permutation equivariant, because they have different units. To my knowledge, other SR approaches do not try to enforce such feature permutation equivariance. The authors should either drop it, or better justify its necessity.



**Summary Of The Paper:**

The authors propose a pre-trained generative model for solving Symbolic Regression for multivariate problems in low dimension (from 2 to 12 dimensions).

A sequence to sequence model, a set transformer encoder and an autoregressive decoder (as in Biggio 2021), is trained on generated datasets of values (X, f(X)) of random functions f, using the (weighted) mean squared error of model predictions. At inference, model predictions are refined by training the decoder, again with a mean squared error loss, while keeping the encoder weights frozen. Finally, a top-k Monte Carlo search is performed to find a best solution (according to the mean squared error).

The authors evaluate their model on a set of benchmarks (over 300 functions overall). The approach achieves state-of-the-art performance on the recovery rate (percentage of correct solutions retrieved).

**Summary Of The Review:**

Overall, the paper is interesting and the results are promising, but the experimental methodology is very disappointing. The test sets are too small to back some of the claims (P3), the main metric is of little interest in practice (it expects the solution is already known, and doesn't account for approximate solutions, or out of domain situations).

The comparison with other approaches is also unsatisfying. Inference speed, one of the basic claims of the authors, is only compared with the slowest models (GP and RL). The fact that other models learn invariance is not credited.

I would encourage the authors to back up their claims with stronger test results. If this can be done during the rebuttal period, I will be happy to increase my recommendation.

---

> ### Author Response · Authors · 2022-11-11
> **Response to Reviewer JMzJ [Part 4/4]**
>
> ___
>
> **(E) Clarity**
>
> **UPDATE:** We have reformulated the appendices table of contents into a hierarchical table of contents to better organize the existing appendices—thereby providing an easier reference for the reader.
>
> ___
>
> **References**
> * Terrell N. Mundhenk, Mikel Landajuela, Ruben Glatt, Claudio P. Santiago, Daniel faissol, and Brenden K. Petersen. Symbolic regression via deep reinforcement learning enhanced genetic programming seeding. In Advances in Neural Information Processing Systems, 2021.
> * Brenden K Petersen, Mikel Landajuela Larma, Terrell N Mundhenk, Claudio Prata Santiago, Soo Kyung Kim, and Joanne Taery Kim. Deep symbolic regression: Recovering mathematical expressions from data via risk-seeking policy gradients. In International Conference on Learning Representations, 2020.
> * Allan Costa, Rumen Dangovski, Owen Dugan, Samuel Kim, Pawan Goyal, Marin Soljačić, and Joseph Jacobson. Fast neural models for symbolic regression at scale. arXiv preprint arXiv:2007.10784, 2020.
> * Guillaume Lample and François Charton. Deep learning for symbolic mathematics. In International Conference on Learning Representations, 2019.
> * Pierre-Alexandre Kamienny, Stéphane d'Ascoli, Guillaume Lample, and François Charton. End-to-end symbolic regression with transformers. arXiv preprint arXiv:2204.10532, 2022.
> * William La Cava, Patryk Orzechowski, Bogdan Burlacu, Fabricio Olivetti de Franca, Marco Virgolin, Ying Jin, Michael Kommenda, and Jason H Moore. Contemporary symbolic regression methods and their relative performance. In Thirty-fifth Conference on Neural Information Processing Systems Datasets and Benchmarks Track (Round 1), 2021.
> * Luca Biggio, Tommaso Bendinelli, Alexander Neitz, Aurelien Lucchi, and Giambattista Parascandolo. Neural symbolic regression that scales. In International Conference on Machine Learning, pp. 936–945. PMLR, 2021.
> * Douglas Adriano Augusto and Helio JC Barbosa. Symbolic regression via genetic programming. In Proceedings. Vol. 1. Sixth Brazilian Symposium on Neural Networks, pp. 173–178. IEEE, 2000.
> * Aaron Meurer, Christopher P Smith, Mateusz Paprocki, Ondřej Čertík, Sergey B Kirpichev, Matthew Rocklin, AMiT Kumar, Sergiu Ivanov, Jason K Moore, Sartaj Singh, et al. Sympy: symbolic computing in python. PeerJ Computer Science, 3:e103, 2017.
> * Stéphane d’Ascoli, Pierre-Alexandre Kamienny, Guillaume Lample, and Francois Charton. Deep symbolic regression for recurrence prediction. In International Conference on Machine Learning, pp. 4520–4536. PMLR, 2022.

---

> ### Author Response · Authors · 2022-11-11
> **Response to Reviewer JMzJ [Part 3/4]**
>
> **(C.2) Learning equation invariances**
>
> Indeed other prior works (Lample & Charton, 2019; d’Ascoli et al., 2022) have also been able to learn equivalences between different equations.
>
> First, let us reiterate the differences between a cross entropy loss function and that of our framework’s end-to-end loss function. Using a cross entropy loss with the predicted sequence of tokens $\bar{\hat{f}}$ to that of the tokens of the known ground truth equation $f^*$ in the training set, i.e., $\text{CE}(\bar{f^*},\bar{\hat{f}})$—will have a different loss value for different equation forms that are mathematically equivalent—therefore the sequence token representation alone is unable to encode any invariance structure, e.g., $x_1 + x_2$ and $x_2 + x_1$ will be deemed as two different sequences. Whereas using our proposed end-to-end NMSE loss, $\text{NMSE}(y,\hat{f}(X))$—will have the same loss value for different equation forms that are mathematically equivalent—therefore this loss is a natural and principled way to incorporate the equation equivalence property, inherent to symbolic regression.
>
> Second, we agree that prior works, specifically Lample & Charton (2019), observed the “surprising” and “very intriguing” result that sampling multiple equations from their pre-trained encoder-decoder model yielded some equations that are equivalent mathematically—which was trained with a cross entropy loss. Furthermore, we agree that the pioneering work of d’Ascoli et al. (2022) has shown this behavior as well, as kindly suggested.
>
> Finally, our framework DGSR, that uses its specific end-to-end NMSE loss is able to
> *better* learn equation equivalences—which is reflected by achieving a higher recovery rate of the true equation, as shown in Table 2 and in Section 5.2.
>
> **UPDATE:** Indeed prior works are also able to learn equation equivalences, and we have further clarified this—clearly stating this in the related work Table 1, and added a discussion of this in the related work, in Section 4.
>
> **(C.3) Gradient refinement at inference**
>
> We clarify that the unique aspect of DGSR is that it is able to perform **gradient** refinement at inference time—thereby updating the weights of the decoder at inference time to converge the generative model to have a high probability density where the true equation of interest $f^*$ lies. Thereby, allowing sampled equations that are drawn from $p_\theta(f|\mathcal{D})$ to have a high probability of generating the true equation $f^*$.
>
> Indeed there exist other non-gradient refinement symbolic regression approaches—that are unable to update their decoder’s weights. These consist of: (1) optimizing the constants in the generated equation form with a secondary optimization step (commonly using the BFGS algorithm) (Petersen et al., 2020; Biggio et al., 2021), and (2) using the MSE of the predicted equation(s) to guide a beam search sampler (d’Ascoli et al., 2022; Kamienny et al., 2022).
>
> **UPDATE:** We have updated the related work to discuss the above other non-gradient refinement methods and have clarified throughout that we explicitly perform gradient refinement.
>
> ___
>
> **(D) Invariances in the architecture**
>
> Certainly there exist many example equations $f$ that are not permutation-invariant to the individual input variables and we have removed the permutation equivariant across the features property.
>
> **UPDATE:** We have amended Section 3.1 to highlight only the two necessary properties that the encoder-decoder architecture should satisfy of: (1) having an encoding function that is *permutation invariant* across the input samples $n$ in the dataset $\mathcal{D}$ and (2) having a decoder that is *autoregressive* to allow the sampling of equations $\hat{f}$.

---

> ### Author Response · Authors · 2022-11-11
> **Response to Reviewer JMzJ [Part 2/4]**
>
> ___
> **(B) Evaluation metrics**
>
> We highlight that the goal of a *good* symbolic regression method is to find a concise mathematical equation $f$ that best fits a given dataset $\mathcal{D}$. These identified equations have concise closed-form expressions; thus, they are interpretable to human experts and amenable to further mathematical analysis (Augusto & Barbosa, 2000). This is in contrast to many existing supervised learning methods, such as deep neural networks, that produce black box models.
>
> Given this goal we evaluate against the standard symbolic regression metric of recovery rate ($A_{\text{Rec}}\\%$)—that is the percentage of runs where the true equation $f^*$ was found, over a set number of $\kappa$ random seed runs—as used by many prior works (Petersen et al., 2020; Mundhenk et al., 2021; La Cava et al., 2021). Where the average recovery rate is evaluated on both the functional form and the optimized numeric constants of the equation, as determined by a computer algebraic system, SymPy (Meurer et al., 2017).
>
> Indeed, other metrics have been used by prior works in symbolic regression (Biggio et al., 2021; Kamienny et al., 2022), such as test mean squared error (MSE). However we emphasize that our goal is to find a concise simple closed-form expression $f$, rather than fit an overly complex equation with many terms to achieve a low test MSE. Naturally, if we find the correct true equation $f^*$ for a given problem—then additional other evaluation metrics that measure fit are satisfied to a perfect score, such as a: test MSE of 0.0, test extrapolation MSE of 0.0, $R^2$-score of 1.0 and an accuracy to tolerance $\tau$ of 100%.
>
> Furthermore, we investigate the Pareto front of the test mean squared error compared to the equation complexity (which is a tradeoff, as one can always achieve a lower test MSE with a more complex equation), in Section 5.2 with plots in Figure 4, which is further detailed in Appendix N.
>
> Moreover, we gently reiterate that we evaluate DGSR on the SRBench blackbox benchmark (La Cava et al., 2021) in Appendix S—where the problems have no known true equation $f^*$. Rather, as quoted with this benchmark, the test RMSE error, $R^2$-score and inference time in seconds are evaluated against. Specifically, we observe that DGSR still remains competitive, ranking fourth in the lowest test RMSE, out of the 24 SRBench implemented methods—of which 7 of these are non-symbolic regression methods, e.g., a neural network.
>
> **UPDATE:** We further quote the additional evaluation metrics for the additional synthetic experiments in (A), of the out of distribution: $R^2$-score, test normalized mean squared error, and accuracy to tolerance $\tau=0.05$. Additionally we include the above discussion in the existing **Appendix K** of evaluation metrics.
>
> ___
>
> **(C) Learning invariances**
>
> **(C.1) Inference speed**
>
> Indeed pre-trained encoder-decoder methods are the most efficient at inference, however they are unable to update the weights of their decoders.
>
> We kindly reiterate that our core claim is that “DGSR uses less inference compute compared to RL techniques”. Of course pre-trained encoder-decoder methods are the most efficient at inference, and we originally reflect this in the paper by: (1) discussing in the related work that the pre-trained encoder-decoder achieves “low complexity at inference” due to only sampling, and (2) clearly including the inference compute results for the pre-trained encoder-decoder method NESYMRES in the main results of Table 2. Specifically, the pre-trained encoder-decoder method of NESYMRES achieves the lowest inference compute of 256 average equation evaluations, compared to 23,969 average equation evaluations for the Feynman (d=2) problem set. However, we reiterate as stated in Section 5.1, “NESYMRES achieves the lowest number of equation evaluations, however, suffers from a significantly lower recovery rate”.
>
> Moreover, when providing empirical evidence for this claim we prefer the average number of equation evaluations $\gamma$ until the true equation $f^*$ is found. We use this metric as a proxy for computational complexity across the benchmark algorithms, as testing many generated equations is a bottleneck in symbolic regression (Petersen et al., 2020). For example, analyzing the standard symbolic regression benchmark problem Nguyen-7c, DGSR finds the true equation in $\gamma = $20,187 equation evaluations, taking a total of 1 minute and 36.9 seconds; whereas NGGP finds the true equation in $\gamma = $30,112 equation evaluations, taking a total of 2 minutes and 20.5 seconds, both results averaged over $\kappa =$10 random seeds, further detailed in Appendix H.
>
> **UPDATE:** Indeed this aspect is important to clarify. We have now updated the related work in Section 4 to make this clearer.

---

> ### Author Response · Authors · 2022-11-11
> **Response to Reviewer JMzJ [Part 1/4]**
>
> Thank you for your thoughtful comments and suggestions! We give answers to each in turn, as well as pointing out corresponding updates to the revised manuscript (additions are indicated in blue):
>
> * (A) Additional results on synthetic equations
> * (B) Evaluation metrics
> * (C) Related work
> * (D) Invariances in the architecture
> * (E) Clarity
>
> ___
>
> **(A) Additional results on synthetic equations**
>
> We performed additional experiments and collected results for a further $\omega=2,200$ synthetically generated equations (A.1)—this consistently verifies our core claim throughout the paper that “DGSR has a higher recovery rate of the true underlying equation in the setting of a larger number of input variables, whilst using less inference compute compared to RL techniques”.
>
> We reiterate that our original experiments evaluated a total of $\omega=311$ unique equations $f^*$, including the additional standard equation problem sets in Appendix H. Therefore, the total number of unique equations $f^*$ that has been evaluated against is $\omega=2,511$ in this work. Crucially, we note that this already surpasses the other standard prior symbolic regression works total amount of unique equations evaluated on, specifically: Mundhenk et al. (2021) uses $\omega=55$, Petersen et al. (2020) uses $\omega=36$ and Costa et al. (2020) uses $\omega=16$. Moreover, we would like to emphasize that as is consistent with the related prior work, for each equation that we evaluate against, we perform the same experiment across a number of random seeds $\kappa=\{10,20\}$ in all our results in the main paper.
>
> We kindly emphasize that due to our limited computational resources of one GPU, we are unable to evaluate millions of synthetically generated equations, however we believe that the results for the combined total $\omega=2,511$ unique equations sufficiently support our claims and contributions.
>
> **(A.1) Additional synthetic experimental results**
>
> The additional synthetic results are in a new Appendix Y labelled “Additional synthetic experiments”, detailed in Tables 30 & 31.
>
> To provide further empirical results of DGSR, we synthetically generated $\omega$ equations using the concise and seminal equation generation framework of Lample & Charton (2019)—which is the same procedure used to generate the pre-training dataset equations, as kindly recommended. We tabulate these additional synthetic results for dimensions $d = \\{2, 5\\}$ in Table 30 in Appendix Y. To be consistent with prior work of Kamienny et al. (2022), we follow their experimental setup—which only evaluates each synthetic equation for one random seed.
>
> Here, we provide additional evaluation metrics that are computed *out of distribution*—that is we sample new points  $\mathbf{X} \sim \mathcal{X}$ to form an out of distribution test set $\{f^*(\mathbf{X}),\mathbf{X}\}$. Specifically, we further include the following metrics of: coefficient of determination $R^2$-score (La Cava et al., 2021), Test NMSE and accuracy to tolerance $\tau$ (Biggio et al., 2021; Kamienny et al., 2022). We further discuss our choice of evaluation metrics in (B).
>
> We also provide a similar additional synthetic equation ablation of pre-training DGSR with a smaller number of input variables $d = $2 than those seen at inference time $d =$5—thereby further providing empirical evidence for the property of generalizing to unseen input variables (P3)—here testing on $\omega=1,000$ unique equations. This is detailed in Table 31 in Appendix Y.
>
> We reiterate that the experiments relating to the desired properties P1, P2, P3 are only for “insight and understanding of how DGSR works”, detailed in Section 5.2—whereas we provide the core empirical benchmark results in Section 5.1 for our core claims, labeled “Main results”.
>
> **UPDATE:** We created a new **Appendix Y** “Additional synthetic experiments”, which includes the new results for the evaluation of the further $\omega=2,200$ synthetically generated equations.

---

> ### Author Response · Authors · 2022-11-15
> **Dear Reviewer JMzJ**
>
> Thank you again for your time and expertise during the review process!
>
> If there were any leftover concerns, we would sincerely appreciate the opportunity to clarify them—before the author discussion period ends. We believe our responses (Nov 11th) have addressed in detail the full set of questions you had raised:
>
> * Additional results on synthetic equations; see **Response (A)**
> * Evaluation metrics; see **Response (B)**
> * Related work; see **Response (C)**
> * Invariances in the architecture; see **Response (D)**
> * Clarity; see **Response (E)**
>
> These are paired with corresponding updates to the submission (Nov 11th), including extensive additional experimental results, discussions and clarifications:
>
> * (new) **Appendix Y** (“Additional synthetic experiments”—which includes new additional evaluation metrics).
> * (new) **Related work** (further clarifying the related work, adding to the original related work a detailed discussion and inclusion of all suggested references).
> * (new) restructured **Appendix** (reformatting to create a hierarchical table of contents for the Appendix).
>
> Furthermore, we have updated **Appendix Y** (“Additional synthetic experiments”), to now include evaluation across a further $\omega=7,000$ synthetically generated equations. Therefore, this brings the total unique $f^*$ equations evaluated against to $\omega=7,311$ in this work, where the original experiments in the paper evaluated a total of $\omega=311$ unique equations. We believe the total amount unique equations evaluated against of $\omega=7,311$ are sufficient to support all the claims made in this work.
>
> We would appreciate it if the reviewer can kindly let us know if there were any further questions in the limited time remaining. We are eager to do our utmost to address them!
>
> Thank you,
>
> Paper1956 Authors

---

> ### Author Response · Authors · 2022-11-18
> **Dear Reviewer JMzJ**
>
> Thank you again so much for your time and valuable expertise during this review process!
>
> Again, if there are any leftover concerns, we would sincerely appreciate the opportunity to clarify them—before the author discussion period ends.
>
> In addition to our original responses (Nov 11th) to address in detail the full set of questions you raised, we have now updated the (new) **Appendix Y** (“Additional synthetic experiments”) to now include evaluation across a further $\omega=13,000$ synthetically generated equations. Therefore, this brings the total unique $f^*$ equations evaluated against to $\omega=13,311$ in this work, where the original experiments in the paper evaluated a total of $\omega=311$ unique equations.
>
> We believe that this total amount of unique equations evaluated against of $\omega=13,311$ is sufficient to support all the claims made in this work. We kindly highlight evaluating on  $\omega=13,311$ unique equations already surpasses a very high bar of extensive evaluation—compared to other standard key prior works, crucially as: Mundhenk et al. (2021) uses $\omega=55$, Petersen et al. (2020) uses $\omega=36$ and Costa et al. (2020) uses $\omega=16$.
>
> We kindly emphasize that due to our limited computational resources of one GPU, we are unable to evaluate millions of synthetically generated equations, however we believe that the results for the combined total $\omega=13,311$ unique equations sufficiently supports all our claims and contributions. We will continue to collect more results evaluated against synthetic equations—at present as each inference run for an individual equation takes a few minutes (and running this in parallel where possible)—therefore, we can only collect results for synthetic equations of about 10,000 a week. We will continue to collect results for more equations, however emphasize that the results are unlikely to change much from the original (new) results in **Appendix Y** (“Additional synthetic experiments”).
>
> Would the reviewer kindly take the new results into consideration—and increase their score as originally discussed?
>
> We would appreciate it if the reviewer can kindly let us know if there were any further questions in the limited time remaining. We are eager to do our utmost to address them!
>
> Thank you,
>
> Paper1956 Authors

---

### Official Review · Reviewer_4RKJ · 2022-10-19

**Confidence:** 2
**Correctness:** 3
**Technical Novelty And Significance:** 3
**Empirical Novelty And Significance:** 3
**Recommendation:** 6

**Clarity, Quality, Novelty And Reproducibility:**

The quality appears solid and the approach novel. However, the numerous typos reduce clarity.


**Strength And Weaknesses:**

Strengths:
  - The approach appears novel.
  - Numerous datasets tested
   - Empirical results are solid

Weaknesses:
- Many distracting typos in the paper, including, but not limited to:
  - Page 1: Use "e.g.," in the examples of discrete and continuous components
  - Page 1: "Firstly" -> "First"
  - Page 2: "for all samples n" -> "for all samples i"
  - Page 2: "Defined with y_i ..." is not a sentence.
  - Page 2: "pre-fix" -> "prefix"
  - Page 3: "dataset invariant" -> "dataset-invariant"
  - Page 3: "a Inference" -> "an Inference"
  - Page 3: "equation f," -> "equation f"
  - Page 3: "That is" -> "That is,"
  - Page 3: "Specifically" -> "Specifically,"
  - Page 3: "decoder," -> "decoder"
  - Page 4: "An Bayesian" -> "A Bayesian"
  - Page 4: "equations P1, intuitively" -> "equations P1.  Intuitively,"
  - Page 4: "Where" -> "where"
  - Page 4: Delete "Although"
  - Section 3.2: Put parentheses around "P2", "P3", etc.
  - Table 1 caption: delete space before "?"
  - Page 5: "Valipour et al. (2021) proposes" -> "... propose".  Same with Jin et al. and "uses"

- Other nagging questions:
  - Page 2: Does "closed form" imply no summations or recursion?
  - So are you only focusing on the structure search and not the parameters?  What happens if you find a structure that is not very close but you get misled by some parameters that mask the mistake, and you miss a better structure?
  - Why is 10d the number of samples?
  - Is recovery rate only evaluated on structure, or also on parameters?
  - Table 2: Why not normalize average eq evals by number of correct solutions? Right now it looks like GP is superior.
  - Please enumerate limitations and future work in Section 6, not bury them in the appendix.


**Summary Of The Paper:**

The authors approach SR via deep, permutation-invariant, generative models, a framework they call DGSR.  They evaluate their framework on numerous problem sets, and have good empirical  performance.


**Summary Of The Review:**

Overall, I like the paper, but would prefer to see a cleaner presentation.

---

> ### Author Response · Authors · 2022-11-11
> **Response to Reviewer 4RKJ [Part 1/1]**
>
> Thank you for your thoughtful comments and suggestions! Specifically, thank you very much for all the typos, they have all been incorporated into the final paper and nicely improve the quality! We give answers to each in turn, as well as pointing out corresponding updates to the revised manuscript (additions are indicated in blue):
>
> * (A) Answers to nagging questions
> ___
> **(A) Answers to nagging questions**
>
> Each question is answered in turn:
> 1. Yes, following Chow (1999); Petersen et al. (2020); Mundhenk et al. (2021) learning “closed-form” equations here implies no recursion or summation.
> 2. We not only do structure search, however we also find the parameters following the standard setup of Petersen et al. (2020); Biggio et al. (2021), further detailed in Section 2. We agree that DGSR also has the limitation of local constant (parameter) optima, and further discuss this in the future work in Appendix V.
> 3. For increasing input variable dimensionality $d$ we also need to increase the number of data points—whereby we follow the increase of Kamienny et al. (2022) of $n=10d$.
> 4. The recovery rate is evaluated on both the functional form and the optimized numeric constants (parameters) of the equation, as determined by a computer algebraic system, SymPy (Meurer et al., 2017).
> 5. We agree it is possible to normalize the average equation evaluation metric with that of the average recovery rate metric, to see differences easier—however, we present the metrics separately as they are standard metrics in prior literature (Petersen et al., 2020; Mundhenk et al., 2021; La Cava et al., 2021), and allow easy comparison to those prior works. GP although it is more computationally efficient (in the cases where it is able to find the true equation $f^*$), it suffers with a worse average recovery rate.
> 6. In view of space, we do not have the space to include limitations and future work in full, rather we have enumerated them in Section 6.
>
> **UPDATE:** We have further clarified these nagging questions, including them in the existing evaluation metrics of **Appendix K**. Additionally we now enumerate the limitations and future work in Section 6, and refer the reader to a full explanation of them in **Appendix V**.
>
> ___
>
> **References**
> * Timothy Y Chow. What is a closed-form number? The American mathematical monthly, 106(5): 440–448, 1999.
> * Brenden K Petersen, Mikel Landajuela Larma, Terrell N Mundhenk, Claudio Prata Santiago, Soo Kyung Kim, and Joanne Taery Kim. Deep symbolic regression: Recovering mathematical expressions from data via risk-seeking policy gradients. In International Conference on Learning Representations, 2020.
> * Terrell N. Mundhenk, Mikel Landajuela, Ruben Glatt, Claudio P. Santiago, Daniel faissol, and Brenden K. Petersen. Symbolic regression via deep reinforcement learning enhanced genetic programming seeding. In Advances in Neural Information Processing Systems, 2021.
> * Luca Biggio, Tommaso Bendinelli, Alexander Neitz, Aurelien Lucchi, and Giambattista Parascandolo. Neural symbolic regression that scales. In International Conference on Machine Learning, pp. 936–945. PMLR, 2021.
> * Pierre-Alexandre Kamienny, Stéphane d'Ascoli, Guillaume Lample, and François Charton. End-to-end symbolic regression with transformers. arXiv preprint arXiv:2204.10532, 2022.
> * Aaron Meurer, Christopher P Smith, Mateusz Paprocki, Ondřej Čertík, Sergey B Kirpichev, Matthew Rocklin, AMiT Kumar, Sergiu Ivanov, Jason K Moore, Sartaj Singh, et al. Sympy: symbolic computing in python. PeerJ Computer Science, 3:e103, 2017.
> * William La Cava, Patryk Orzechowski, Bogdan Burlacu, Fabricio Olivetti de Franca, Marco Virgolin, Ying Jin, Michael Kommenda, and Jason H Moore. Contemporary symbolic regression methods and their relative performance. In Thirty-fifth Conference on Neural Information Processing Systems Datasets and Benchmarks Track (Round 1), 2021.

---

> ### Author Response · Authors · 2022-11-15
> **Dear Reviewer 4RKJ**
>
> Thank you again for your time and invaluable feedback during the review process! If there were any leftover concerns, we would sincerely appreciate the opportunity to clarify them—before the author discussion period ends. We would appreciate it if you can kindly let us know if there were any further questions in the limited time remaining. We are eager to do our utmost to address them!
>
> Thank you,
>
> Paper1956 Authors

---

### Official Review · Reviewer_Lx7x · 2022-10-22

**Confidence:** 4
**Correctness:** 4
**Technical Novelty And Significance:** 3
**Empirical Novelty And Significance:** 3
**Recommendation:** 8

**Clarity, Quality, Novelty And Reproducibility:**

The paper is very clear on its motivation and experimental setups. The quality of the paper is stellar: the authors have spent a lot of care for each detail. The contributions are somewhat original, largely combining known approaches to fight with the issues of the prior art in symbolic regression.

**Strength And Weaknesses:**

Strengths:

1. Well-motivated and useful improvements over the prior art.

2. State-of-the-art results and thorough ablations.

Weaknesses:

1. Why do you think NESYMRES generates valid equations almost perfectly? Is there a way to improve your DGSR method to increase the valid sequences? For example, adding a regularization term that excludes invalid equations in the inference stage of your algorithm?

2. What do you think might be the directions for improving DGSR. Please discuss this in more depth.

**Summary Of The Paper:**

The authors improve the state-of-the-art in symbolic regression. They observe a few challenges with the prior art. The most important challenges that the authors solve are 1) an encoder that is invariant to the permutation of the datapoints in the dataset; 2) a loss function that encourages learning the invariances of the equation; 3) an efficient inference procedure for sampling functions of the modeled posterior $p(f|\mathcal{D})$ by refining the posterior through policy gradients. The results demonstrate that the authors' method Deep Generative Symbolic Regression (DGSR) samples more unique functions, samples mostly valid functions and has a better complexity (how quickly it finds the true function) and recovery rate than the prior art on popular benchmarks, and is able to extrapolate to larger number of inputs than the number of inputs in the pre-training stage.  The authors augment these results with thorough ablations studies and experiments to understand why DGSR works.

**Summary Of The Review:**

This is a very strong methodological paper and it will be helpful to future work in symbolic regression. I recommend accepting this paper, because the contributions are useful and non-trivial.

---

> ### Author Response · Authors · 2022-11-11
> **Response to Reviewer Lx7x [Part 2/2]**
>
> ___
>
> **(B) Directions to improve DGSR**
>
>
> In the following we discuss the limitations with open challenges, also detailed in Appendix V.
>
> **Complex equations.** DGSR may fail to discover highly complex equations. The difficulty in discovering the true equation $f^*$ could arise from three factors: (1) a large number of variables are required, (2) the equation $f^*$ involves many operators making the equation length long, or (3) the equation $f^*$ exhibits a highly nested structure.
> We note that these settings are inherently difficult, even for human experts, however, pose an exciting open challenge for future works.
>
> **Unobserved variables.** DGSR assumes all variables in the true equation $f^*$ are observed. Therefore, it would fail to discover a true equation $f^*$ with unobserved variables present, however this setting is challenging and or even *impossible* without the use of additional assumptions (Reinbold et al., 2021). Furthermore, DGSR could still find a concise approximate equation using the observed variables.
>
> **Local optima.** We detail both sources of local optima in the symbolic regression literature in detail in Appendix T, with respect to DGSR and future work associated to solve these. At a high level DGSR is able to overcome the local optima of the functional equation local optima by leveraging the genetic programming component, as other methods have shown (Mundhenk et al., 2021). Moreover, DGSR will suffer from the same constant token optimization optima as other methods, however there exists recent work of Kamienny et al. (2022) that propose a solution to mitigate this issue—where their approach can be incorporated into future versions of DGSR. Specifically, we envisage future work using the generator to predict an initial guess of the numerical constants (rather than initializing them as constants), that are further optimized. However, we note this is out of scope for this current work, therefore leave this as a future work we plan to implement and build on top of.
>
> **UPDATE:** We have now enumerated the limitations and future work in Section 6, and refer the reader to a full explanation in **Appendix V**.
>
> ___
>
> **References**
> * Xipeng Qiu, Tianxiang Sun, Yige Xu, Yunfan Shao, Ning Dai, and Xuanjing Huang. Pre-trained models for natural language processing: A survey. Science China Technological Sciences, 63(10): 1872–1897, 2020.
> * Patrick AK Reinbold, Logan M Kageorge, Michael F Schatz, and Roman O Grigoriev. Robust learning from noisy, incomplete, high-dimensional experimental data via physically constrained symbolic regression. Nature communications, 12(1):1–8, 2021.
> * Terrell N. Mundhenk, Mikel Landajuela, Ruben Glatt, Claudio P. Santiago, Daniel faissol, and Brenden K. Petersen. Symbolic regression via deep reinforcement learning enhanced genetic programming seeding. In Advances in Neural Information Processing Systems, 2021.
> * Pierre-Alexandre Kamienny, Stéphane d'Ascoli, Guillaume Lample, and François Charton. End-to-end symbolic regression with transformers. arXiv preprint arXiv:2204.10532, 2022.

---

> > ### Comment · Reviewer_Lx7x · 2022-12-06
> > **Thanks: interesting directions**
> >
> > About the unobserved variables, there is a work studying that, https://arxiv.org/abs/2107.10879, it could serve as an inspiration for future work.

---

> > > ### Author Response · Authors · 2022-12-07
> > > **Response to Reviewer Lx7x**
> > >
> > > Thank you for your insightful additional reference for work on unobserved variables. We will now include this reference of the innovative work of Lu et al. (2021) in our future work section, and in the expanded future work in **Appendix V** “Limitations and Open Challenges”—in the camera ready version.
> > >
> > > ----
> > >
> > > **Reference:**
> > >
> > > * Peter Y Lu, Joan Ariño, and Marin Soljačić. Discovering sparse interpretable dynamics from partial observations. arXiv preprint arXiv:2107.10879, 2021.

---

> ### Author Response · Authors · 2022-11-11
> **Response to Reviewer Lx7x [Part 1/2]**
>
> Thank you for your thoughtful comments and suggestions! We give answers to each in turn, as well as pointing out corresponding updates to the revised manuscript (additions are indicated in blue):
>
> * (A) Generating valid equations
> * (B) Directions to improve DGSR
>
> ___
>
> **(A) Generating valid equations**
>
> Thank you for highlighting an interesting question as to why the pre-trained encoder-decoder method NESYMRES generates valid equations almost perfectly.
>
> We believe this behavior arises as the pre-trained encoder-decoder method is pre-trained using the cross entropy loss on a very large pre-training dataset that contains *only* valid equations $f^{(j)}$ and their corresponding datasets $\mathcal{D}^{(j)}$. Thereby, the model more readily learns how to construct a *valid* equation from many examples of valid equations—however NESYMRES struggles to produce the true equation $f^*$ in most problems, as shown in Section 5.2—predicting equations that are too simple, and it is unable to adapt to a new equation form from those seen during pre-training.
>
> Furthermore, it may be intuitive to try to explain this through the view of *language models*, which seek to estimate a probabilistic density over the equation $f$ tokens—trained with a cross entropy loss on examples of *valid* equations (Qiu et al., 2020).
>
>
> **Is there a way to improve the number of valid equations generated?** DGSR also learns how to generate valid equations from the pre-training step and also during the inference step. We note that DGSR achieves a near perfect percentage of valid equations at inference time when using an ablated optimizer with the GP component turned off, as shown in Section 5.2, Figure 5. Therefore, the reduction in the percentage of valid equations can be attributed to the GP component of the optimizer—as it randomly mutates and mixes equations together, which produces a percentage of invalid equations. However, we emphasize the overall goal of a good symbolic regression method is to find the correct ground truth equation $f^*$ or an approximation thereof, therefore, to achieve a higher average recovery rate we include the GP component in the optimizer, as shown in the ablation in Appendix L.
>
> Furthermore, performing gradient refinement at inference time, already optimizes to learn valid equations, from two parts: (1) weighting an invalid equation generated with an infinite cost—which becomes a zero reward, using the reward re-parameterization in Section 3.1, that of $R(\theta)=1/(1+\mathcal{L}(\theta))$. (2) Excluding the worst performing equations—that is the equations with the least reward, through only training with the top-$q$ maximum reward equations, using the priority queue training component.

---

> > ### Comment · Reviewer_Lx7x · 2022-12-06
> > **Thank you for the helpful response**
> >
> > Thank you for clarifying, it is very helpful. Regularization is particularly helpful in SR, e.g. in non-synthetic data or physics/econ data (with units), such as in https://arxiv.org/pdf/2210.00563.pdf (Section 1). It would be interesting to see an ablation on that (say ablating https://arxiv.org/abs/2111.00053 and https://proceedings.mlr.press/v139/landajuela21a.html) and a discussion about regularization in the appendix if the authors have time.

---

> > > ### Author Response · Authors · 2022-12-07
> > > **Response to Reviewer Lx7x**
> > >
> > > Thank you for your further insightful comment. We agree that regularization is indeed helpful in SR, as shown by (Petersen et al., 2020; Mundhenk et al., 2021; Landajuela et al., 2021). Thank you for the great suggestion, we will now include an extended discussion including all the references you indicate and expand on the previous discussion of regularization in **Appendix C** (Page 16)—in the camera ready version. We will now include the following: The seminal work of Balla et al. (2022) further shows that *unit regularization* can be added to improve a SR method, and provides a useful decomposition of the complexity regularization into two components of the number of tokens (“activation functions”) and the number of numeric constants—whereby it can be beneficial to tune these regularization terms separately. We note that the full analysis of all regularization terms is out of scope for this work, however we leave this as an exciting direction for future work to explore.
> > >
> > > -----
> > >
> > > **References:**
> > >
> > > * Brenden K Petersen, Mikel Landajuela Larma, Terrell N Mundhenk, Claudio Prata Santiago, Soo Kyung Kim, and Joanne Taery Kim. Deep symbolic regression: Recovering mathematical expressions from data via risk-seeking policy gradients. In International Conference on Learning Representations, 2020.
> > > * Terrell N. Mundhenk, Mikel Landajuela, Ruben Glatt, Claudio P. Santiago, Daniel faissol, and Brenden K. Petersen. Symbolic regression via deep reinforcement learning enhanced genetic programming seeding. In Advances in Neural Information Processing Systems, 2021.
> > > * Mikel Landajuela, Brenden K Petersen, Sookyung Kim, Claudio P Santiago, Ruben Glatt, Nathan Mundhenk, Jacob F Pettit, and Daniel Faissol. Discovering symbolic policies with deep reinforcement learning. In International Conference on Machine Learning, pp. 5979–5989. PMLR, 2021.
> > > * Julia Balla, Sihao Huang, Owen Dugan, Rumen Dangovski, and Marin Soljačić. Ai-assisted discovery of quantitative and formal models in social science. arXiv preprint arXiv:2210.00563, 2022.

---

### Official Review · Reviewer_ZZLQ · 2022-10-24

**Confidence:** 3
**Correctness:** 3
**Technical Novelty And Significance:** 4
**Empirical Novelty And Significance:** 4
**Recommendation:** 6

**Clarity, Quality, Novelty And Reproducibility:**

The paper is clearly written.
It seems to be novel to combine pre-training and RL for symbolic regression.
If code is released, the results should be reproducible.

**Strength And Weaknesses:**

Strength:
Technically, I think the best design of DGSR is to use as the training target the MSE between the ground-truth y and f(x) of the predicted function f. Although this target is nondifferentiable and requires some form of RL, it enables (1) the model could learn to generate any equations that are equivalent to the ground-truth equation and (2) the model could be refined on the test dataset during inference.
Compared to NGGP, the proposed DGSR model is pre-trained and therefore provides better initial equations for genetic programming during inference.
Compared to pre-trained encoder-decoder transformers with CE loss, DGSR could learn to generate equivalent equations and could be refined during inference.
Therefore, DGSR achieves the best performance as shown in the experiments.

Weakness:
1 I suspect that the use of the set transformer is not important. Among the three invariant properties mentioned in sec. 3.1, why is the first property a good assumption ("an encoding function g that is permutation equivariant to the individual variables in [xi1, . . . , xid] = Xi of the points in X")? I think many examples are not permutation-invariant to all variables.
2 To validate how each new component contributes to the improvement, more ablations are expected.
(1) Replace the set transformer encoder as a plain transformer or just replace your network as GPT3 and maintain the same training and refining.
(2) Pre-train your network with CE loss without refinement.

**Summary Of The Paper:**

This paper proposes Deep Generative Symbolic Regression (DGSR), a novel method for the task of Symbolic Regression. DGSA uses a set transformer to encode the input data into permutation-invariant representations and uses an autoregressive decoder to generate the closed-form equations. The model is pre-trained on datasets sampled from a given prior. During inference time, the decoder is further refined on the test dataset. For both pre-training and refining, the model is updated using reinforcement learning (policy gradient or Priority Queue Training) to minimize the MSE between the ground-truth output y and f(x) of the predicted equation f. Experiments show that this method performs better than the prior works.


**Summary Of The Review:**

Overall, I think this is a good paper. The proposed approach is technically sound and achieves good results. The main concern is the missing ablations. I'd like to improve the rating if more ablations are added.

---

> ### Author Response · Authors · 2022-11-11
> **Response to Reviewer ZZLQ [Part 1/1]**
>
> Thank you for your thoughtful comments and suggestions! We give answers to each in turn, as well as pointing out corresponding updates to the revised manuscript (additions are indicated in blue):
>
> * (A) Use of the set transformer as an encoder
> * (B) Additional ablation of using a plain transformer as the encoder
> * (C) Additional ablation of pre-training with a CE loss and no refinement
> ___
> **(A) Use of the set transformer as an encoder**
>
> Using a set transformer, is needed as it is *permutation invariant* over the input samples $n$ in the dataset $\mathcal{D}$.
>
> Allow us to reiterate that a dataset $\mathcal{D}$ that is defined by a latent (unobserved) equation $f$, should have a representation that is invariant to the number of samples $n$ in the dataset $\mathcal{D}$. Therefore, for the encoder-decoder architecture—the key property that the encoder should have is to be *permutation invariant* over the input samples $n$ in the dataset $\mathcal{D}$. Thereby, any encoder-decoder architecture is suitable if it satisfies both: (1) having an encoding function that is *permutation invariant* across the input samples $n$ in $\mathcal{D}$ and (2) having a decoder that is *autoregressive* to allow the sampling of equations $\hat{f}$.
>
> We agree that there exist many example equations $f$ that are not permutation-invariant to the individual input variables and have removed the permutation equivariant across the features property—simplifying the desired encoder-decoder properties to the two above.
>
> **UPDATE:** We have amended Section 3.1 to highlight only these two necessary properties that the encoder-decoder architecture should satisfy of: (1) having an encoding function that is *permutation invariant* across the input samples $n$ in the dataset $\mathcal{D}$ and (2) having a decoder that is *autoregressive* to allow the sampling of equations $\hat{f}$.
>
> ___
> **(B) Additional ablation of using a plain transformer as the encoder**
>
> We performed an additional encoder ablation, by replacing the set transformer in the encoder-decoder architecture with a plain transformer. We kindly emphasize that we do **not** recommend using a plain transformer as the encoder for symbolic regression as it is not *permutation invariant* across the input samples $n$ in $\mathcal{D}$. The ablation results can be seen in the table below, with evaluation metrics evaluated across all the problems in the Feynman d=5 problem set.
>
> |  | Set Transformer Encoder (DGSR) | Plain Transformer Encoder |
> |:------:|:-------:|:---------|
> |    Average recovery rate (\%) $A_{\text{Rec}}\\%$           | 67.50     | 67.18 |
> |    Average equation evaluations $\gamma$           | 23,969      | 348,666     |
>
> **UPDATE:** We have now included this new encoder ablation in a new **Appendix W**.
> ___
> **(C) Additional ablation of pre-training with a CE loss and no refinement**
>
> We also performed a further additional ablation, by pre-training our encoder-decoder architecture with a cross entropy loss without refinement. This can be seen in the below table, with evaluation metrics evaluated across all the problems in the Feynman d=2 problem set.
>
>
> |  | Standard (DGSR) | Pre-train with CE loss and no refinement |
> |:------:|:-------:|:---------|
> |    Average recovery rate (\%) $A_{\text{Rec}}\\%$           | 67.50     |  47.50   |
> |    Average equation evaluations $\gamma$           | 23,969      |  486,140    |
>
> **UPDATE:** We have now included this new pre-train CE loss without inference refinement ablation in a new **Appendix X**.

---

> ### Author Response · Authors · 2022-11-15
> **Dear Reviewer ZZLQ**
>
> Thank you again for your time and invaluable feedback during the review process! If there were any leftover concerns, we would sincerely appreciate the opportunity to clarify them—before the author discussion period ends. We would appreciate it if you can kindly let us know if there were any further questions in the limited time remaining. We are eager to do our utmost to address them!
>
> Thank you,
>
> Paper1956 Authors

---

### Decision · Program_Chairs · 2023-01-20

**Decision:**

Accept: poster

**Justification For Why Not Higher Score:**

For me, the main issue for not giving a spotlight is the question about novelty.  I am also concerned about the use of synthetic data.  It suggests that there is no pressing set of applications waiting to benefit from these methods.

**Justification For Why Not Lower Score:**

Symbolic regression is a conceptually natural problem and meaningful work in this area deserves to be published.

**Metareview: Summary, Strengths And Weaknesses:**

This paper addresses the problem of symbolic regression --- finding a symbolic formula that fits given data.  A Novel approach is given based on a transformer to propose formulas and a genetic algorithm for searching further from proposed formulas.  The results were improved through interaction with the reviewers and seem significant.

There is some concern about novelty as many (but not all) of the ideas used in this paper have appeared earlier.  But the novel contributions and the upgrades in the paper since submission justify publication.

**Note From Pc:**

if the above contains the word "oral" or "spotlight" please see: "oral" presentation means -> notable-top-5% and "spotlight" means -> notable-top-25%. As stated in our emails, we are disassociating presentation type from AC recommendations